



# Aggregation effects on tritium-based mean transit times and young water fractions in spatially heterogeneous catchments and groundwater systems, and implications for past and future applications of tritium

Michael K. Stewart[1], Uwe Morgenstern[2], Maksym A. Gusyev[3,4], Piotr Maloszewski[5]

[1]Aquifer Dynamics & GNS Science, P.O. Box 30368, Lower Hutt, 5040, New Zealand
[2]GNS Science, Tritium & Water Dating Laboratory, Avalon, Lower Hutt, 5040, New Zealand
[3]International Centre for Water Hazard and Risk Management (ICHARM), Public Works Research Institute (PWRI), Tsukuba, Japan
[4]National Graduate Institute for Policy Studies (GRIPS), Tokyo, Japan
[5]AGH University of Science and Technology Cracow, Department of Hydrogeology and Engineering Geology, Al. Mickiewicza 30, 30-059 Cracow, Poland

*Correspondence to*: Michael K. Stewart (m.stewart@gns.cri.nz)

**Abstract.** Applications of simple lumped parameter models to describe aspects of hydrological systems rest on assumptions of homogeneity that are rarely valid. The lumped parameters are supposed to represent the quantities within the system as well as those of the overall system, but such quantities will obviously vary greatly from place to place within heterogeneous systems. Less appreciated is the fact that aggregation errors will affect overall system parameters as well. Kirchner (2016a) recently demonstrated that aggregation errors due to heterogeneity in catchments could cause severe underestimation of the mean transit times (MTTs) of water travelling through catchments when simple lumped parameter models were applied to interpret seasonal tracer cycles. Here we examine the effects of such errors on the MTTs and young water fractions estimated using tritium concentrations. We find that MTTs derived from tritium concentrations in streamflow are just as susceptible to aggregation bias as those from seasonal tracer cycles. Likewise, groundwater wells or springs fed by two or more water sources with different MTTs will also show aggregation bias. However, the transit times over which the biases are manifested are very different; for seasonal tracer cycles it is 2-3 months up to about 5 years, while for tritium concentrations it is 6-12 years up to about 200 years. We also find that young water fractions derived from tritium are almost immune to aggregation errors as were those derived from seasonal tracer cycles.

To investigate the implications of these findings for past and future use of tritium for estimating MTTs in catchments and groundwater systems, we examined case studies from the literature in which simple and more complicated lumped parameter models had been used. We find that MTT aggregation errors are small when either component waters are young (less than 6-12 years, as found in many catchments), or component waters have similar MTTs to each other. On the other hand, aggregation errors are large when very young water components are mixed with old components. In general, well-chosen compound lumped parameter models should be used as they will eliminate potential aggregation errors due to the application of simple lumped




parameter models. The choice of a suitable lumped parameter model can be assisted by matching simulations to time series of tritium measurements (underlining the value of long series of tritium measurements), but such results should also be finally validated to ensure that the parameters found by modelling correspond to reality.

## 1 Introduction

Water can take very complex flow pathways in catchments through shallow and deep aquifers. Environmental tracers are commonly used to obtain transit time distributions (TTDs) in groundwater systems (Maloszewski and Zuber, 1982) or catchments (McDonnell et al., 2010). Transit time is the time it takes for rainfall to travel through a system and emerge in a well, spring or stream. TTDs provide important information about transport, mixing and storage of water in systems and therefore on the retention and release of pollutants. Mean transit times (MTTs) determined from these distributions provide
practical information for various aspects of water resources management. For example, MTTs have been used to estimate the volume of groundwater storage providing baseflow in catchments (Morgenstern et al., 2010; Gusyev et al., 2016) and to predict lag times and life expectancies of contaminants in the subsurface (Hrachowitz et al., 2016). The drinking water securities of wells in New Zealand are partly assessed by an absence of water with less than one-year travel time by the NZ drinking water quality standard (Ministry of Health, 2008). As useful as they are, TTDs cannot be measured directly in the field and have to
be inferred from tracer concentrations with the use of lumped parameter models (LPMs).

Catchments are inherently heterogeneous on various scales. Point-scale properties vary greatly from place to place, while streams integrate the various catchment outputs. The top-down approach uses catchment outputs, such as streamflow and stream chemistry, to infer or predict catchment TTDs. The hope is that these average out local heterogeneities allowing one simple LPM to provide a good fit and its parameters to be representative of the catchment (a well-known relatively successful
application of an LPM to model catchment acidification is described by Cosby et al., 1985). But individual areas within catchments can vary greatly because of geology, geography, aspect, etc. Groundwater systems also show heterogeneity. Kirchner (2016a) showed by means of virtual experiments that aggregating subcatchments with different TTDs can lead to severe underestimation of the composite MTT when simple LPMs were applied to interpret seasonal tracer cycles. This is because the smoothing out of the seasonal cycles is a non-linear process which acts more rapidly on the younger water
components thereby causing underestimation of the composite MTT. He also found that the young water fraction was a much more robust metric than the MTT against aggregation error using seasonal tracer cycles. These results raise an important question: Are tritium-derived MTTs also susceptible to aggregation error due to spatial heterogeneity?

Hence, examination of the effects of aggregating differing flows on tritium-based MTTs is needed to test the applicability of tritium-based MTTs in catchments and groundwater systems. Seasonal tracer cycle and tritium-based MTTs are determined
by different methods and have given very different results in catchments. The seasonal tracer cycle method depends on damping of input cycles on passing through a system into the output, whereas the tritium method depends on radioactive decay of tritium between input and output (with half-life of 12.32 yr). Effects of mixing within systems also need to be accounted





for in both cases (Maloszewski and Zuber, 1982). Results from seasonal tracer cycles have given MTTs up to about 5 years at which point the input cycles are completely damped within tracer measurement errors, while results from tritium measurements show that large proportions of the flow in many streams have MTTs of one to two decades or more (Stewart et al., 2010; Seegar and Weiler, 2014; Michel et al., 2015). This difference makes it clear that TTDs determined from seasonal tracer cycles often underestimate the real MTTs in streams. Such underestimation is due to the non-linearity of the damping of the seasonal tracer cycles as noted above. Similarly, radioactive decay of tritium is a non-linear process and therefore spatial aggregation errors are expected when water components with different ages are combined.

Calibration of LPMs using environmental radioisotope and stable isotope data has been the subject of study for many years (see Maloszewski and Zuber (1982) and early work summarised therein). If a catchment outflow is a mixture of two (or possibly more) components of different water ages, it is impossible to calibrate an LPM uniquely when we only have data for the one tracer. For example, for springs in Czatkowice, Poland, only when the proportion in which the water components (water fluxes) were mixed was known could the unique answer based on tritium measurements be found (Grabczak et al., 1984; Małoszewski and Zuber, 1993). In heterogeneous catchments, it is always required to (i) measure a variable tracer periodically, and (ii) to combine those data with water fluxes in the inputs and outputs to separate "fast" and "slow" components; see for example studies at Lainbach Valley, Germany (Maloszewski et al., 1983), and Schneealpe, Austria (Maloszewski et al., 2002). The choice of LPM, or equivalently the TTD function, must be based more on the hydrogeological situation and not on artificial mathematical (fitting) considerations. Calculation of hydrological parameters known independently (e.g. mean thickness of the water bearing layers in the catchment) is required for model validation in order to examine if the model used is applicable to the real natural situation. We can have a very well-calibrated model in terms of tracer data being fitted by an LPM, but the MTT can be far from the hydrological reality.

The aim of this paper is to examine the aggregation effects of spatially heterogeneous catchments and groundwater systems on MTTs and young water fractions determined using tritium concentrations. We conducted our investigation by combining dissimilar water components and comparing the true mixed MTTs with the tritium-inferred apparent MTTs, as Kirchner (2016a) did with seasonal tracer cycles. We also examined aggregation effects for young water fractions estimated using tritium. Our calculations are based on the gamma LPM with shape factors ($\alpha$) between 0.3 and 10, which is also representative of other frequently-used simple LPMs such as the exponential, exponential-piston and dispersion models. The different tritium input functions for Northern and Southern Hemisphere locations were tested. We also surveyed some applications of tritium dating from the literature, which had sufficient data to constrain the parameters of mixtures of young and old waters. MTTs from simple and compound LPMs applied to the data were compared to examine the aggregation errors for these real examples. This has allowed us to consider the practical implications of our findings and provide guidance for tritium sampling and interpretation in heterogeneous catchments and groundwater systems.

Our experiments did not include examination of non-stationary hydrological systems. Kirchner (2016b) had found similar underestimation of MTTs and robustness of young water fractions based on seasonal tracer cycles in a non-stationary system





(a two-box model). We briefly consider non-stationary aggregation effects in the light of the differences in the tritium and seasonal tracer cycles methods.

## 2 Methods

### 2.1 Transit time determination

The different flow paths of water through the subsurface of catchments imply that outflows contain water with different transit times. i.e. The water in the stream does not have a discrete age, but has a distribution of ages. This distribution is often described by a conceptual flow or mixing model (LPM), which reflects the average conditions in the catchment.

Rainfall incident on a catchment is affected by immediate surface/near surface runoff and longer-term evapotranspiration loss. The remainder constitutes recharge to the subsurface water stores. Tracer inputs to the subsurface water stores (i.e. seasonal

tracer cycles and tritium concentrations in the recharge water) are modified during passage through the hydrological system by mixing of water of different ages (represented by a flow model) and radioactive decay in the case of tritium before appearing in the output. The convolution integral and an appropriate flow model are used to relate the tracer input and output. The convolution integral is given by

$$C_{out}(t) = \int_0^\infty C_{in}(t - \tau) h(\tau) \exp(-\lambda \tau) \, d\tau \tag{1}$$

where $C_{in}$ and $C_{out}$ are the input and output concentrations in the precipitation and baseflow respectively. $t$ is calendar time and the integration is carried out over the transit times $\tau$. $h(\tau)$ is the transit time distribution (TTD) function of the hydrological system constructed based on the distribution of the water fluxes in the catchment (flow model). The exponential term accounts for radioactive decay of tritium. ($\lambda$ is the tritium decay constant (= $\ln 2/T_{\frac{1}{2}}$, where $T_{\frac{1}{2}}$ is the half-life of tritium (12.32 years).) Tritium input ($C_{in}$) was different in each hemisphere. Input functions (tritium concentrations in monthly samples of

precipitation) at Kaitoke, New Zealand in the Southern Hemisphere (Morgenstern and Taylor, 2009) and Trier, Germany in the Northern Hemisphere (IAEA/WMO, 2016) are given as examples in Fig. 1. Both curves have pronounced bomb peaks due to nuclear weapons testing mainly in the Northern Hemisphere during the 1950s and 1960s. The peak was much larger in the Northern Hemisphere than in the Southern Hemisphere. Since then there have been steady declines due to leakage of tritium from the stratosphere into the troposphere followed by removal by rainout and radioactive decay. However, the tritium

concentrations in the troposphere are now reaching the background cosmogenic levels which they had before the dawn of the nuclear age (conventionally taken as 1950). The levelling-out process occurred about 20 years ago in the Southern Hemisphere and 5-10 years ago in the Northern Hemisphere. The bomb peaks have been good markers of 1960s precipitation in past tritium studies, but the steady declines which mimic radioactive decay of tritium have caused problems with ambiguous (i.e. multiple) age estimations for given tritium values (Stewart et al., 2010).

The curves also show smaller variations due to annual peaks in tritium concentrations caused by increased stratospheric leakage during spring in each hemisphere, and to small longer-term variations related to sunspot cycles. Tritium concentrations are





expected to remain at the present cosmogenic levels for the foreseeable future, and this means that tritium is becoming increasingly useful for dating because multiple age solutions are now much less of a problem (Stewart et al., 2012; Stewart and Morgenstern, 2016; Gusyev et al., 2016). Effective use of tritium does require highly sensitive and accurate tritium measurements, however, because the natural cosmogenic tritium concentrations and variations are very low.

Several flow models (LPMs) are commonly used in tracer studies. The piston flow model (PFM) describes systems for which all outputs have the same transit time (MTT or $\tau_m$), i.e. the outputs are not combinations of different component fluxes with different transit times.

$$h(\tau) = \delta(t - \tau_m) \tag{2}$$

and $C_{out}(t) = C_{in}(t - \tau_m)\exp(-\lambda\tau_m)$ (3)

The exponential model (EM) is given by

$$h(\tau) = \frac{1}{\tau_m}\exp(-\frac{\tau}{\tau_m}) \tag{4}$$

where the single parameter is $\tau_m$[yr]. In this model, water parcels with different transit times combine in the outflow to approximate the exponential TTD. It is mathematically equivalent to the well-mixed model (also called the linear reservoir), but it does not imply that full mixing occurs within real systems.

The gamma model (GM) has TTDs based on the gamma distribution

$$h(\tau) = \frac{\tau^{\alpha-1}}{\beta^\alpha \Gamma(\alpha)} e^{-\tau/\beta} \tag{5}$$

where the two parameters $\alpha$[-] and $\beta$[yr$^{-1}$] are shape and scale factors respectively, and $\tau_m = \alpha\beta$ (Kirchner et al., 2000). The gamma distribution reduces to the exponential distribution for the special case of $\alpha = 1$.

The exponential-piston flow model (EPM) combines a volume with exponential transit times followed by a piston flow volume

to give a model with two parameters (Maloszewski and Zuber, 1982). The TTD is given by

$$h(\tau) = 0 \qquad\qquad\qquad \text{for } \tau < \tau_m(1-f) \tag{6a}$$

$$h(\tau) = \frac{1}{f\tau_m}\exp(-\frac{\tau}{f\tau_m} + \frac{1}{f} - 1) \qquad\qquad \text{for } \tau \geq \tau_m(1-f) \tag{6b}$$

where f is the ratio of the exponential volume to the total volume. (Maloszewski and Zuber (1982) used the parameter η; f=1/η.) $\tau_m(1-f)$ is the time required for water to flow through the piston flow section, while $f\tau_m$ is the mean transit time through

the exponential volume.

The dispersion model (DM) assumes a tracer transport which is controlled by advection and dispersion processes (Maloszewski and Zuber, 1982), with a TTD of

$$h(\tau) = \frac{1}{\tau\sqrt{4\pi(P_D)\tau/\tau_m}}exp\left[-\frac{(1-\frac{\tau}{\tau_m})^2}{4(P_D)\tau/\tau_m}\right] \tag{7}$$



where $P_D$[-] is the dispersion parameter (being the measure of the variance of the transit time distribution, i.e. the sum of the variance resulting from the space distribution of the infiltration through the catchment surface and variance resulting from the dispersive flow through the underground). The two parameters are $\tau_m$ and $P_D$.

This paper makes a particular distinction between *simple LPMs* (meaning specifically the GM, EPM with end members PFM
5 and EM, and DM LPMs) and *compound LPMs* (binary or other combinations of LPMs). Compound LPMs have often been used to represent more complicated systems (e.g. Maloszewski et al., 1993; Stewart and Thomas, 2008; Morgenstern et al., 2015). The combination of two EM models in parallel, called the 'double exponential model' when applied to tritium (Michel, 1992; Taylor et al., 1992) and the 'two parallel linear reservoirs' (TPLR) model when applied to seasonal tracer cycles (Weiler et al., 2003), is given by

$$h(\tau) = \frac{b}{\tau_f} \exp\left(-\frac{\tau}{\tau_f}\right) + \frac{(1-b)}{\tau_s} \exp\left(-\frac{\tau}{\tau_s}\right) \qquad (8)$$

where b is the fraction of the young component (fast reservoir), and $\tau_f$ and $\tau_s$ are the MTTs of the fast and slow reservoirs respectively. The model has three parameters with the overall combined MTT ($\tau_m$) being

$$\tau_m = b\tau_f + (1-b)\tau_s \qquad (9)$$

## 2.2 Comparison of transit time distributions of different flow models

The TTDs of the four cases of the GM investigated in this work are illustrated in Fig. 2a, as normalised probability density function (i.e. h(τ) x $\tau_m$) versus normalised transit time (τ/$\tau_m$). These cover a wide range of TTDs observed in streams using seasonal tracer cycles, and streams and groundwater using tritium concentrations. They are also approximately representative of the other simple flow models described above. The GM case α=0.3 has 'heavy' tails (short and long transit times are strongly emphasised compared with transit times close to the mean). Case α=1 is the exponential distribution (linear storage). Cases
α=3 and 10 are peaked and have smaller tails (short and long transit times are reduced compared to transit times close to the mean).

Distribution shapes described by the range of α values between 0.3 and 1.0 for the GM have been found useful for interpreting seasonal tracer cycles in streams (Kirchner, 2016a), but have generally not been as effective for interpreting tritium concentrations in stream baseflow or groundwater (see below). Of the simple LPMs, the EPM has no equivalent shapes to the
25 GM for α less than 1 and the DM has none for α less than 0.8 (Table 1). The standard deviation (sd) is used to quantify the goodness-of-fit between the GM and the best-fitting version of each of the other models, where

$$sd = \sqrt{\frac{\sum_1^N (GM_i - LPM_i)^2}{N}} \qquad (10)$$

The most basic of compound LPMs (the TPLR model) gives TTD shapes which are comparable with the GM shapes in this range of α (Fig. 1d).





The GM, EPM and TPLR models give the exponential model when they have the appropriate parameters ($\alpha = 1$ for GM, $f = 1$ for EPM, and both $\tau_f$ and $\tau_s$ equal to $\tau_m$ for TPLR).

TTD shapes for the GM with $\alpha$ between 1 and 10 are equivalent to EPM shapes with exponential fractions (f) between 1.0 and 0.4 (Table 1), which have been found suitable for interpreting tritium concentrations in baseflow and groundwater (e.g.

5      Maloszewski et al., 1983; Stewart et al., 2007; Morgenstern and Stewart, 2004). The useful range of the DM has dispersion parameters ($P_D$) between about 2.5 and 0.05 corresponding to the GM with $\alpha$ between 0.8 and 10 (Table 1). The GM and EPM shapes become less similar to each other as $\alpha$ increases to 10, while the GM and DM shapes become more similar. The TPLR does not mimic the GM in this range (the EM version ($\tau_f = \tau_s = \tau_m$) gives the best fits to the GM with $\alpha$ greater than 1), and probably gives more realistic shapes than the simple LPMs when there are two major water components present in streams.

## 3 Results

### 3.1 Aggregation effects on mean transit times determined using tritium

This section examines the effect of combining two different water components on the MTT of the mixture. The 'true' MTT is determined by mixing the two waters according to Eq. (9) with b being the fraction of the first component in the mixture ((1-b) is the fraction of the second component), and $\tau_1$ and $\tau_2$ are the MTTs of the two components. The 'apparent' MTT is

determined by fitting a simple LPM (the GM) to the tritium concentrations of the mixture ($C_m$) given by

$$C_m = b.C_1 + (1-b).C_2 \tag{11}$$

where $C_1$ and $C_2$ are the tritium concentrations in components 1 and 2 respectively. b is assumed to be 0.5 for simplicity in what follows, i.e equal fractions of both components. Following Kirchner (2016a), we did not consider evapotranspiration in our analysis of tritium aggregation effects.

The relationship between mean transit time and tritium concentration is illustrated in Fig. 3 for the assumption of constant input tritium concentration of 2 TU over time, i.e. without the bomb pulse during the nuclear age. This simplifying assumption is necessary to allow for the analysis shown in Fig. 3; with the real peaked input the figures would be scrambled. The assumption of a constant tritium input function is however becoming increasingly realistic in the Southern Hemisphere, with the bomb tritium from 50 years ago now fading away and assuming no more large-scale releases of tritium to the atmosphere.

This assumption is not limited to tritium but would also be valid for all radioactive tracers with constant input such as carbon-14 and argon-39.

Fig. 3a shows the relationship for the piston flow model. The red points indicate the assumed water components (with MTTs of 3 and 197 yr respectively) and the red dashed line is the mixing relationship between them (described by Eqs. 9 and 11). The 'true' MTT (100 yr) of a 50:50 mixture of the components is shown on the red dashed line. The black curve is the result

of applying the piston flow model (PFM) to the mixed tritium concentrations (Eq. 10). The PFM describes a flow system in which tritium decays radioactively with a half-life of 12.32 yr, but there is no mixing so that all of the water has the same age.





A 50:50 mixture of the components gives the 'apparent' MTT shown (15 yr), which is much less than the 'true' MTT. This results from the strongly non-linear character of the black curve (Fig. 3a) and therefore combining two dissimilar subcatchments causes aggregation bias in a similar way to that demonstrated for seasonal tracer cycles by Kirchner (2016a) in his Fig. 5 (and also for radioactive decay by Bethke and Johnson (2008) in their Fig. 3a).

Figs. 3b-e show the same calculations applied to the GM with different shape factors. The different shape factors describe different fractional contributions of past water inputs to the present water output as illustrated by the TTDs in Fig. 2a. Application of the GM takes account of input of new water (with tritium concentration of 2 TU) mixing with old water already present in the system with lower tritium concentrations because of radioactive decay. The heavy-tailed TTD ($\alpha = 0.3$, Fig. 3b) causes a flattening of the curves but there is still a considerable difference between the true and apparent MTTs (100 to 44 yr).

The exponential TTD ($\alpha = 1$) shows a larger difference (100 to 26 yr), and the more peaked TTDs ($\alpha = 3$ and 10) have greater differences still. The most sharply peaked TTD ($\alpha = 10$) approaches the result for the PFM.

Fig. 4 compares the true versus the apparent MTTs calculated using the real peaked tritium input function from Kaitoke (Fig.1). The calculations were structured as virtual experiments in which the two water components were initially assumed to have the same MTTs (i.e. $\tau_1 = \tau_2$) and therefore the mixture had the same true and apparent MTTs (Eqs. 9 and 11) and plotted on the

1:1 lines. The second component (MTT2) was then allowed to become older so that the difference in MTTs between the two components increased. This caused the apparent MTT to become younger than the true MTT and the points to move further and further away from the 1:1 line as shown by the curves in Fig. 4. As expected, the greatest age differences caused the biggest deviations from the 1:1 lines.

The different values of $\alpha$ cause differences to the patterns observed, but the patterns are similar overall. They are tighter around

the 1:1 line for $\alpha = 0.3$ showing smaller aggregation effects, and are most divergent for $\alpha = 10$. Errors of fitting for determining the apparent MTTs are greatest when component 1 is youngest, these are shown for 3 and 25 years. The errors are largest with $\alpha = 10$. The fitting errors are important because big errors would lead researchers to apply more complicated and therefore more realistic LPMs (such as binary LPMs), as many have in the past (e.g. Maloszewski et al., 1983; Uhlenbrook et al., 2002; Stewart and Thomas, 2008; Morgenstern et al., 2015).

Using the Trier (Northern Hemisphere) tritium input function (Fig. 1) results in very similar aggregation biases for tritium MTTs (Fig. 5) compared to those obtained with the Kaitoke input (Fig. 4). Using Northern or Southern Hemisphere tritium input functions makes almost no difference. Note that the problem of multiple age solutions often experienced using tritium with the Northern Hemispheric input function (e.g. Stewart et al., 2010) does not arise here because we calculate around 50 tritium values (one for each year) and this constrains the final 'apparent' fitting to a single unique solution. However, the fitting

errors for the apparent MTTs are larger than those determined with the Kaitoke input function.

## 3.2 Aggregation effects on young water fractions

The effect of combining two different water components on the young water fraction ($Y_f$) of a mixture is examined in this section. $Y_f$ is the fraction of water with ages between zero and a young water threshold ($t_y$), i.e.



$$Y_f = \int_0^{t_y} h(\tau).d\tau \tag{12}$$

The young water threshold for tritium has been estimated as the value that gives agreement within 10% of the apparent and true young water fractions for the case with the greatest difference in ages between the two water components (i.e. waters with MTTs of 3 and 397 years respectively in this study). This case gives the greatest difference between the apparent and true

MTTs. As the threshold increases, the apparent $Y_f$ increases relative to the true $Y_f$. The 'true' $Y_f$ is determined by mixing the two waters according to the equation

$$Y_{ftrue} = b.Y_{f1} + (1-b).Y_{f2} \tag{13}$$

in analogy with Eq. (9). b is the fraction of component 1 in the mixture, and $Y_{f1}$ are $Y_{f2}$ are the young water fractions of the two components. The 'apparent' $Y_f$ is determined from an LPM fitted to the tritium concentrations of the mixture (Eq. 11). b

is assumed to be 0.5. Tritium aggregation biases are much smaller for young water fractions than for MTTs (i.e. the points plot much closer to the 1:1 line) as observed by Kirchner (2016a) for seasonal tracer cycles. Fig. 6 gives the results calculated with the GM with different shape parameters

The young fractions are dependent on the choice of the young water thresholds used in Eq. (12) because water with transit times longer than the thresholds are omitted from the calculation. However, the result would hold with any reasonable and

consistent choice of the young water thresholds, because the normally underestimated water with old ages (i.e. the long tail portions of the TTDs) would be cut off in all cases. They would then not be causing the apparent young fractions to deviate from the true young fractions. On the other hand, the MTTs include the underestimated long tails and therefore give discrepant results.

The young water thresholds ($t_y$) used for these calculations are plotted against α in Fig 8. The method used for determination

of $t_y$ is described earlier in this paper. The points fit a power law given by

$$t_y = 11.6\alpha^{-0.274} \tag{14}$$

The reason for this relationship, which is similar to that found by Kirchner for seasonal tracer cycles, is to be found in the gamma distribution. It is important to note the differences in resulting thresholds between tritium and stable isotope tracers. For stable isotopes, Kirchner (2016a) reported a young water threshold range from 0.1 to 0.25 years (or approximately two

25  months) for the GM shape factor from 0.2 to 2, respectively. From our tritium evaluation, the young water threshold of tritium-based transit times ranged from 16 to 6 years with shape factors from 0.3 to 10. In addition, this relationship is also applicable to estimate the young water fractions of other LPMs with parameters corresponding to α in Table 1.

Young water fractions evaluated using tritium are of practical interest for various threshold ages, for example one year for assessing drinking water security of groundwater wells (water mixtures without any fraction of water of less than one year are

regarded as secure in terms of potential for pathogen contamination (Close et al., 2000; Ministry of Health, 2008)), or 60 years to assess the fraction of water that has already been impacted by high-intensity industrial agriculture starting after WWII.





### 3.3 Aggregation effects on MTTs for seasonal tracer cycles

Aggregation effects for seasonal tracer cycles have been determined by the methods of Kirchner (2016a) for comparison with the tritium effects. The rainfall input variation has been approximated as a sine wave with a one-year period to imitate the seasonal tracer cycle, and the sine wave has been traced through the convolution using the gamma distribution (GM). Fig. 9 shows the aggregation effects for the effective range of GM shapes for streams (with α between 0.3 and 1.0). These patterns are very similar to those observed using tritium concentrations (Fig. 4), so it is clear that the effects are effectively the same whether seasonal tracer cycles or radioactive isotopes are being used.

### 3.4 Case studies from the literature

The calculations above have shown that fitting simple LPMs to tritium data can potentially cause significant (or severe) underestimation of the MTT when two or more dissimilar tributaries feed a sampled outlet. On the other hand, the young water fraction is not likely to be affected by such errors (as also found for seasonal tracer cycles by Kirchner, 2016a). A number of studies using tritium to determine MTTs have been reported in the literature. How significant have such errors been in actual practice? This section describes some case studies from the literature to explore this question in different hydrogeologic settings of New Zealand. The case studies have been chosen to cover the age dating range of the tritium method in well, spring or stream flows.

### 3.4.1 Two water components at Waikoropupu Springs (karstic springs fed by Arthur Marble overlain by Tertiary sediments)

Tritium measurements at the Waikoropupu Springs began in 1966 and cover almost the rise and fall of the tritium bomb spike in precipitation (Stewart and Thomas, 2008). Fig. 10a shows the tritium concentrations of the recharge, the Main Spring, and the best-fitting model simulations of the data. The mixing models used were two simple LPMs (EPM and DM) and a compound LPM (the double DM or DDM).

The DDM was used because $\delta^{18}O$ and Cl measurements showed that there were two separate water systems contributing to the Main Spring (a shallow system and a deep system, see Stewart and Thomas (2008)). The residence time distributions of the models have similar shapes with peaks of very young water and long tails of much older water (Fig. 10b). The variation of the quality of the fits with MRT is shown in Fig. 9c, with the goodness-of-fit being expressed as the standard deviation (sd) of the simulations about the measurements.

All three models gave good fits to the data, and the mean residence times were sharply constrained close to 8 years (Table 2, Fig. 10c). The best-fitting EPM and DM models had MTTs of 7.9 and 8.2 yr respectively. The DDM model fitted very well indeed (sd 0.08 TU) with overall MRT of 7.9 yr from Eq. 9. This model had 26% water from the shallow system with MRT = 1.2 yr, and 74% from the deep system with MRT = 10.2 yr.



There is little aggregation error in the MTTs in this case (i.e. the 'apparent' MTTs of the EPM and DM models are very similar to the 'true' overall MTT of the DDM model). This is because both systems are young in relation to the young water threshold for tritium applying to this case ($t_y$ is 11 yr). The young fractions are also similar to each other at about 0.7.

### 3.4.2 Kuratau River (volcanic ash deposits and andesite)

Kuratau River flows into Lake Taupo in the North Island of New Zealand. Samples from the river were analysed for tritium from 1960 to the present making it the longest tritium time-series in New Zealand (Morgenstern and Taylor, 2009). The best EPM model (almost an EM, Table 3) had an MTT of 4 yr but fitted relatively poorly to samples collected around 1970 (Fig. 11), while a compound mixing model (double EPM) with overall MTT of 11 yr fitted much better. This comprised 65% of an EM with MTT of 1 year and 35% of an EPM with MTT of 30 years. The younger component is believed to be derived from drainage from the almost impermeable andesite in the catchment, while the older component comes from very porous volcanic ash deposits; distributed groundwater models calibrated with groundwater levels, river discharges and tritium concentrations substantiated these flows (Gusyev et al., 2013; 2014). Strong aggregation bias is shown by the marked difference in MTTs between the simple EPM model and the compound model (giving apparent and true MTTs of 4 and 11 yr respectively), due to the dominance of the young component. Note that the samples collected after 2000 are not capable of distinguishing between the models in the Southern Hemisphere, although this is still possible in the Northern Hemisphere. The young fractions of 0.93 and 0.65 were determined from the TTDs with threshold $t_y$ of 10 yr (Table 3).

### 3.4.3 Hangarua Spring and Hamurana Stream (volcanic ash deposits, Mamaku Ignimbrite)

Hangarua Spring drains from the Mamaku Plateau and flows into Lake Rotorua via the Hamurana Stream which also gains water from other springs. Tritium samples were collected from about 1970 to the present. The best fitting EPM for Hangarua Spring has MTT of 58 yr but fits poorly to the measured data, while the compound model (DEPM) with MTT of 90 yr fits well (Fig. 12a). It consists of 35% 16-year-old water and 65% 130-year-old water (f parameters listed in Table 3). A moderate aggregation bias is demonstrated by the difference in MTTs (Table 3). Similarly, Hamurana Stream shows a moderate aggregation bias (apparent to true MTTs of 70 and 124 yr) (Fig. 12b). The compound model consists of 35% 12-year-old water and 65% 185-year-old water. The young fractions are all about 0.15 based on $t_y$ of 10 yr (Table 3).

### 3.4.4 Reconciliation of tritium and carbon-14 results: Christchurch groundwater system (interleaved alluvial gravel and marine sediments)

Samples from a deep groundwater well in Christchurch, New Zealand, demonstrate possible effects of two water feeds with different mean transit times to the well (Stewart 2012). The well (M35/3637) taps the Wainoni Aquifer (Aq. 4), where it is unconfined in west Christchurch. The first tritium measurement was in 1986 and five subsequent measurements showed a



steady rise from near-zero tritium in 1986 as the bomb tritium peak passed through the site (Fig. 13a). The best-fitting EPM simulation to all points had f = 0.75 and MTT = 105 years. EPM fits to each individual point gave mean ages close to 105 yr. A double EPM (DEPM) model simulation which included an old water component with zero tritium concentration was also applied to the M35/3637 data, in order to investigate whether the mean age could really be older than the 105 years given by the EPM simulation (Stewart, 2012). Addition of the old water component did not improve the fit to the tritium data, but did not make it worse for addition of a small proportion (up to about 20%) of tritium-free water (Fig 13b). (The DEPM curve shown in Fig. 13a has two water components with mean ages of 100 and 1200 years respectively, with f=0.75 for each. The older water makes up 15% of the mixture.) So the mean water age could easily be older than the 105 years given by the EPM model (e.g. 265 years with the DEPM parameter values above), because of the aggregation error due to input of two water components with different ages.

Carbon-14 measurements collected at the same time as the tritium measurements (1986 to 2006) gave mean ages of 94, 283, 190 and 324 years according to the EPM model with f = 0.75 (Stewart, 2012). These show that the water feeding the well became older on average after 1986. The later tritium samples were not able to show this increase in mean age because the extra old water added had very little tritium and therefore was 'invisible' to the tritium method.

## 4 Discussion

### 4.1 Implications of tritium MTT aggregation bias

The analysis of Section 3.1 has shown that tritium-derived MTTs are just as susceptible to aggregation bias as seasonal tracer cycles when flows from dissimilar parts of catchments are combined using simple LPMs. Likewise, groundwater wells or springs fed by two or more water sources with different MTTs will show aggregation bias. However, there is an important difference between the biases of these methods and that is the time periods over which the biases are manifested. The bias applies for transit times greater than 2-3 months (i.e. the young water threshold) for seasonal tracer cycles (Kirchner, 2016a), whereas it applies for times greater than 6-16 years for tritium as demonstrated here, with the values of both depending on the GM shape factor. Note particularly that the bias *not only* applies to samples at the limits of the methods (i.e. with very small tracer cycles or near-zero tritium concentrations), *but also* applies to MTTs far below these limits.

### 4.2 How much have aggregation effects affected tritium MTTs in past studies?

Seasonal tracer cycles have been far more widely used to determine MTTs in streams than tritium concentrations. It is clear that many (if not most) studies using seasonal tracer cycles *interpreted with simple LPMs* will have been affected by aggregation bias because the values of the MTTs determined were between 2-3 months and 4-5 years. But, we contend that tritium studies will have been affected far less, despite aggregation bias also applying to tritium-derived MTTs, because:

(1) Tritium-derived MTTs can only have been affected by aggregation bias if they are greater than 6-16 years (or 6-12 years for α in the range 1 to 10) and were determined using simple LPMs. Stewart et al. (2010) surveyed most of the





tritium studies on streamflow up to then. They found that MTTs determined in baseflow averaged about $10 \pm 8$ years for headwater catchments (with two outlying volcanic ash flow catchments omitted) and $10 \pm 5$ years for large rivers. Baseflow averaged about 50% of the total flows from the catchments in both groups. In addition, Michel et al. (2015) concluded that "the range of residence times for active water in the majority of [surface] hydrologic systems throughout the world is on the order of one to two decades" based on a surface water database (GNIR) of over 6500 measurements of tritium assembled by the IAEA from measurements made between the late 1940s and the present. These suggest that aggregation bias is not important for many streamflow studies because of the transit times involved.

(2) Many of the tritium studies in the literature applied compound models calibrated by fitting to time series of tritium measurements rather than or as well as using simple LPMs. Provided the compound LPMs were well-chosen based on the characteristics of the catchments, they will produce more accurate TTDs than the simple LPMs and therefore will eliminate or reduce aggregation bias on MTTs.

A very good example is the study of Blavoux et al. (2013) describing the interpretation of an exceptionally long and very detailed record of tritium concentrations from the Evian-Cachat Spring in France. The tritium record was much too complicated to be fitted by a simple LPM. Instead, the detailed records of input and output allowed accurate specification of a combined model comprising of an EM ($\tau_m = 8$ yr) and DM ($\tau_m = 60$ yr) in series, with a small bypass flow in parallel with them, followed by a PFM ($\tau_m = 2.5$ yr) in series giving an overall $\tau_m$ of 70 yr. The combined model was closely related to the hydrogeology of the area and produced an accurate TTD for the average stationary state of the system, so there is little possibility of aggregation bias.

Four key examples of such studies were also described by Stewart et al. (2010) when comparing stable isotope and tritium estimations of MTTs. These studies nicely showed truncation of stable isotope TTDs compared to tritium TTDs. The studies were of Lainbach Valley streamflow in Germany (Maloszewski et al., 1983), Brugga Basin streamflow in Germany (Uhlenbrook et al., 2002), Waikoropupu Springs flow in New Zealand (the first case study described above) and Pukemanga streamflow in New Zealand (Stewart et al., 2007). Both simple and compound LPMs were applied in the original studies to interpret the MTTs in these streams. The compound LPMs were based on streamflow characteristics and gave better fits to the tritium data, but more importantly separated young and old water flows. The studies also illustrate the first point above, in that not much aggregation error is expected in the tritium MTTs for these streams because all of the identified water components are young relative to the young water threshold age (6-16 yr) for tritium.

## 4.3 Aggregation effects due to non-stationarity in systems

This study has not looked specifically at aggregation effects due to the non-stationary nature of hydrological systems. These will be different for catchments and groundwater systems, with streamflow variations often being far more dynamic than those in flows from wells and springs.

Methods of determining TTDs from tritium concentrations are quite different from those used for seasonal tracer cycles. The latter method in principle requires a series of samples from both input and output of a hydrological system in order to determine



the reduction in their variation during transport through the system. Using assumptions about mixing, this damping is then interpreted to give a TTD that is characteristic of a stationary system. But flows through hydrological systems such as catchments are never stationary because they are driven by intrinsically variable rainfall. Consequently, seasonal tracer cycle methods produce TTDs which are averages of the TTDs during the period of sampling. However, methods have now been

developed for stable isotope/chloride variations which allow determination of time-variant TTDs (Botter et al., 2010; Rinaldo et al., 2011; Hrachowitz et al., 2013).

On the other hand, use of tritium for determining TTDs depends on its radioactive decay rate which is applicable to single samples. Hence a series of tritium measurements can in principle yield a series of TTDs and the behaviour of a catchment during a range of hydrological conditions can be investigated. Interpretation of single tritium measurements to yield TTDs is

of course not necessarily straight-forward, because:

(1) The type of LPM and parameter value to apply to a single sample needs to be assumed (e.g. the shape parameter pre-determined). The specification of the LPM has often been determined by fitting LPMs to time series of tritium samples (as described in the case studies, Section 3.4), which obviously cannot be done with single samples. This work shows that it would be dangerous in terms of aggregation error to use a simple LPM for this. Hence the recommended

procedure would be to sample several tritium samples separated in time at each of a number of different streamflows and experiment with fitting different LPMs at each of the flows.

(2) Several values of the age parameter may allow the simulation to fit the measurement (i.e. there can be multiple solutions or ambiguous ages) because of input variations resulting from nuclear weapons testing in the past. This still applies for sites in the Northern Hemisphere, but is now largely past for the Southern Hemisphere (Stewart et al.,

2012; Stewart and Morgenstern, 2016; Gallart et al., 2016).

## 4.4    Considerations for future use of tritium MTTs

This work sounds a death knell for application of simple LPMs to hydrological systems (at least for estimation of MTTs) because of the risk of underestimation of MTTs due to aggregation bias, unless the simple LPMs are based on long series of tritium measurements. Seasonal tracer cycles should probably not be interpreted at all using simple LPMs. Fortunately, there

can be good reasons for choosing compound LPMs which are more realistic than simple LPMs. Hydrological reasons can be based on baseflow separation methods (Stewart, 2015; Duvert et al., 2016) or conceptual models of catchments (Hale et al., 2016). For example, Maloszewski et al. (1983) tested three LPMs of increasing complexity at Lainbach Valley with the most complex including a bypass flow representing direct runoff (30% of total flow) and shallow and deep reservoirs (52.5 and 17.5% of flow respectively) representing indirect runoff. Deuterium and tritium measurements were used to calibrate the

LPMs. Other reasons can be hydrogeological (two rock types in catchments, illustrated by the Kuratau River case study) or chemical (mixtures of water types, illustrated by the Waikoropupu Spring case study).



## 5 Summary and Conclusions

MTT estimations based on tritium concentrations show very similar aggregation effects to those for seasonal tracer variations. Kirchner (2016a) recently demonstrated that aggregation errors due to heterogeneity in catchments could cause severe underestimation of the mean transit times (MTTs) of water travelling through catchments when simple lumped parameter models (LPMs) were applied to interpret seasonal tracer cycles. Here we examine the effects of such errors on the MTTs and young water fractions estimated using tritium concentrations. We find that MTTs derived from tritium concentrations in streamflow are just as susceptible to aggregation bias as those from seasonal tracer cycles. Likewise, groundwater wells or springs fed by two or more water sources with different MTTs will also show aggregation bias. However, the transit times over which the biases are manifested are very different; for seasonal tracer cycles it is 2-3 months up to about 5 years, while for tritium concentrations it is 6-12 years up to about 200 years. We also find that young water fractions derived from tritium are almost immune to aggregation errors as were those derived from seasonal tracer cycles.

To investigate the implications of these findings for past and future use of tritium for estimating mean transit times in catchments and groundwater systems, we examined case studies from the literature in which simple and more complicated LPMs had been used. We find that MTT aggregation errors are small when either all of the component waters are young (less than 6-12 years, as found in many catchments), or they have similar MTTs to each other. On the other hand, aggregation errors are large when very young water components are mixed with very old components. In general, well-chosen compound LPMs should be used as they will eliminate or reduce potential aggregation errors due to the application of simple LPMs. Well-chosen means that the (compound) LPM is based on hydrologically and geologically validated information. The choice of a suitable LPM can be assisted by matching simulations to time series of tritium measurements (underlining the value of long series of tritium measurements), but such results should also be finally validated to ensure that the parameters won from modelling correspond to reality (since we nearly always have sufficient hydrological/geological data to examine the modelling results).

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





**Table 1: Comparison of the shapes of the gamma (GM), exponential piston flow (EPM), dispersion model (DM) and two parallel linear reservoirs (TPLR) transit time distributions. The shape parameters of the best-fitting versions of the other models and the goodnesses-of-fit (sd) between them and the GM are given. Blank cells indicate α values for which no fits with sd < 0.25 can be found.**

| GM α | 0.3 | 0.6 | 0.8 | 1.0 | 3.0 | 10.0 |
|---|---|---|---|---|---|---|
| EPM f | | | | 1.00 | 0.74 | 0.44 |
| sd | | | | 0.00 | 0.15 | 0.24 |
| DM $P_D$ | | | 2.25 | 1.36 | 0.22 | 0.05 |
| sd | | | 0.24 | 0.12 | 0.05 | 0.03 |
| TPLR $\tau_f/\tau_m$ | 0.003 | 0.005 | 0.009 | 1.00 | 1.00 | |
| TPLR b | 0.45 | 0.09 | 0.03 | 0 - 1 | 0 - 1 | |
| sd | 0.25 | 0.12 | 0.05 | 0.00 | 0.21 | |

**Table 2: Results of model simulations for the Main Spring of the Waikoropupu Springs. The young water fraction ($Y_f$) is the fraction of water with ages less than the young water threshold age.**

| Model (parameters) | Shallow fraction | Mean transit time | Young fraction | Standard deviation |
|---|---|---|---|---|
| | b | $\tau_m$ (yr) | $Y_f$ | sd (TU) |
| EPM (f=0.95) | -- | 7.9 | 0.75 | 0.40 |
| DM ($P_D$=1.8) | -- | 8.2 | 0.71 | 0.29 |
| DDM ($\tau_s$=1.2 yr, $P_{Ds}$=0.12) ($\tau_d$=10.2 yr, $P_{Ds}$=0.6) | 0.26 | 7.9 | 0.65 | 0.08 |

10  **Table 3: Parameters of simple and compound LPMs applied to tritium measurements from the Kuratau River, Hangarua Spring and Hamurana Stream.**

| Feature | Model | b | Overall MTT $\tau_m$ (yr) | $Y_f$ | Component 1 $\tau_1$ (yr) | $f_1$ | Component 2 $\tau_2$ (yr) | $f_2$ |
|---|---|---|---|---|---|---|---|---|
| Kuratau River | EPM | 1.00 | 4 | 0.93 | 4 | 0.99 | -- | -- |
| | DEPM | 0.65 | 11 | 0.65 | 1.0 | 1.00 | 30 | 0.50 |
| Hangaroa Spring | EPM | 1.00 | 58 | 0.10 | 58 | 0.91 | -- | -- |
| | DEPM | 0.35 | 90 | 0.17 | 16 | 0.63 | 130 | 0.87 |
| Hamurana Stream | EM | 1.00 | 70 | 0.14 | 70 | 1.00 | -- | -- |
| | DEPM | 0.35 | 124 | 0.21 | 12 | 0.77 | 185 | 0.82 |

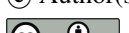



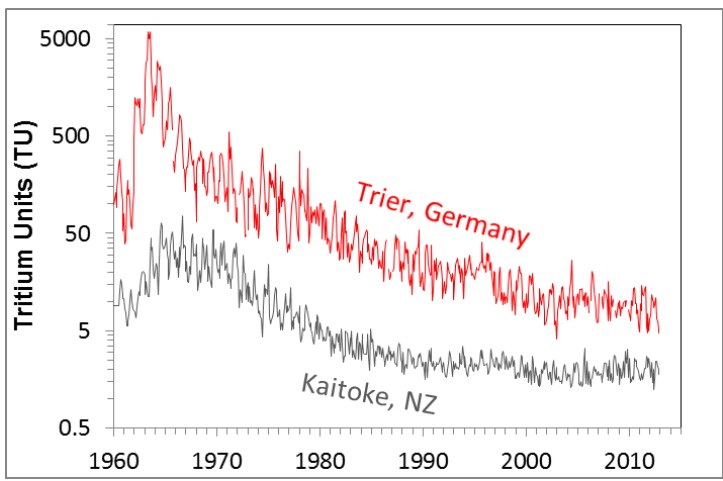

**Figure 1. Tritium concentrations in monthly precipitation samples at Kaitoke, New Zealand in the Southern Hemisphere, and Trier, Germany in the Northern Hemisphere.**

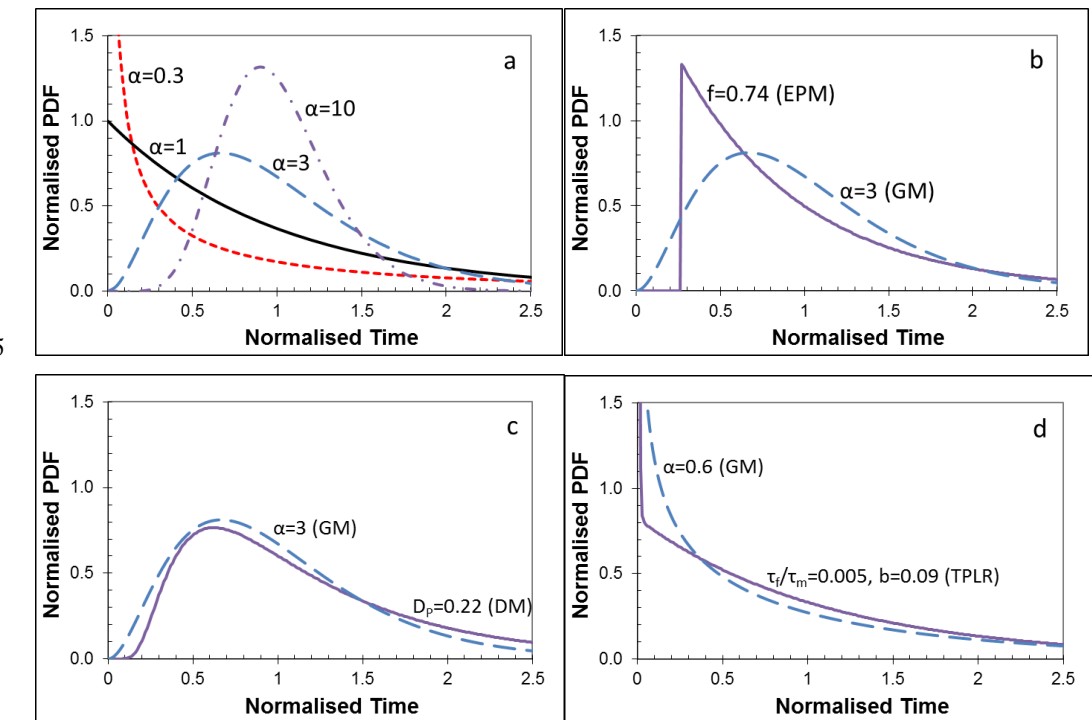

**Figure 2. (a) Gamma distributions (GM) for shape factors α between 0.3 and 10. The axes show normalised transit time ($\tau/\tau_m$) and normalised PDF ($h(\tau)$ x $\tau_m$). (b-d) Comparison between particular GMs and the best fitting EPM, DM and TPLR flow models.**




**Figure 3(a-e). Aggregation error when mean transit time (MTT) is inferred from tritium concentration in mixed runoff from two subcatchments with different tritium concentrations and MTTs (shown by red dots) using the piston flow and a range of gamma models. The tritium input concentration is assumed to be constant at 2 TU for clarity. The relationships between MTTs and tritium concentrations are strongly non-linear causing marked differences between the true and apparent MTTs.**





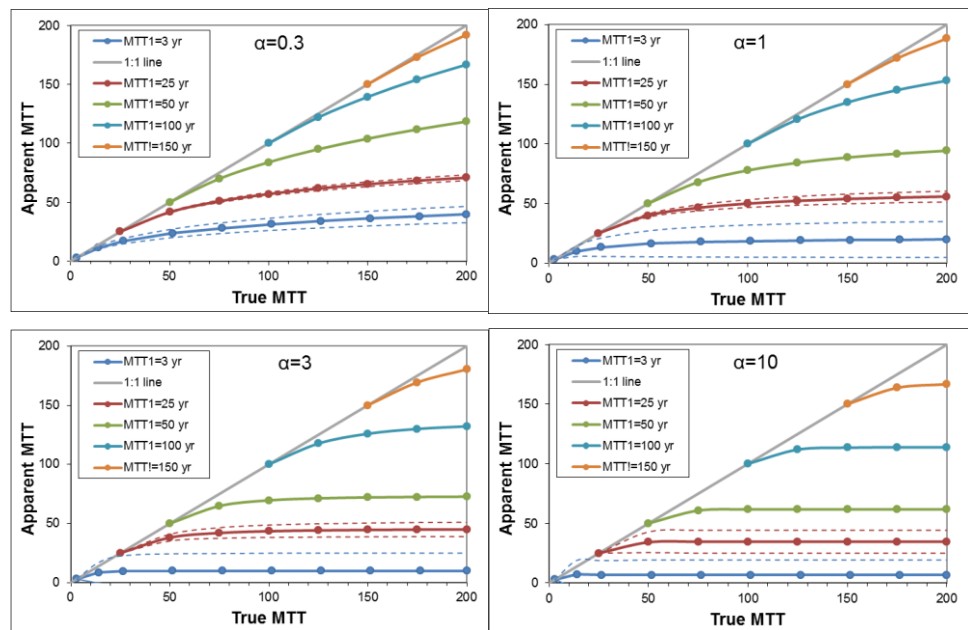

**Figure 4: Aggregation effects for tritium MTTs using the Kaitoke input function. Fitting errors in the apparent MTTs are shown for MTT1 = 3 and 25 yr by dashed lines.**

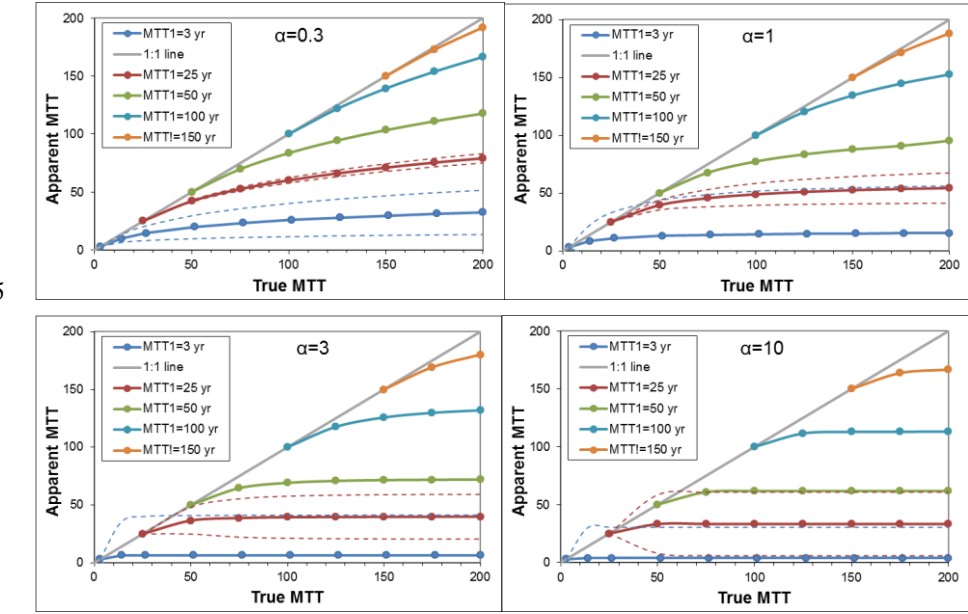

**Figure 5. Aggregation effects for tritium MTTs using the Trier input function. Fitting errors in the apparent MTTs are shown for MTT1 = 3 and 25 yr by dashed lighter lines.**





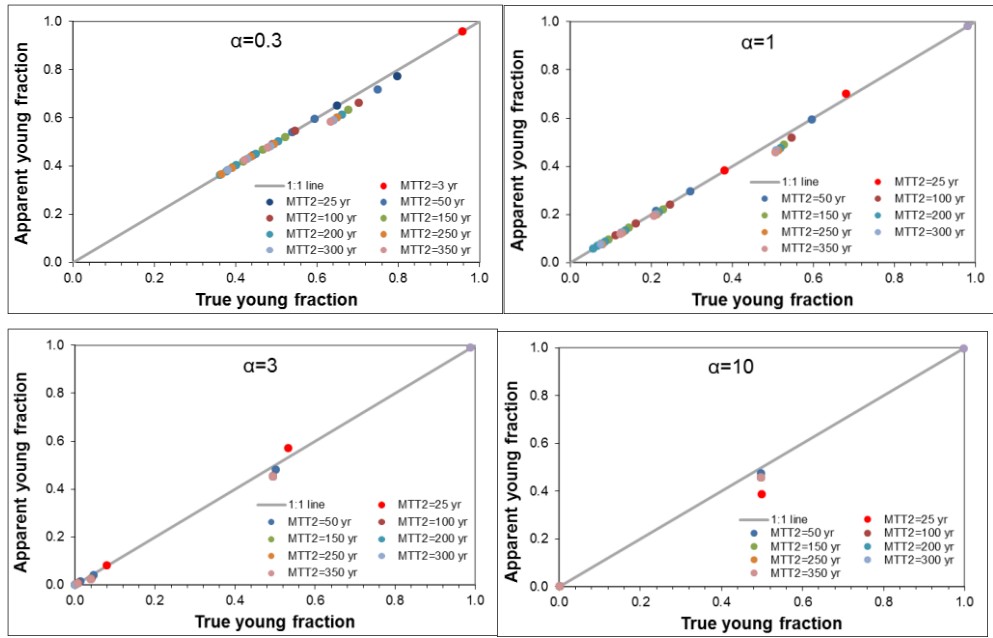

**Fig. 6. True versus apparent tritium young water fractions for different values of α with the GM for the Kaitoke input function.**

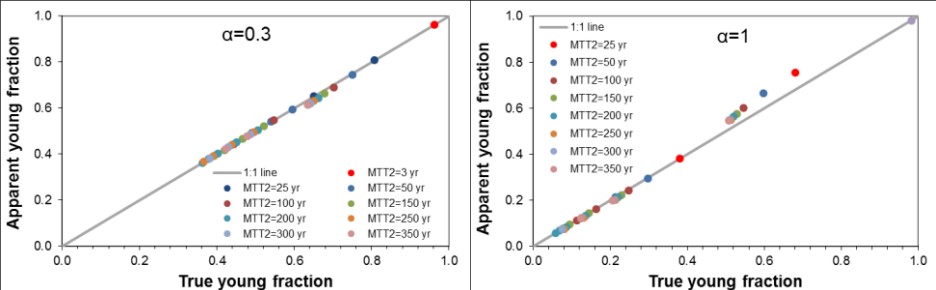

5  **Figure 7. Tritium young water fractions using the Trier, Germany tritium input function.**

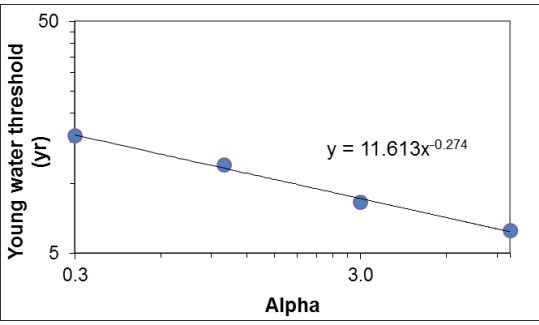

$y = 11.613x^{-0.274}$

**Figure 8. Plot of young water threshold versus α for tritium. Note the log scales.**





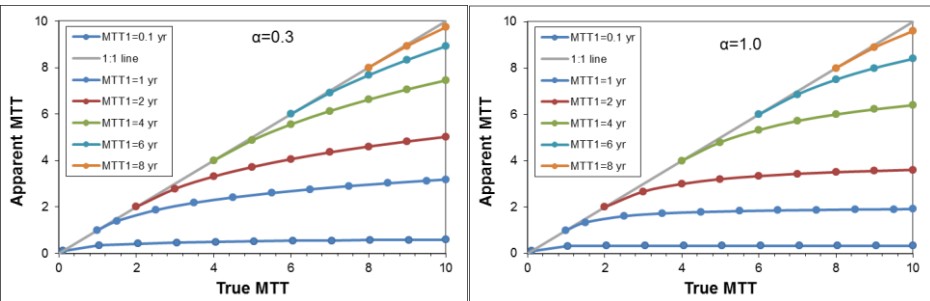

**Figure 9. Aggregation effects on MTTs determined using seasonal tracer cycles for the effective range of GM shapes (α = 0.3 – 1.0).**

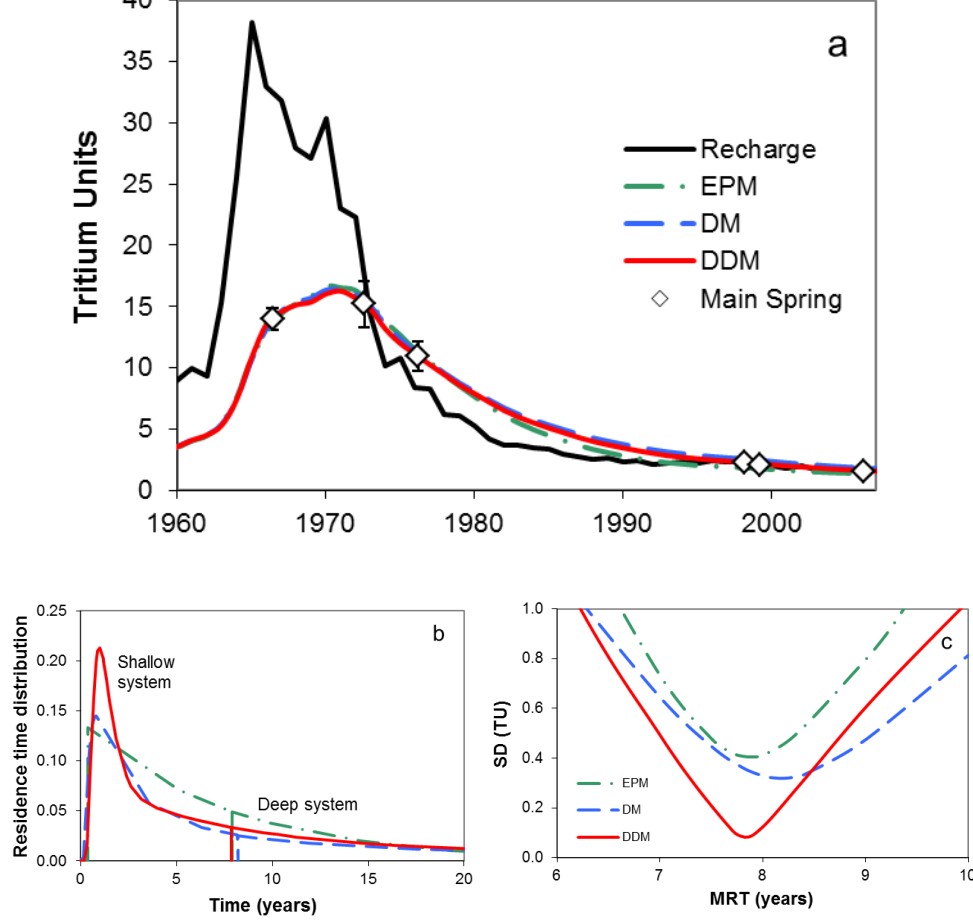

5     **Figure 10. (a) Tritium measurements of the Main Spring of the Waikoropupu Springs, New Zealand. (b) Residence time distributions of the best-fit simulations. (c) Variation of goodness-of-fit criterion with MRT.**





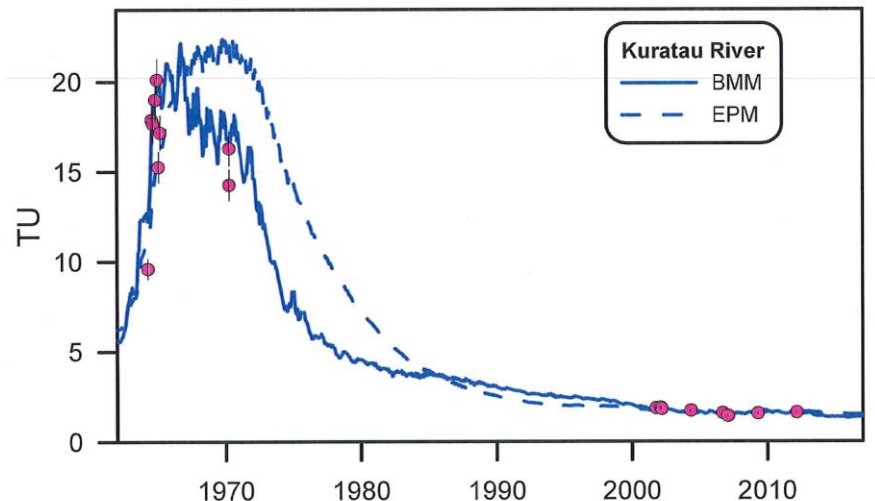

**Figure 11. Tritium measurements and model fits for Kuratau River, New Zealand.**

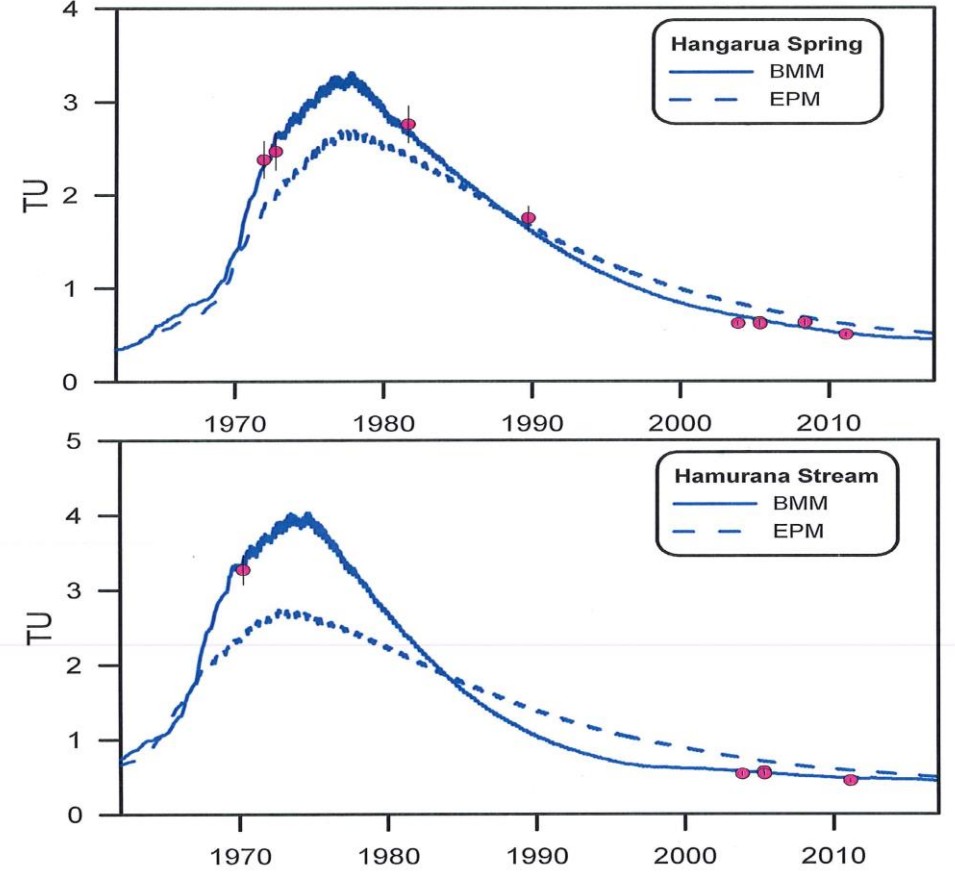

**Figure 12. Tritium measurements and model fits for (a) Hangarua Spring and (b) Hamurana Stream, Rotorua, New Zealand.**





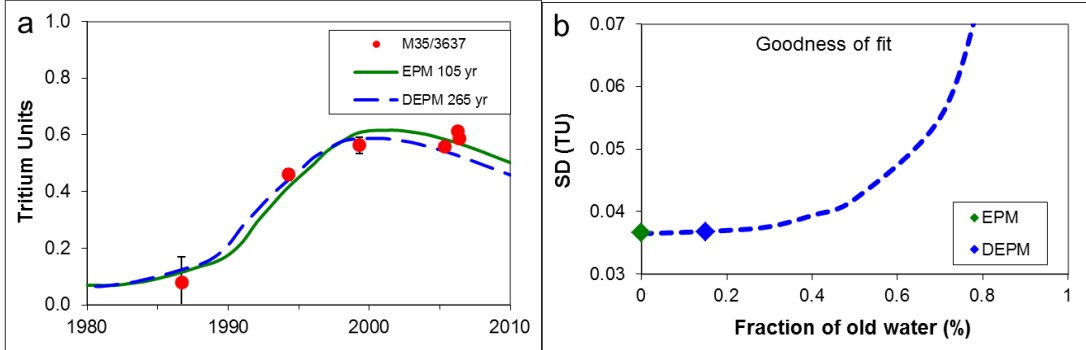

**Figure 13. (a) Tritium measurements and simulations for groundwater well M35/3637 in Christchurch, New Zealand. (b) Variation of the goodness-of-fit criterion (sd) with fraction of old water.**