# Peer review of "Aggregation effects on tritium-based mean transit times and young water fractions in spatially heterogeneous catchments and groundwater systems"

_Hydrology and Earth System Sciences, 2016_

## Short Comment (SC1) · 31 Dec 2016

Dear authors, dear editor,

The study by Stewart et al. is very thorough in that it tries to estimate the "aggregation error" systematically for different parameterisations of the gamma model. I feel however that it misses a central question (also ignored in Kirchner's paper where the method was first presented), which is whether the toy model adopted is appropriate at

all to study the effects of heterogeneities on the transit time distribution and hence on the estimates of the mean transit time. I think the question is not as trivial as it seems (and might actually be quite important for the future development of water dating). The relationships between mean transit time and tritium activity shown on Figure 3 clearly display approximately linear segments over which the mixture of water coming in equal volume from two different subcatchments would lead to a negligible underestimation of the true MTT (for instance, on figure 3d for MTTs between 0 and 20 years). Only by assuming heterogeneities so massive as to lead to MTT subcatchments differing by 200 years (!) does one observe equally enormous underestimations of the true MTT. This observation however begs the question: what degree of heterogeneity, and hence how large a difference in subcatchments' MTTs can usually be expected in real world catchments ? While Kirchner only mentions in passing a factor 2 as characteristic for "true heterogeneity" without further elaboration, Stewart et al. provide a much more detailed analysis in paragraph 4.2, where they conclude that "aggregation error" was probably small in most published studies because the estimated MTTs mostly lie between one and two decades (a window which Stewart et al. determined to be nearly "aggregation error" free). Their conclusion however rests on estimates of headwater catchments and surface waters where tritium was sampled, which does not really give any idea of MTTs' variations and range in smaller subcatchments and subcatchment's subcatchment (and subcatchments' subcatchment's subcatchment and so on...). Another way to look at this question is this. One could be tempted to answer that since we do not know how to quantify the degree of heterogeneity, it could be anything, and consequently assuming a large difference is conservative. I am concerned however that too much conservativeness leads to confusing or over cautious results, but additionally, there IS (at least) one study that addressed this question mechanistically for a number of case in an heterogeneous aquifer, namely that of Luther and Haitjema in Journal of Hydrology (1998). The authors show that in many cases ("stratified, unstratified, confined or unconfined [aquifers]"), the simple exponential distribution (i.e. a special case of the gamme function with the alpha term being equal to 1) is a good

approximation of the transit time distribution (TTD) of heterogeneous catchments as long as heterogeneity is "not significant and distinct" (which would obviously be the case if the resulting MTTs are respectively 1 year and 200 years as in Stewart et al.). The strength of Luther and Haitjema's approach lies in its clear definition of heterogeneity, as it can readily be related to measurable field variables (porosity, recharge rate and hydraulic conductivity). By contrast, the conceptual model used by Kirchner and adopted by Stewart et al. offers a simple and attractive way to study the effect of mixing subcatchments' contribution, but simultaneously, by forsaking flow equations, it excludes measurable physical quantities from the entire analysis. Of course, Luther and Haitjema's work only pertains to groundwater systems, and one can suppose that for catchments where the contribution from surface runoff and interflow is sufficiently large compared to baseflow, the mixture of water from these different reservoirs with possibly "significant and distinct" differences in MTTs (says a couple of months for the interflow and 50 years for baseflow) will lead to a transit time distribution that sufficiently deviates from a gamma function to affect the results of the inverse modelling (i.e. the estimation of the MTT). This difference should be made clearly so as to avoid all possible confusion:

-For groundwater systems, the results of Luther and Haitjema show that the "homogeneous assumption" holds in many real-world situations. Thus, Kirchner's conclusion that "MTT's estimated from seasonal tracer cycles are fundamentally unreliable" is too broad and must be corrected urgently.

-For catchments where streamflow is not only sustained by baseflow (or in other cases I cannot think of), one can expect a TTD significantly different from an exponential or general gamma model due to the contribution from reservoirs characterised by largely different MTTs. But in that later case, the use of compound LPMs can help to reduce deviation by conceptually catering for it (this is a VERY important point made by Stewart et al.).

To conclude, I think the work by Luther and Haitjema should be cited prominently since

it is a serious physically-based alternative to the toy model proposed by Kirchner, while the serious weakness of Kirchner's model (its inability to relate the degree of "heterogeneity" to measurable field variables) should be emphasized as well, especialy since it leads to conclusions that are too broad and conservative concerning the robustness of simple lumped parameter models used in heterogeneous catchments.

Additional comments

-Figure 1: As far as I know, the time series of tritium in precipitation only starts in 1978 for the station Trier. Unless I am mistaken, data prior to that year have been calculated from regression, probably using the station Vienna. If that's indeed the case, it should be stated.

-Some figures have obviously been made using excel. I know they are just figures, but shouldn't excel be banned from scientific publishing altogether ?

Best regards, Julien Farlin

---

## Referee Comment (RC1) · Anonymous Referee #1 · 2 Jan 2017

The paper builds on the publications by Kirchner (2016) analyzing the effects of aggregation of transit times distributions on estimating the MTT from 2 systems using tritium as a tracer. In addition, the paper also discusses the effects for a couple of examples from the literature. The paper is well written, however, it could be improved when better linking the examples with the analysis of the aggregation effects, considering from the beginning the use of compound LPMs and improving the readability and celerity of the figures. The list of comments and ideas below should help to improve the paper to be

published in HESS.

General comments:

1) The definition of heterogeneity needs to be clearer in this paper. Most LPMs assume a certain heterogeneity in the system, otherwise there would not be a transit time distribution – the specific effect considered in this paper regarding the aggregation effects are two distinctly different systems, each heterogeneous, but usually separate flow systems resulting in the investigated effects – this could be – as nicely shown in the examples, a shallow and deep GW system, a confined or unconfined system etc. So, it would help to make this clear, heterogeneity is everywhere, but you are only analyzing a specific set of heterogeneity with you analysis.

2) As you mention at some locations in the paper, a compound LPR usually addresses the possible effects of the aggregation error – at least this is how you define it in the example section. Hence, there is no aggregation effect if we would always use compound LPRs – you partly propose this in section 4, but in my opinion, this is not strong enough. If we apply a compound LPR and the two compounds are quite similar, we could also use a simple LPM (maybe simple is not a good phrase, I would prefer single – and there will be no aggregation effect. However, if the two compounds are different, a single LPR would result in a strong bias. Hence, we should propose to always use compound LPRs and then analyze the results in order to deal with the aggregation effect.

3) Why are you only using the standard deviation as your objective function. This considers only part of the fit – maybe you should consider the paper of Kling and Gupta or others to better select an appropriate function to be used, in particular when comparing the different models in Table 1.

4) The definition of the young water fraction is problematic. Why should the threshold (ty) change with changing alpha. So, we would always need to define this ty and then apply it differently to the different sets or models etc. If we use something like the young

water fraction, we should define it based on a fixed value that is related to the analysis and the question behind, However, for me the young water fraction is just another measure in order to avoid displaying the whole transit time distribution – this is what we need (as done nicely in Fig 10 – this should be repeated for the other examples as well) and not another factor in addition to the MTT. It would be helpful to discuss this and also to better clarify what the assumption of a non-constant ty would mean with respect to the whole analysis.

5) The whole section 3.3 is not necessary and should be removed including the figure. It is sufficient if you discuss without showing a similar analysis as already done in Kirchner (2016).

6) as already mention in some other points, you should make the examples better comparable in terms of defining the compounds LPRs, but also in respect to the figures and results shown. I think you should at least do the following: a) use the most common single LPRs (EPM, EM, GM, DM) and the best compound (or even better also a reference compound model) model and fit it to the observed TU showing the resulting fit (including a quantification of the fit) and the resulting TTD of each model. b) based on these figures, discuss the possible aggregation effects based on the models and the objective functions. c) define MTT, young water fraction for all model combination (in figure or table).

7) In the discussion, you should clearly differentiate what is necessary or problematic when using single LPRs or when using compound LPRs. For example, section 4.1 is only relevant when applying single LPRs.

8) the statement in section 4.3 that individual tritium measurement can/should be used to estimate a series of TTDs for one system is in my opinion not helpful. We should rather use the series of tritium measurements to apply the best possible compound LPRs that is necessary to describe the heterogeneity of the system as nicely presented in several of the examples. A single tritium measurement is not helpful at all, as there

is too much ambiguity. This could only be solved when combining the single tritium measurement with other tracers to identify the individual systems (multi components) and hence the complexity of the system studied. Hence, the whole issue of aggregation error will be solved to begin with.

Specific comments:

1) The title is very long and awkward – it would be helpful to focus on the implications and less on the different ideas of aggregation – please shorten considerable.

2) P2/L32 – the tritium method does not only depend on the radioactive decay, but still also on the bomb peak. . ..

3) P5: The list of LPMs should be more consistent and maybe best in a table listing all used LPMs and possible compound LPMs. In the moment, this section is incomplete, as there are additional compound LPMs in the result section not introduced in the method. You should also consider a systematic use of the compound LPMs – 2P_EM, 2S_EM, or EM_P-EM, EM_S_EM, PFM_P_DM etc, I think this would generally help to get a better understanding how the compound LPMs are set up, in particular the one introduced in the example section.

4) P7: why did you select 3 and 197 years for the analysis of the aggregation effect. It would be better the select a more realistic difference, which is commonly observed in the tritium analysis. In addition, you could also use TU/TU of Input as a relative measure in order to avoid defining the constant 2 TU for the analysis.

5) P8/L21: it is not clear to me how the "error of fitting" was determined – please define this in the method section and do not only show it for one selected combination but for all.

6) P8/L25-30. The discussed small difference between the northern and southern Hemisphere are very difficult to see, as the results are shown in separate figures – either combine the figure to see the effects or use only the figure from one location

and mention that the results are similar for the other site (maybe with some kind of quantification).

6) P9/L4: I assume you mean 197 years.

7) P9/L19: mention that alpha is from the Gamma model – not directly clear in this context.

8) P12/L3: The assumption that the old water component includes a zero tritium concentration is not clear for at all – is this possible at all?

9) The figure should be improved considerably – the layout is changing, the borders are annoying, the arrangement is not consistent, points cannot be seen (e.g. Fig 6 and 7), style is changing in the paper (Fig 1-9 and 10-end).

10) Please describe the different lines in Fig 3 in a legend.
* * *

---

## Short Comment (SC2) · 2 Jan 2017

Dear Referee, dear authors, dear editor,

I would like to react to a couple of points that have been raised by referee #1.

Firstly, I was very surprised to read in comment 1 that referee #1 is explaining the transit time distribution by the presence of heterogeneities. This is plain wrong for the exponential model, which describes the TTD of a homogeneous semi-confined aquifer

exactly, and where the distribution is due to the different path lengths in the catchment, and isnot accurate at all for the dispersion model and the gamma model either, both of which include dispersion and diffusion, but ALSO the distribution of pathlength. The distribution collapses to a piston flow ONLY if all flow lines can be assumed to be of equal length. In the rest of that first comment, referee #1 seems to confuse two related issues: complex systems and heterogeneous systems. In a complex system, different and conceptually clearly separate reservoirs sustain discharge at the system's outlet. Usually, hydrogeological understanding can lead to a choice of LPM combination which best simulates that system (say two exponential components for the quickflow and baseflow reservoirs), or the fit improves significantly by doing so. This assumes that each reservoirs is sufficiently homogeneous to use a given LMP shape, all of which have been developed for a homogeneous medium. Piotr Maloszewski and co-authors have over the years shown quite a few example of improving the fit to measured tritium activity by combining models and hydrogeological understanding. Kirchner's model is somewhat similar in that TTDs are added up to simulate a heterogeneous system. This is not the same however, first because one would expect a heterogeneous catchment to be made up of more than two or three subcatchments (and hence two or three TTDs) which flow into one another, and second because while this combination is a conceptual contrivance, the combination used in complex systems is a conscious decision of the experimenter made based on data. The last sentence of that comment is not clear to me. What does the referee mean by "a specific set" ? What other set of heterogeneities does he think about ? Maybe he is pointing to the shortcoming of the model I was writing about in my comment.

In comment 2, the confusion between complexity and heterogeneity I mention above appears again. I think it is useful to clarify what is happening to the TTD in heterogeneous catchments (this is what we have tried to show, Piotr Maloszewski and I, in our manuscript currently under review at HESS). If the MTTs of the subcatchments are very different (say 60 days and 5 years) and assuming exponential TTDs, then the total TTD will be more curved than an exponential. So fitting an exponential model to the
tracer output of that system will result in a parameterised TTD that underestimates the younger fraction and often overestimates the older fraction to compensate (I suppose this is what the referee means with "a single LPM would results in a strong bias"). But since the modeller in a real world case has no access to the true TTD (which is more curved than the one he fitted), he will never know, and will probably not notice unless the output time series is exceptionally long and the fit bad. Referee #1 seems to suggest that we can guard ourselves against this problem by always using combined LPMs. This is not really the case. First, it can well be that the true TTD is so curved that even a double LPM will deviate from it significantly. And second, this increases the number of free parameters, in many cases beyond what can be robustly fitted to the data available. So no, using "compound LPMs" does not mean we get rid of the "aggregation effect".

I also disagree with comment 6 and would support the approach adopted by Stewart et al.. I think hydrologists should stop fitting blindly all kinds of models to output time series and look at exotic measures of fit to decide what model is "best". Stewart et al.'s case by case analysis is done with system understanding in mind, and emphasizes the need for the modeller to ask himself whether a model improvement makes sense physically.

The referee may not know previous work by Mike Stewart and Uwe Morgensten showing that the "ambiguity" he alludes to in comment 8 disappears once the memory of the bomb peak has "faded" from a groundwater system. Of course, as long a the tail of the bomb peak is measurable, more information can be gained from it. In particular, it is sometimes possible to recognise the need for a compound LPM, which becomes impossible using a single post-peak value (this has also been recognised by Stewart and Morgenstern). All this is not a matter of opinion, but is firmly rooted on established results.

As for the specific comments, number 4 raises the question of what IS a realistic difference in MTTs. In my comment I contend that answering this question is very problem-

HESSD
atic as long as the problem is posed in terms of MTTs instead of hydraulic variables (whose range of variation can be measured in the field).

Best regards.

---

## Short Comment (SC3) · 3 Jan 2017

In his interactive comment on the manuscript by Mike Stewart and colleagues, Julien Farlin substantially misrepresents a recent paper of mine (Kirchner, 2016), as well as earlier work by Luther and Haitjema (1998).

Farlin writes that "Kirchner only mentions in passing a factor of 2 as characteristic for "true heterogeneity" without further elaboration." This misrepresents a passage in

[Figure]

Kirchner (2016) that says nothing about real-world patterns of heterogeneity, but instead describes Monte Carlo trials of hypothetical pairs of subcatchments, which were combined to determine whether their runoff yielded reliable estimates of mean transit time (MTT). Monte Carlo trials can, by chance, lead to cases where both subcatchments are nearly identical, and therefore the resulting combined catchment is not meaningfully heterogeneous. Therefore, as Kirchner (2016) says, "Pairs with MTTs that differed by a factor of two were excluded, so that the entire sample consisted of truly heterogeneous catchments" (that is, so that the sample excluded catchments that by chance were nearly homogeneous). This statement very clearly refers to hypothetical Monte Carlo trials (and about how variable these hypothetical catchments need to be, in order to be considered as heterogeneous rather than homogeneous in that context). Instead it has been misrepresented by Farlin as a making a claim about how much heterogeneity actually exists in the real world.

Farlin characterizes my analysis as a "toy model" and asks "whether the toy model adopted is appropriate at all to study the effects of heterogeneities on the transit time distribution and hence on the estimates of the mean transit time." This misrepresents the point of the analysis, which was not "to study the effects of heterogeneities ON the transit time distribution", but rather to study how spatial variations IN transit time distributions (whatever their cause) will affect the reliability of MTT estimates. Farlin, in short, complains that my analysis is not appropriate, but for a task that was never its goal in the first place: a classic straw-man argument.

These confusions could have been cleared up in minutes via phone or e-mail, so it is regrettable that Farlin did not contact me directly before launching his public attack.

Nonetheless, Farlin's comment indirectly raises two potentially important questions. First, how is heterogeneity in catchments' characteristics related to heterogeneity in their transit time distributions? Neither Kirchner (2016) nor the present manuscript by Stewart et al. address this question, but it is an interesting one. However, even if this relationship were known, it would simply substitute one factor that is often poorly

quantified (the spatial heterogeneity in catchments' characteristics) for another that is also poorly quantified (the spatial heterogeneity is their transit time distributions). The second question raised by Farlin's comment is, given how little we know about the patterns of heterogeneity in catchments' characteristics and/or their transit time distributions, how sanguine should we be about the risk of aggregation errors?

My answer is the following. We know that important catchment properties (hydraulic conductivity, depth to bedrock, soil characteristic curves, etc.) typically vary by large factors, in spatially correlated fashion, across all the scales at which they can be measured. Given this pervasive multiscale heterogeneity, the burden of proof should be on those who claim that it doesn't matter, or who want to use techniques that are prone to aggregation errors (such as estimating MTT from seasonal tracer cycles). Alternatively, we should develop – and use – methods that are much less vulnerable to aggregation errors (such as the young water fraction concept presented by Kirchner, 2016).

Farlin's answer appears to be different: "One could be tempted to answer that since we do not know how to quantify the degree of heterogeneity, it could be anything, and consequently assuming a large difference is conservative. I am concerned however that too much conservativeness leads to confusing or over cautious results, but additionally, there IS (at least) one study that addressed this question mechanistically for a number of case in an heterogeneous aquifer, namely that of Luther and Haitjema in Journal of Hydrology (1998). The authors show that in many cases ("stratified, unstratified, confined or unconfined [aquifers]"), the simple exponential distribution (i.e. a special case of the gamme [sic] function with the alpha term being equal to 1) is a good approximation of the transit time distribution (TTD) of heterogeneous catchments as long as heterogeneity is "not significant and distinct" [...] For groundwater systems, the results of Luther and Haitjema show that the "homogeneous assumption" holds in many real-world situations. Thus, Kirchner's conclusion that "MTT's estimated from seasonal tracer cycles are fundamentally unreliable" is too broad and must be corrected urgently."

The crucial misrepresentation here is that Luther and Haitjema never say that their results are relevant to "many real-world situations", perhaps because, in fact, they aren't. For example, in Luther and Haitjema's horizontally stratified simulations, the simulated transit time distributions deviated significantly from the exponential distribution (what Farlin calls the "homogeneous assumption"), even though the maximum conductivity difference was always less than a factor of 10. In the one horizontally stratified case shown in the paper (Case H, Fig. 8), the largest conductivity contrast between the layers was less than a factor of 3. A factor of 3, or even 10, is vastly less than the 10,000-fold variation in conductivity that one finds just among different types of sand, or the 1,000,000-fold variation in conductivity among glacial tills, or the 5 to 8 orders-of-magnitude variability in hydraulic conductivity that one finds even within individual lithologic groups, or the roughly 10-14 orders of magnitude that separate igneous rocks from gravel (Gleeson et al., 2011).

As another example, when Luther and Haitjema simulated random variations in aquifer properties, they varied conductivity, recharge, and porosity by small factors (whereas in the real world, conductivity alone can vary by orders of magnitude), and they then assigned these random aquifer properties individually to the roughly 73,000 cells within the model watershed, with zero spatial correlation. Thus it is unsurprising that the resulting simulations conformed to the "homogeneous assumption", for the simple reason that at every scale larger than a few grid cells, the model watershed was extremely homogeneous indeed. As a representation of real-world heterogeneity, the variation in aquifer properties was too small to be realistic; furthermore, the lack of spatial correlation is inconsistent with every set of field data that I have ever seen.

In characterizing Luther and Haitjema's analysis as "serious" and "physically based", Farlin has apparently overlooked its obviously nonphysical assumptions. To note just one example, Luther and Haitjema allow their "confined" aquifers to receive spatially uniform recharge. Thus the "confining" layer must somehow allow recharge to pass through vertically, while simultaneously confining the aquifer by preventing vertical flow.

How, exactly, is this supposed to work?

Farlin has also apparently overlooked the abundant evidence (e.g., Kirchner et al., 2000, Godsey et al., 2010, Kirchner and Neal, 2013, Aubert et al., 2014) showing that tracer fluctuations in a wide variety of real-world catchments have spectral signatures that are inconsistent with exponential transit time distributions. This empirical evidence would seem to refute the notion that Luther and Haitjema's results hold "in many real-world situations".

Thus even if Luther and Haitjema's results are correct for idealized (and/or nonphysical) theoretical cases, with heterogeneity that is "not significant or distinct", they are inapplicable to much of the real world, in which heterogeneity is both significant and distinct (that is, correlated up to the hillslope or catchment scale).

In summary, Farlin's strident claim that "... Kirchner's conclusion that "MTT's estimated from seasonal tracer cycles are fundamentally unreliable" is too broad and must be corrected urgently" is not supported by any scientific evidence, either from his letter or from the work that he cites.

Aubert A, Kirchner JW, Gascuel-Odoux C, Faucheux M, Gruau G, and Mérot M, Fractal water quality fluctuations spanning the periodic table in an intensively farmed watershed, Environmental Science and Technology, 48, 930-937, doi: 10.1021/es403723r, 2014.

Gleeson T, Smith L, Moosdorf N, Hartmann J, Dürr HH, Manning AH, van Beek LPH, and Jellinek AM, Mapping permeability over the surface of the Earth, Geophysical Research Letters, 38, L02401, doi:10.1029/2010GL045565, 2011.

Godsey SE, Aas W, Clair T, Dennis I, de Wit H, Fernandez I, Kahl S, Malcolm I, Neal C, Neal M, Nelson S, Norton S, Palucis S, Tetzlaff D, Skjelkvaale BL, Soulsby C, and Kirchner JW, Generality of fractal 1/f scaling in catchment tracer time series, and its implications for catchment travel time distributions, Hydrological Processes, 24, 1660–

1671, 2010.

Kirchner JW, Aggregation in environmental systems – Part 1: Seasonal tracer cycles quantify young water fractions, but not mean transit times, in spatially heterogeneous catchments, Hydrology and Earth System Sciences, 20, 279-297, doi:10.5194/hess-20-279-2016, 2016.

Kirchner JW and Neal C, Universal fractal scaling in stream chemistry and its implications for solute transport and water quality trend detection, Proceedings of the National Academy of Sciences, 110 (30), 12213-12218, doi: 10.1073/pnas.1304328110, 2013.

Kirchner JW, Feng XH, and Neal C, Fractal stream chemistry and its implications for contaminant transport in catchments, Nature, 403, 524-527, 2000.

Luther KH and Haitjema HM, Numerical experiments on the residence time distributions of heterogeneous groundwatersheds, Journal of Hydrology 207, 1-17, 1998).
* * *

---

## Short Comment (SC4) · 9 Jan 2017

Dear colleagues,

I think most of what Jim Kirchner reproaches to my strident claim is due to our different perspectives. To a hydrogeologist like me, "many real-world situations" means "aquifers with exploitable yields" while a hydrologist such as Jim Kirchner may think of flashy catchments or maybe even of a hill slope.

This explains for instance why I prefer the exponential to the gamma model for certain applications, and why I take Luther and Haitjema's work very seriously.

There is indeed abundant evidence that the exponential TTD does not fit the spectral signature of conservative tracers in many catchments, while the gamma distribution does. As far as I know, however, all studies showing this were catchment studies. I have never yet read an article where the spectral signature of the tracer signal of a permeable aquifer (and not of a whole catchment) has been studied. Since a number of papers (not just Luther and Haitjema, but also Haitjema 1995, or Etcheverry, 2001) come to the conclusion that the exponential model describes exactly the transit time distribution of piecewise homogeneous aquifers, one cannot dismiss it as lacking scientific evidence.

Jim Kirchner's objection that confined aquifers should not allow vertical recharge, and that such an assumption clearly demonstrates the lack of realism of Luther and Haitjema's paper, can be answered very simply. In fact, Luther and Haitjema modelled a semi-confined (also referred to as leaky) aquifer, which is an aquifer bounded at the top and the bottom by aquitards, aquitards being geological units permeable enough to transmit water vertically in significant quantities over large areas and long periods. Such semi-confined formations are quite common all around the world and known to all hydrogeologists . As an example, here's a citation from South Africa's Water and Sanitation Department: "Most aquifers in South Africa are semi-confined or semi-unconfined in character" (https://www.dwa.gov.za/Groundwater/Groundwater_Dictionary/index.html?introduction_semi_confined_aquifer.htm). Furthermore, for permeable aquifers that are not topography controlled (and they are far from being a rarity, including in published groundwater dating studies. But they are not the type of catchments hydrologists are studying), the range of variation in transmissivity or porosity adopted by Luther and Haitjema is quite typical, and the lack of spatial correlation for transmissivity or porosity not uncommon either at the scale of pumping tests (see for instance Krasny, 1993).

These technical issues aside, I wish to stress that my comment was not an attack, but a criticism, which together with the possibility offered to anyone to respond to it and disagree or modify it, seems to me the very stuff of scientific peer review. A discussion over the phone instead of this public debate might have been just as profitable to Jim Kirchner and me, but would have excluded other potentially interested colleagues (probably very few) as well as the authors of the manuscript under review.

I find Jim Kirchner's efforts to assess the effect of heterogeneities on the calibration of lumped parameter models extremely interesting and useful, and I was not in the least criticizing him for asking that question, nor was I suggesting we should not care about it and smugly assume homogeneity everywhere. What I was pointing out is this: since the MTTs as the toy model's parameters ("toy model" was never meant to be derogatory, by the way. It is a very standard term in physics) cannot be constrained by experience or observation, one is forced to adopt a conservative stance and use a large range. This is fine, but it is nonetheless a problematic flaw, because unless one is careful in interpreting the results, it leads to a radical rejection of methods that are most probably fine in particular but important cases. Such a particular case for tritium-based estimations has now been explored by Stewart et al.. So we are making progress thanks to that toy model, and progress also means being aware of the limitations of whatever tool one uses, and encouraging others to test it further. To put it concisely, one should be careful neither to claim that heterogeneities do not matter, nor come to the conclusion that they have has such weight that lumped parameter models are fundamentally unreliable. Empirical evidence may one day allow such a radical rejection, but I do not think that Jim Kirchner's conceptual model actually does that, given that the range of MTTs assumed for the subcatchments is for now completely unconstrained. So here is my question again: "what degree of heterogeneity, and hence how large a difference in subcatchments' MTTs can usually be expected in real world catchments ?" As long as this question cannot be answered , even approximately, the toy model will be an interesting thought provoking tool, but hardly one that can lead to radical conclusions concerning the reliability of lumped parameter models. And I find Stewart

et al.'s answer, as praiseworthy an attempt as it is, insufficient in that regard.

Finally, I do not see how the question as "whether the toy model adopted is appropriate at all to study the effect of heterogeneities on the transit time distribution and hence on the estimates of the mean transit time" is a "misrepresentation" of Jim Kirchner's analysis. We simply do not agree, which is altogether different. It is perfectly clear that only the estimation of the catchment's MTT was of interest in his article. I think this is a methodological mistake and that one should have a good look at the different shapes of the synthetic TTDs and compare them to the best fit from lumped parameter models calibrated on the catchment's output signal. This is why is agree with Stewart et al. when they suggest the use of compound lumped parameter models, since this shifts the emphasis back to the shape of the TTD and away from the MTT as characteristic measure of a catchment's mean transit time.

Best regards,

Julien Farlin

Etcheverry, D., Une approche déterministe des distributions des temps de transit de l'eau souterraine par la théorie des réservoirs, PhD Thesis, University of Neuchâtel, 2001, 118 pages

Haitjema, H.M., On the residence time distribution in idealized groundwatersheds, Journal of Hydrology 172, 1995, 127-146

Krasny, J., ClassificatioN of transmissivity magnitude and variation, Groundwater 31 (2), 1993, 230-236
* * *

---

## Referee Comment (RC2) · Anonymous Referee #2 · 11 Jan 2017

In their manuscript, Stewart et al. investigated tritium-based estimates of mean transit times (MTT) and the fraction of young water (Yf) in light of aggregation bias due to catchment heterogeneities. Furthermore, past studies are reinvestigated and evaluated in respect to aggregation bias. This topic is highly interesting, as most commonly the stable isotopes of water (Oxygen-18, Deuterium) are applied in tracer studies. In comparison to this, tritium is used more seldom, but it has the potential to elucidate longer transit times, where stable isotopes hit a boundary at about 4-5 years. I hope my

comments and suggestions will be helpful to the authors and improve the manuscript.

General Comments

1) Manuscript structure: I found the structuring of the text to be all over the place, making it hard to read for me, as I was expecting to have all the tools necessary to understand the paper after reading the Methods. However, the "Results" section basically starts with several paragraphs of new Methods. I would suggest to either changing the order of the text to properly divide Methods, Results and Discussion, or rename the header titles from "Results" and so on to something else to avoid confusion. Please see specific comments about my ideas which paragraph could be shifted to different sections.

2) Methods: After introducing tritium (H3)-based TTD estimation, LPMs and their properties, "Results" starts and I am left with an unsure feeling of how the paper addresses the issues raised in the Introduction. I know you use the four GMs from Fig 2a, but in which combinations for the two virtual catchments? Only selected combinations, or all possible ones? How were the catchments mixed? (I know it is 50:50 because it says so later on, in Results…which links back to my comment about structure of the paper). Did you use the GM of each sub-catchment in Equation 1 and forward-propagated Northern and Southern hemisphere H3-data, then mixed it 50:50? All this information is missing, and young water fraction calculation or the literature reevaluation is not even mentioned here. Furthermore, I think that the description of the individual LPM can be shortened without losing important information. Also, I think Table 1 and showing that the GM can mimic the shapes of other LPM is not essential for understanding of the paper. It is interesting to quickly summarize which LPM is useful for which application, however.

3) Yf calculation: It is unclear to me how you calculated this. Yf is determined and calculated from the threshold age $t_y$ (Equ. 12), yet it seems the threshold age was calculated by comparison to apparent and true Yf already existing (page 9, first few

lines)? True Yf comes from the individual Yf1 and Yf2 (Equ.13), but where are they and their ty1 and ty2 coming from? Also, ty should give good agreement with 10% of apparent and true Yf. 10% of what? What is the 100%? Maybe an explanatory figure would help here, and this should be also in Methods.

4) Apparent Yf determined from LPM fitted to H3 of mixture: To my understanding, Kirchner 2016 showed that Yf of the mixture can retrieve the "true" Yf (calculated from our knowledge of the virtual system) using a gamma function, but only Yf is valid and the corresponding gamma function itself is not valid (otherwise we would have a valid gamma function and thus a correct MTT again, i.e., no aggregation bias). Equipped with this knowledge, how can we reliably trust the apparent Yf result if it comes from a LPM function that is fitted to the H3 mixture and will most likely not be e.g. gamma distributed anymore, but hyper-gamma distributed?

5) Chapter 3.3 seems unnecessary to me, and is very short in itself already. If you want to keep it, please elaborate on its importance.

6) Chapter 3.4: I am unclear as to which results were already obtained by the cited studies and which results were calculated by the present manuscript.

7) I generally doubt the validity of Chapter 3.4, the literature review. To me there is suddenly a huge leap in logic/faith: that using compound LPM will give the true MTT. Or one that is "truer" than the versions of single LPM. It is assumed that just a good fit of tritium tracer data warrants to say that the model gives true results. I do not say they are wrong, I do not say they are true. I do say we cannot know, or I do not see any evidence here that would substantiate your assumption that the compound LPM would give the true MTT. Even if both parts that feed the mixed water in all described studies would be homogeneous in themselves: the virtual experiment catchments of Kirchner 2016 were also homogeneous in themselves, but different from each other, and still led to aggregation bias. We would need proof that the individual catchments are "homogeneous enough" (whatever that means) and that compound LPM, which are

just simplifications of processes that we think occur in a catchment domain, correctly mix the two flows in a way that surely avoids aggregation bias. Just a good fit of observed tritium data is surely not a bad start, but not enough in my opinion. I am in favor or a) deleting Chapter 3.4 OR b) rewriting it much more cautiously, with discussing the considerations uttered here.

Specific Comments (page-line)

2-24: "young water" appears here the first time. Maybe define it a bit more clearly. How young does it need to be to be considered young water?

3-11: "[...] the one tracer". This makes it seem to me that two different tracers are used, but I rather get from this paragraph, that actually "when we only have tracer data of the mixture" is meant. Please clarify.

3-16: Choice of LPM based on hydrogeological situation: please give an example or reference at this point.

3-18: "water-bearing layers" to avoid confusion while reading (had to read three times)

4-6 "times. i.e. The water [...]" please correct the capital T and also the period after times seems strange.

4-15: with "calender time" you refer to daily time steps? Monthly? Yearly?

5-23: Starting the sentence with a side-sentence in brackets "(Maloszweski [...]" looks weird, in my opinion.

7-6: I would write 2.25 instead of 2.5, if you already use two digits after the comma for 0.05. Except for that I recommend not showing this information anymore, see General comment #2.

7-12 to 7-25: Methods

7-20: Please explain Fig. 3 a bit more in the manuscript to assist in fully understanding

it. As far as I understand it, the black curves show TU that one would measure in streamflow. It seems to be the fitting result of the LPM to the mixed H3 signal (p7-L29f). How did you find the two catchments TU concentrations (necessary to find the mixed TU signal) based on the desired MTTs of 3 and 197 years? There must be some other MTT-TU function behind it, that is not shown? I'm basing my last assumption on Kirchner 2016, where the combination of e.g. two exponential distributions did not lead to another exponential distribution, but a hyper-exponential distribution. Thinking along these lines, this confuses me even more now: every red dot in Fig 3, that is, every TU-MTT combination of the two individual catchments, lies exactly on the black curves that come from fitting the mixed runoff TU signal. But according to the logic here, the black curves should be wrong. How can the red dots lie on the black curve, if the shape of the distribution of the mixed runoff is not known and should be some hyper-something version?

7-24: With the assumption of a constant H3 input, are you not basically assuming that no groundwater much older than 50 years significantly contributes to runoff (that is, no groundwater which could possibly include the bomb peak). How realistic is this?

7-28: I would get rid of the reference to Equ. 9 here

7-30: Equ. 10 is the standard deviation and seems to be not fitting the text here. Do you mean Equ. 1?

7-31: All the water in streamflow has the same age, not in soil/aquifer. I would specify that here.

8-12: Regarding Fig 4: Earlier it was mentioned that real input TU data would cause scrambled results in Fig 3. But you use real data now. Please clarify why we can suddenly use them, or if the scrambling would just have made analyzing the figure more difficult, but not prevent the data from being used. Also, the paragraph explains Methods (8-12 to 8-18). Additional information is needed: what were the MTT2 increments? Was the GM model used, as you talk about alpha parameter later? How was MTT2

changed then, by changing beta parameter in certain increments?

8-20: How were the uncertainties for fitting young waters of MTT1 calculated? Might also be good to call it MTT1 here one time since it is only used in Fig. 4 and leads to some confusion initially when looking at Fig. 4.

8-22: The fitting errors are important because of more complex LPM in a good or bad way?

9-4: I guess it is 197 instead of 397.

9-14: Please define a "reasonable" choice of young water threshold.

9-16f: I disagree that cutting-off of the long tail after ty and thus leaving only the short tail will ensure that the apparent Yf does not deviate from the true Yf. As the TTD sums to "1", the long tail influences the short parts of the TTD and vice versa. If one is changed, the other changes too. If we know that the long tail is wrong, we can't be certain that the short TTD part is correct if we basically just ignore the existence of the long tail by cutting it off. To use a metaphor, this is to me like healing a bleeding wound by just not looking at it. And if REALLY the part of the TTD model before ty is correct, how can it be that the part after it is NOT correct? The equation and the parameters to calculate the complete TTD do not suddenly change...we just cut off a certain section of it. If the part before the threshold is true, the part after it is true. If the part after it is wrong, the part before it is wrong. Maybe the question is: how wrong? Significantly? Probably not, looking at results from Kirchner 2016.

9-22: No explanation follows why the reason for this relationship is found in the gamma distribution.

9-26: 6 to 16 years

9-28 to 9-31: Discussion

10-25: MRT, which I assume to be Mean Residence Time, is introduced for the first

time and replaces the MTT without explanation. Please rectify. Also, it should be Fig 10c.

12-21f: Following the reasoning about the different bias thresholds: Does this not mean that using tritium methods for streamflow, we would get bias-free estimates for transit times smaller than 6 years, which must include the seasonal cycle results if we would apply it to the stream, and ultimately agree with them?

Title of 4.2: Consider changing to "How much has aggregation affected tritium MTTs in past studies?"

12-30: Conclusion #1: I disagree with ONLY affected by bias if "older than 6 years" and "if determined by simple LPM", for reasons already explained above.

13-1f: If we take the variance into account in the given examples of 10 plusminus 8 and 10 plusminus 5, there seem to be quite a few catchments that have less than 6 years MTT.

13-8 Conclusion #2: As mentioned above, I see no evidence for this statement.

14-23: I must have missed the part in the manuscript that shows that simple LPM still work in case of long series of tritium measurements? Where is that shown?

Table 1: In the description change "The shape parameter of the best-fitting versions of the other models [. . .]", since it is not always the shape parameter for the other models, e.g. it is the dispersion parameter for DM.

Figure 2a: the scale parameter beta was fixed for each GM? Which value did it have?

Figure 4: In the legend: the orange MTT1 actually says "MTT!"

Figure 7 is not mentioned in the manuscript.

Generally the figures should be unified more in layout, e.g., get rid of the outer border.

---

## Referee Comment (RC3) · Anonymous Referee #3 · 12 Jan 2017

General comments

The results summarized in the first paragraph of the abstract and Figures 1-9 of the manuscript itself are, to the best of my knowledge, broadly correct.

However, the results reported in the second paragraph (and the related discussion and analyses in section 3.4 and 4) need to be reconsidered.

1. The statement that "well-chosen compound lumped parameter models should be

used as they will eliminate potential aggregation errors due to the application of simple lumped parameter models" directly implies that aggregation errors only arise with simple LPM's, but not with compound LPM's, or at least not with "well-chosen" ones (whatever that means).

1.a. This is not consistent with the analysis presented elsewhere in the paper. Figures 3-5 show clear aggregation errors from the use of simple lumped parameter models, but they would also show clear aggregation errors if more complex lumped parameter models were used. For example, a gamma model with alpha=0.3 closely approximates a compound LPM, but it is clearly vulnerable to aggregation errors, as shown in Figures 4 and 5.

1.b. Since the analyses in Figures 3-5 have been used to demonstrate aggregation errors in simple LPM's, exactly the same analyses must be applied to compound LPM's to demonstrate that these aggregation errors disappear. Until this is done, the claims in the abstract have not been demonstrated, and must be removed.

2. In some of the examples that are presented, the compound LPM's clearly fit the data better, but of course they should, because they have more free parameters. Whether these parameters are fitted by formal calibration or by "expert judgment" and fitting by hand makes little practical difference; in either case they make the fitted curves more flexible and thus more conformable to the data. (This comes at the cost of greater parameter uncertainty; more about that below.)

2.a. In the case of the "DDM" in Figure 10 and Table 2, for example, there are FIVE adjustable parameters: b, tau_s, tau_d, P_Ds, and P_Dd (incorrectly labeled as a second "P_Ds" in the table). So Figure 10 shows a five-parameter fit to just six data points (which are themselves not fully indepenent of one another). Is it any surprise that the curve fits well? The other models have at least two parameters, for a data set that effectively has only two or three unique values; those near the peak and those in the 2000's. Again, it is not at all surprising that these can be calibrated to fit the data.

3. Figure 10c is presented as evidence that "the mean residence times were sharply constrained close to 8 years". This is at best unproven and at worst misleading.

3.a. Consider, for example, the red curve for the DDM. In the DDM, the mean residence time (MRT) is a function of three parameters (b, tau_s, and tau_d), and the tritium curve, and thus the fit to the data (SD) is determined by these three parameters, plus two others (P_Ds and P_Dd). It is mathematically impossible for the relationship between MRT (which depends on three independent parameters) and SD (which depends on five independent parameters) to be described by a single curve. There will be multiple combinations of b, tau_s, and tau_d that give the same MRT but different values of SD, and the range of SD will be inflated further by variations in P_Ds and P_Dd.

3.b. The same problem arises, in simpler form, for the EPM and DM. The DM, for example, depends on a residence time and a dispersion parameter P_D; for any individual value of the dispersion parameter, one can draw a curve relating the residence time to the misfit parameter SD. But to describe the relationship between SD and the residence time, one needs a full family of curves, to represent the range of possible values of the dispersion parameter.

3.c. It is impossible to know for sure (since the methods are unacceptably vague on this point), but it seems likely that Figure 10c was generated by choosing fixed values for all-but-one parameter in each model, and then varying just one parameter and tracing out the resulting relationship between MRT and SD.

3.d. From a parameter estimation standpoint, this is a fundamentally flawed procedure, because (1) it ignores the extra degrees of freedom from the other parameters that are arbitrarily held constant, and (2) it therefore underestimates the uncertainty in the MRT, possibly by large factors. This is true even if the parameters were fixed by "expert judgment" rather than algorithms, as long as the experts were free to revise their "judgment" based on whether the tritium curves made sense.

3.e. Methods for multi-variable parameter estimation and uncertainty analysis are widely available. There is no valid excuse for not using them. The revised manuscript must eliminate all claims (explicit or implied) about MRT's estimated from tritium measurements using multi-parameter models, unless and until proper parameter estimation and uncertainty analysis are done.

3.f. There is likewise no valid reason for ignoring the uncertainties in the tritium measurements themselves, and their consequences for parameter uncertainties. Looking at the error bars in Figure 10a, for example, one can estimate that the pooled standard deviation (due to the measurement uncertainties themselves) is about 1-2 TU. Therefore, Figure 10c implies that the MRT is only constrained within about plus or minus two years (for a standard deviation of 1 TU) or about plus or minus four years (for a standard deviation of 2 TU), which is quite a contrast to the paper's assertions that the MRT is "sharply constrained". And this estimate does not even begin to account for the additional uncertainty introduced by the other four parameters. Again, there are standard methods for propagating these uncertainties in parameter estimation, and there is no valid excuse for not using them.

4. As was also pointed out by another reviewer, the claims that compound LPM's have less aggregation bias are not supported by clear lines of reasoning. For example:

4.a. In 3.4.1, the manuscript says that there is little aggregation bias because the simple and compound LPM's have similar mean residence times. But why does this imply an absence of aggregation bias, rather than a similar aggregation bias across all three LPM's? The manuscript also argues that we should expect little aggregation bias because the two model components have MRT's that are similar to, or shorter than, the half-life of tritium. This is only a valid argument if we have independent evidence about the ages of the system components. What evidence do we have that the deep aquifer really contributes 74% of the flow and has a MRT of 10.2 years, instead of (say) 35% of the flow with a MRT of 100 years? If such independent information exists, the reader should be made aware of it. Alternatively, the manuscript needs to demonstrate that

the MRT's of the individual system components can be reliably constrained through parameter estimation (which will not be easy).

4.b. In 3.4.2 and 3.4.3, the claim seems to be that the simple LPM's are subject to aggregation bias because they disagree with each other or with the compound LPM, which fits the data better. But again, the compound LPM has at least twice as many parameters as the simple LPM's, so one would need to somehow show that the better fit does not simply arise from this rather obvious explanation. And of course the simple LPM's will disagree with each other; they have different shapes, so it is unsurprising that they may have different MRT's when fitted to data.

5. One needs to recognize that the abstract's claim that "The choice of a suitable lumped parameter model can be assisted by matching simulations to time series of tritium measurements (underlining the value of long series of tritium measurements)" is mostly a statement about the past, and is misleading as a generalization about the future.

5.a. In the (few) springs and aquifers where tritium analyses were performed decades ago, during and after the bomb peak, those analyses have turned out to be quite useful for comparison with the more recent measurements. Indeed, as Figure 12 shows, it is these early samples that allow one to distinguish between the differently shaped LPM's, and the more recent samples have almost no power to discriminate between those same LPM's.

5.b. And that is precisely the problem: going forward into the future, long time series will be much less useful, for the simple reason that the bomb pulse tritium is largely gone and we are approaching an equilibrium between tritium production and decay. Thus, going forward, long time series will not help, because tritium concentrations are becoming less and less dynamic over time. As the bomb pulse tritium vanishes, we will just be measuring the same value over and over.

5.c. I am sure the authors know this, and it is disingenuous not to make it clear to the

reader, particularly because they celebrate the one clear benefit of the fading of the bomb pulse (the end of double solutions for many tritium models).

5.d. The fading of the bomb pulse will make the parameter estimation problem outlined above even more impossible than it is already. Consider the red curves in Figure 10 as an example. As mentioned above, these are five-parameter fits to six data points. In the future, anywhere that we do not already have measurements of bomb pulse tritium, we will instead have a five-parameter fit to what is effectively just ONE data point (because in equilibrium, all future measurements are redundant).

5.e. There will still be value in sampling across a range of discharges in order to quantify how modeled tritium ages vary with different wetness conditions, as previous work from the New Zealand group has very nicely demonstrated.

Specific comments

1. As other reviewers have pointed out, the organization and clarity of the presentation must be improved. Many necessary details have also been left unmentioned.

2. Needless confusion is created by the alphabet soup of acronyms. Saying "dispersion model", "exponential model", "lumped parameter model", and so on is preferable to forcing your readers to learn a dozen acronyms just so they can get through your paper.

3. Inconsistencies abound. The double exponential piston flow model is called both DEPM and (apparently) BMM. Using inconsistent terminology like this is bad enough, but what's even worse is that readers are never told, and are left to figure this out for themselves. Most of the text uses MTT but some of the figures and captions use MRT, and again readers are never told whether these are the same things or different things. These are just a few examples of a general problem, and it should not be a reviewer's job to flag all these issues.

4. The wiggles in the black curves in Figures 3b and 3c are obvious numerical artifacts,

since the real theoretical curves should be smooth. It is troubling that such visually obvious numerical errors have not been noticed and corrected. One naturally wonders whether there are other technical issues that are less visually obvious, and also have not been caught.

5. at line 8 on page 3, Bethke and Johnson (2008) should be cited; otherwise it looks like the authors are taking credit for this observation.

---

## Short Comment (SC5) · 12 Jan 2017

Julien Farlin's latest comment further misrepresents the work of Luther and Haitjema (1998).

I had commented that Luther and Haitjema's model of confined aquifers relies on the physically impossible assumption that the confining layer somehow "confines" the aquifer by preventing vertical flow, while simultaneously allowing the aquifer to receive

vertical recharge. Farlin's response was to attempt to school me in the difference between confined and semi-confined aquifers, and to claim that Luther and Haitjema modeled a semi-confined aquifer rather than a confined one.

This claim is false. Although Luther and Haitjema do present a model of what they describe as a semi-confined aquifer in their Appendix B, their main paper clearly refers to "confined" and "unconfined" aquifers, but never semi-confined aquifers. And regardless of the terminology that is used, the mathematics of their model makes it clear that confined means confined. In their confined aquifer model, the thickness of the confined aquifer is fixed and the flow field does not feel the effects of a variable free surface. This can only happen if the confining layer is really confining.

If the confining layer is actually confining, then it prevents upward flow through the confining layer wherever the head gradient points upward (this is the only way to keep the aquifer thickness fixed, as their model requires). But their model also requires spatially distributed recharge, including recharge that would have to flow against this upward head gradient. This is obviously nonphysical.

Farlin's latest comment also cites Haitjema (1995) as if it were further independent confirmation for his views, but Haitjema (1995) makes the same nonphysical assumptions as Luther and Haitjema (1998).

Haitjema HM, On the residence time distribution in idealized groundwatersheds, Journal of Hydrology 172, 127-146, 1995.

Luther KH and Haitjema HM, Numerical experiments on the residence time distributions of heterogeneous groundwatersheds, Journal of Hydrology 207, 1-17, 1998).

---

## Author Comment (AC1) · 26 Jan 2017

"Aggregation effects on tritium-based mean transit times and young water fractions in spatially heterogeneous catchments and groundwater systems, and implications for past and future applications of tritium" by Stewart, M.K., Morgenstern, U., Gusyev, M.A., Maloszewski, P., Hydrol. Earth Syst. Sci., doi:10.5194/hess-2016-532.

**Authors' Response to Referee #1**

We appreciate the many helpful comments of Anonymous Referee #1

**Ref #1:** The paper builds on the publications by Kirchner (2016) analyzing the effects of aggregation of transit times distributions on estimating the MTT from 2 systems using tritium as a tracer. In addition, the paper also discusses the effects for a couple of examples from the literature. The paper is well written, however, it could be improved when better linking the examples with the analysis of the aggregation effects, considering from the beginning the use of compound LPMs and improving the readability and celerity of the figures. The list of comments and ideas below should help to improve the paper to be published in HESS.

**Reply:** We will aim to link the aggregation effects in the case studies better with the aggregation effect calculations, introduce the compound LPMs earlier and improve the readability and clarity of the figures.

**Ref #1: General comments:** 1) The definition of heterogeneity needs to be clearer in this paper. Most LPMs assume a certain heterogeneity in the system, otherwise there would not be a transit time distribution – the specific effect considered in this paper regarding the aggregation effects are two distinctly different systems, each heterogeneous, but usually separate flow systems resulting in the investigated effects – this could be – as nicely shown in the examples, a shallow and deep GW system, a confined or unconfined system etc. So, it would help to make this clear, heterogeneity is everywhere, but you are only analysing a specific set of heterogeneity with you analysis.

**Reply:** Simple LPMs are based on the assumption that the hydrological system is homogeneous – this gives rise to the transit time distribution described by the LPM. Our virtual experimental system assumes two different homogeneous subsystems each with their own transit time distributions (TTDs), which are allowed to range from being the same to being very different as shown by their mean transit times (MTTs). Having only two subsystems is the most extreme system, which will show the largest aggregation effects when the MTTs of the subsystems are most different. If there were more subsystems, aggregation effects would be smaller because of averaging over the larger number of subsystems. We agree that this needs to be pointed out.

**Ref #1:** 2) As you mention at some locations in the paper, a compound LPM usually addresses the possible effects of the aggregation error – at least this is how you define it in the example section. Hence, there is no aggregation effect if we would always use compound LPMs – you partly propose this in section 4, but in my opinion, this is not strong enough. If we apply a compound LPM and the two compounds are quite similar, we could also use a simple LPM (maybe simple is not a good phrase, I would prefer single – and there will be no aggregation effect. However, if the two compounds are different, a single LPM would result in a strong

bias. Hence, we should propose to always use compound LPRs and then analyze the results in order to deal with the aggregation effect.

**Reply:** A compound LPM (which is a combination of two simple LPMs) is not likely to be a perfect choice for any particular system, but it will certainly give much less MTT aggregation error than a simple LPM in systems which are prone to aggregation error (i.e. have substantial contributions of water with very young MTT combining with water with old MTT). This topic is also addressed in the replies to Refs #2 and #3. A possible problem with compound LPMs is that there may not be enough data to constrain their parameters.

**Ref #1:** 3) Why are you only using the standard deviation as your objective function. This considers only part of the fit – maybe you should consider the paper of Kling and Gupta or others to better select an appropriate function to be used, in particular when comparing the different models in Table 1.

**Reply:** We have now used the Nash-Sutcliffe Efficiency (NSE) and will add this to Table 1. This gives fits at the same model parameter values as the standard deviation. The qualities of the fits (i.e. the NSE values) however, show different patterns.

**Ref #1:** 4) The definition of the young water fraction is problematic. Why should the threshold (ty) change with changing alpha. So, we would always need to define this ty and then apply it differently to the different sets or models etc. If we use something like the young water fraction, we should define it based on a fixed value that is related to the analysis and the question behind, However, for me the young water fraction is just another measure in order to avoid displaying the whole transit time distribution – this is what we need (as done nicely in Fig 10 – this should be repeated for the other examples as well) and not another factor in addition to the MTT. It would be helpful to discuss this and also to better clarify what the assumption of a non-constant ty would mean with respect to the whole analysis.

**Reply:** This is a good point. After submitting the manuscript, we experimented with keeping the threshold ($t_y$) constant and found that the value of 17.5 years gave consistent results for the young water fraction (YWF) over the whole range of α (0.1 – 10). This will be used in the revised version of the paper.

The larger story here is that we have now changed the method of finding the apparent TTDs by fitting gamma models (GMs) to the mixed tritium concentrations by varying both of the GM parameters (i.e. α and β, with MTT= α*β). Previously (as described in the Discussion paper) the apparent fit was found by varying β only. Aggregation effects on MTTs are similar using both methods, but the threshold (ty) for determining the YWF is now better kept constant at 17.5 years with the revised method.

However, we agree that getting the TTDs right is the important thing if it can be done, and using compound LPMs is an important step in this direction. Then the MTTs are likely to be more accurate. (The point of the YWF was that with the right threshold it would be approximately correct regardless of whether the TTD was correct or not. On the other hand, the MTT would be incorrect if the TTD was incorrect.)

**Ref #1:** 5) The whole section 3.3 is not necessary and should be removed including the figure. It is sufficient if you discuss without showing a similar analysis as already done in Kirchner (2016).

**Reply:** This section is short, and we feel it is better kept in to establish the link with Kirchner's (2016) work on seasonal tracer cycles. Although our methodology was the same as Kirchner's in that two components were combined, we followed a process of starting with the same MTTs and then allowing the second component to become older. In addition, this section shows the dependence of the aggregation error on the difference in MTTs more explicitly than the random sampling of non-similar MTT components method of Kirchner. We will consider expanding this section with the above text in the revised paper.

**Ref #1:** 6) as already mention in some other points, you should make the examples better comparable in terms of defining the compounds LPMs, but also in respect to the figures and results shown. I think you should at least do the following: a) use the most common single LPMs (EPM, EM, GM, DM) and the best compound (or even better also a reference compound model) model and fit it to the observed TU showing the resulting fit (including a quantification of the fit) and the resulting TTD of each model. b) based on these figures, discuss the possible aggregation effects based on the models and the objective functions. c) define MTT, young water fraction for all model combination (in figure or table).

**Reply:** We will do further work on these case studies with these recommendations in mind. Changes will also be made in response to the comments of Refs #2 and #3.

**Ref #1:** 7) In the discussion, you should clearly differentiate what is necessary or problematic when using single LPMs or when using compound LPMs. For example, section 4.1 is only relevant when applying single LPMs.

**Reply:** We will carefully consider this in the rewriting.

**Ref #1:** 8) the statement in section 4.3 that individual tritium measurement can/should be used to estimate a series of TTDs for one system is in my opinion not helpful. We should rather use the series of tritium measurements to apply the best possible compound LPRs that is necessary to describe the heterogeneity of the system as nicely presented in several of the examples. A single tritium measurement is not helpful at all, as there is too much ambiguity. This could only be solved when combining the single tritium measurement with other tracers to identify the individual systems (multi components) and hence the complexity of the system studied. Hence, the whole issue of aggregation error will be solved to begin with.

**Reply:** We (or some of us) have tended to promote use of individual tritium measurements to determine TTDs (at least in the Southern Hemisphere), but we agree that there are often fishhooks especially in Northern Hemisphere systems, e.g. Japanese headwater catchments demonstrated by Gusyev et al. (2016). We support use of other tracers and other types of measurements on hydrological systems, e.g., time-series sampling, to resolve these ambiguities.

**Ref #1: Specific comments:** 1) The title is very long and awkward – it would be helpful to focus on the implications and less on the different ideas of aggregation – please shorten considerable.

**Reply:** Agreed, we will change this.

**Ref #1:** 2) P2/L32 – the tritium method does not only depend on the radioactive decay, but still also on the bomb peak: : :.

**Reply:** Bomb tritium is not required for tritium dating. Detecting the bomb peak was useful for dating in the past, but it is no longer useful in that way. Now the bomb

peak just complicates the age dating based on radioactive decay of tritium and natural cosmogenic tritium is sufficient for dating.

**Ref #1:** 3) P5: The list of LPMs should be more consistent and maybe best in a table listing all used LPMs and possible compound LPMs. In the moment, this section is incomplete, as there are additional compound LPMs in the result section not introduced in the method. You should also consider a systematic use of the compound LPMs – 2P_EM, 2S_EM, or EM_P-EM, EM_S_EM, PFM_P_DM etc, I think this would generally help to get a better understanding how the compound LPMs are set up, in particular the one introduced in the example section.

**Reply:** We agree that the methods section should be tidied up, but feel that a comprehensive listing would be beyond the scope of the paper. We will re-write Equations (8 & 9) in more general form of $LPM_1$ and $LPM_2$ instead of EM allowing the use of any LPM in the equation. The TPLR will be given as an example.

**Ref #1:** 4) P7: why did you select 3 and 197 years for the analysis of the aggregation effect. It would be better the select a more realistic difference, which is commonly observed in the tritium analysis. In addition, you could also use TU/TU of Input as a relative measure in order to avoid defining the constant 2 TU for the analysis.

**Reply:** We selected MTTs of 3 and 197 years to display the relationships between mean transit time and tritium concentration in Fig. 3 because these (1) give a true MTT of 100 years, and (2) there is a good big difference between them. Any two MTTs would do as long as there was a sufficient difference between them.

**Ref #1:** 5) P8/L21: it is not clear to me how the "error of fitting" was determined – please define this in the method section and do not only show it for one selected combination but for all.

**Reply:** The "error of fitting" is the error of the apparent MTT resulting from fitting a simple LPM (a GM) to the tritium concentrations of the mixture of the two subsystem waters. It is one standard deviation. The error decreases as MTT1 increases (and the difference between MTT1 and MTT2 becomes less). Hence we have only shown it for the two lowest MTT1s. However, we will show it for all cases and add a brief explanation in the revised manuscript.

**Ref #1:** 6) P8/L25-30. The discussed small difference between the northern and southern Hemisphere are very difficult to see, as the results are shown in separate figures – either combine the figure to see the effects or use only the figure from one location and mention that the results are similar for the other site (maybe with some kind of quantification).

**Reply:** We will consider some quantification of the small differences.

**Ref #1:** 6) P9/L4: I assume you mean 197 years.

**Reply:** Yes, we fixed the typo.

**Ref #1:** 7) P9/L19: mention that alpha is from the Gamma model – not directly clear in this context.

**Reply:** We will clarify the meaning in the revised manuscript as suggested.

**Ref #1:** 8) P12/L3: The assumption that the old water component includes a zero tritium concentration is not clear for at all – is this possible at all?

**Reply:** Zero tritium concentration means less than the detection limit for tritium which depends on the measurement error at low tritium concentrations (usually taken as twice the standard deviation of measurement). Certainly 1200-year-old water would have far far less tritium than the detection limit.

**Ref #1:** 9) The figure should be improved considerably – the layout is changing, the borders are annoying, the arrangement is not consistent, points cannot be seen (e.g. Fig 6 and 7), style is changing in the paper (Fig 1-9 and 10-end).

**Reply:** The figures will be improved as suggested.

**Ref #1:** 10) Please describe the different lines in Fig 3 in a legend.

**Reply:** Agreed, we will add a legend in Fig 3.

---

## Author Comment (AC2) · 30 Jan 2017

We appreciate the many helpful comments of Anonymous Referee #2, and reply to the detailed comments below.

**Ref #2:** In their manuscript, Stewart et al. investigated tritium-based estimates of mean transit times (MTT) and the fraction of young water (Yf) in light of aggregation bias due to catchment heterogeneities. Furthermore, past studies are reinvestigated and evaluated in respect to aggregation bias. This topic is highly interesting, as most commonly the stable isotopes of water (Oxygen-18, Deuterium) are applied in tracer studies. In comparison to this, tritium is used more seldom, but it has the potential to elucidate longer transit times, where stable isotopes hit a boundary at about 4-5 years. I hope my comments and suggestions will be helpful to the authors and improve the manuscript.

**Reply:** We thank the Referee for this summary.

**Ref #2:** General Comments

1) Manuscript structure: I found the structuring of the text to be all over the place, making it hard to read for me, as I was expecting to have all the tools necessary to understand the paper after reading the Methods. However, the "Results" section basically starts with several paragraphs of new Methods. I would suggest to either changing the order of the text to properly divide Methods, Results and Discussion, or rename the header titles from "Results" and so on to something else to avoid confusion. Please see specific comments about my ideas which paragraph could be shifted to different sections.

**Reply:** We agree with this comment and will revise the Methods, Results and Discussion sections to improve the readability of the paper. The last paragraph of the Introduction section will also be re-written to reflect the updated structure, e.g. describing the young water fraction method. We will also look at whether it will help to separate the Results section into the Fig. 3 part (showing nonlinearity of apparent MTT with tritium concentrations) and Fig. 4 & 5 parts (showing aggregation errors).

**Ref #2:** 2) Methods: After introducing tritium (H3)-based TTD estimation, LPMs and their properties, "Results" starts and I am left with an unsure feeling of how the paper addresses the issues raised in the Introduction. I know you use the four GMs from Fig 2a, but in which combinations for the two virtual catchments? Only selected combinations, or all possible ones? How were the catchments mixed? (I know it is 50:50 because it says so later on, in Results: : :which links back to my comment about structure of the paper). Did you use the GM of each sub-catchment in Equation 1 and forward-propagated Northern and Southern hemisphere H3-data, then mixed it 50:50? All this information is missing, and young water fraction calculation or the literature re-evaluation is not even mentioned here. Furthermore, I think that the description of the individual LPM can be shortened without losing important information. Also, I think Table 1 and showing that the GM can mimic the shapes of other LPM is not essential for understanding of the paper. It is interesting to quickly summarize which LPM is useful for which application, however.

**Reply:** This is helpful comment for revising the Methods. We will update the Methods section with the requested information, and will add an extra subsection describing how the virtual calculations were carried out. (Your summary of how the tritium concentration of the mixture were calculated "Did you use the GM of each sub-catchment in Equation 1 and forward-propagate Northern and Southern

hemisphere H3-data, then mix it 50:50?" is essentially correct, although we might change a few words).

We agree that Table 1 is not essential for the paper, but it helps the reader to use our analyses without investigating all possible combinations of LPMs. The point is that by using the full range of the GM we have covered most of the other simple and frequently used LPMs (i.e. the EPM and DM) as well, since the GM can mimic their useful ranges of shapes. If the estimated errors between GM and other distributions are small in Table 1, GM can be a good approximation. From Table 1, EPM with exponential components (f) less than 0.44 do not give good matches with GM. This occurs when the piston flow component becomes dominant in the EPM, see the horizontal part of the EPM curve in Figure 2b. In such cases, the young water threshold analysis of GM is not applicable and is likely to cause large errors.

**Ref #2:** 3) Yf calculation: It is unclear to me how you calculated this. Yf is determined and calculated from the threshold age ty (Equ. 12), yet it seems the threshold age was calculated by comparison to apparent and true Yf already existing (page 9, first few lines)? True Yf comes from the individual Yf1 and Yf2 (Equ.13), but where are they and their ty1 and ty2 coming from? Also, ty should give good agreement with 10% of apparent and true Yf. 10% of what? What is the 100%? Maybe an explanatory figure would help here, and this should be also in Methods.

**Reply:** We will describe this more clearly in the Methods. As explained in the reply to Ref #1, we have now changed our procedure for calculating the apparent MTTs and Yfs. The new procedure causes very little change to the apparent MTTs, but changes the Yfs in that the threshold age (ty) can now be kept constant for the whole range of α (0.3 – 10). This simplifies the procedure, but there is still the choice of ty to be made. Our present choice of 17.5 years was made essentially by trial and error by determining the biggest value of ty that will still produce minimal aggregation error in Yf for all values of α.

**Ref #2:** 4) Apparent Yf determined from LPM fitted to H3 of mixture: To my understanding, Kirchner 2016 showed that Yf of the mixture can retrieve the "true" Yf (calculated from our knowledge of the virtual system) using a gamma function, but only Yf is valid and the corresponding gamma function itself is not valid (otherwise we would have a valid gamma function and thus a correct MTT again, i.e., no aggregation bias). Equipped with this knowledge, how can we reliably trust the apparent Yf result if it comes from a LPM function that is fitted to the H3 mixture and will most likely not be e.g. gamma distributed anymore, but hyper-gamma distributed?

**Reply:** The Yf can be correct when the TTD is not, because the Yf only includes the part of the TTD with transit times between zero and the young threshold. The TTD becomes more and more flawed towards the long transit time end of its range (e.g. it contains no transit times greater than 4-5 years using seasonal tracer cycles). The MTT (which is an average over the whole range of transit times) is therefore also flawed.

**Ref #2:** 5) Chapter 3.3 seems unnecessary to me, and is very short in itself already. If you want to keep it, please elaborate on its importance.

**Reply:** We will elaborate this section a little, as commented in the reply to Ref #1.

**Ref #2:** 6) Chapter 3.4: I am unclear as to which results were already obtained by the cited studies and which results were calculated by the present manuscript.

**Reply:** So far the calculations are all from the cited studies, but we will make further calculations to explore the MTT and TTD interpretations further.

**Ref #2:** 7) I generally doubt the validity of Chapter 3.4, the literature review. To me there is suddenly a huge leap in logic/faith: that using compound LPM will give the true MTT. Or one that is "truer" than the versions of single LPM. It is assumed that just a good fit of tritium tracer data warrants to say that the model gives true results. I do not say they are wrong, I do not say they are true. I do say we cannot know, or I do not see any evidence here that would substantiate your assumption that the compound LPM would give the true MTT. Even if both parts that feed the mixed water in all described studies would be homogeneous in themselves: the virtual experiment catchments of Kirchner 2016 were also homogeneous in themselves, but different from each other, and still led to aggregation bias. We would need proof that the individual catchments are "homogeneous enough" (whatever that means) and that compound LPM, which are just simplifications of processes that we think occur in a catchment domain, correctly mix the two flows in a way that surely avoids aggregation bias. Just a good fit of observed tritium data is surely not a bad start, but not enough in my opinion. I am in favor or a) deleting Chapter 3.4 OR b) rewriting it much more cautiously, with discussing the considerations uttered here.

**Reply:** This is an important point, which we will address by rewriting this section more carefully and cautiously. We are not intending to say that using a compound LPM removes all possibility of aggregation error, only that using a "well-chosen" compound LPM will help to reduce aggregation error. There is then the problem of what is a "well-chosen" LPM? Our paper considered other information such as chemical and geological information as well as the observed tritium fits to answer this. Further aspects will be considered in the reply to Ref #3.

Specific Comments (page-line)

**Ref #2:** 2-24: "young water" appears here the first time. Maybe define it a bit more clearly. How young does it need to be to be considered young water?

**Reply:** This line refers to "younger water components", and it is a general rather than a specific reference at this point in the paper. It means water components younger than the average of the mixture of water making up the catchment or groundwater system outflow. We will add this explanation to the text.

**Ref #2:** 3-11: "[: : :] the one tracer". This makes it seem to me that two different tracers are used, but I rather get from this paragraph, that actually "when we only have tracer data of the mixture" is meant. Please clarify.

**Reply:** Agreed, we will change this.

**Ref #2:** 3-16: Choice of LPM based on hydrogeological situation: please give an example or reference at this point.

**Reply:** References are Maloszewski and Zuber, 1993 and Maloszewski et al. (2002).

**Ref #2:** 3-18: "water-bearing layers" to avoid confusion while reading (had to read three times)

**Reply:** Ok, we will make this change.

**Ref #2:** 4-6 "times. i.e. The water [: : :]" please correct the capital T and also the period after times seems strange.

**Reply:** We will change this.

**Ref #2:** 4-15: with "calender time" you refer to daily time steps? Monthly? Yearly?

**Reply:** We refer to yearly time steps, and will clarify in the text.

**Ref #2:** 5-23: Starting the sentence with a side-sentence in brackets "(Maloszweski [: : :]" looks weird, in my opinion.

**Reply:** We don't think there is anything wrong with this.

**Ref #2:** 7-6: I would write 2.25 instead of 2.5, if you already use two digits after the comma for 0.05. Except for that I recommend not showing this information anymore, see General comment #2.

**Reply:** We used the expression "about 2.5" which 2.25 is.

**Ref #2:** 7-12 to 7-25: Methods

**Reply:** We think this would cause reader confusion.

**Ref #2:** 7-20: Please explain Fig. 3 a bit more in the manuscript to assist in fully understanding it. As far as I understand it, the black curves show TU that one would measure in streamflow. It seems to be the fitting result of the LPM to the mixed H3 signal (p7-L29f). How did you find the two catchments TU concentrations (necessary to find the mixed TU signal) based on the desired MTTs of 3 and 197 years? There must be some other MTT-TU function behind it, that is not shown? I'm basing my last assumption on Kirchner 2016, where the combination of e.g. two exponential distributions did not lead to another exponential distribution, but a hyper-exponential distribution. Thinking along these lines, this confuses me even more now: every red dot in Fig 3, that is, every TUMTT combination of the two individual catchments, lies exactly on the black curves that come from fitting the mixed runoff TU signal. But according to the logic here, the black curves should be wrong. How can the red dots lie on the black curve, if the shape of the distribution of the mixed runoff is not known and should be some hyper-something version?

**Reply:** We will explain this part better. We have applied a simple procedure which is explained in the paper. In Fig. 3, the red curve is the mixing relationship given by Equations 9 and 11, which have the fraction of the young component (b) as their parameter. The black curve shows the apparent MTT resulting from application of a simple LPM (a piston flow model in Fig. 3a or GM models in Figs. 3b-e) to the mixed tritium concentrations.

**Ref #2:** 7-24: With the assumption of a constant H3 input, are you not basically assuming that no groundwater much older than 50 years significantly contributes to runoff (that is, no groundwater which could possibly include the bomb peak). How realistic is this?

**Reply:** Our statement here is really beside the point, it doesn't matter whether assuming constant tritium input is realistic (in the Southern Hemisphere now or in the Northern Hemisphere in the future) or not. The text from 7-20 to 8-11 describes a thought experiment which allows us to demonstrate the non-linearity of the apparent MTT with tritium concentration, and therefore the reason for the aggregation errors.

**Ref #2:** 7-28: I would get rid of the reference to Equ. 9 here

**Reply:** We don't see the reasoning for this, but we will consider in the revised manuscript.

**Ref #2:** 7-30: Equ. 10 is the standard deviation and seems to be not fitting the text here. Do you mean Equ. 1?

**Reply:** Agreed. We actually meant Eqns 2 and 5.

**Ref #2:** 7-31: All the water in streamflow has the same age, not in soil/aquifer. I would specify that here.

**Reply:** Agreed, we will change this.

**Ref #2:** 8-12: Regarding Fig 4: Earlier it was mentioned that real input TU data would cause scrambled results in Fig 3. But you use real data now. Please clarify why we can suddenly use them, or if the scrambling would just have made analyzing the figure more difficult, but not prevent the data from being used. Also, the paragraph explains Methods (8-12 to 8-18). Additional information is needed: what were the MTT2 increments? Was the GM model used, as you talk about alpha parameter later? How was MTT2 changed then, by changing beta parameter in certain increments?

**Reply:** The reason that the real input data were not used in the calculations shown in Fig. 3, but were used in those in Fig. 4 was because different quantities were plotted in the respective figures. Fig. 3 was used to demonstrate the non-linearity of the apparent MTTs with tritium concentrations; this would have been obscured if the real input data had been used. The real input data could easily have been used, but then we could not have demonstrated the non-linearity because the figures would have been a mess. Fig. 4 demonstrated the difference between the true and apparent MTTs and the real input data could be used (and was necessary) because tritium concentrations were not plotted as one of the axes.

The MTT2 increments were 50 years which causes 25 year increments in True MTT. The GM model was used to calculate the tritium concentrations of the two components. MTT2 was changed by changing $\beta$ since $\tau_m = \alpha\beta$ (5-15). $\alpha$ was kept constant for the two components as shown in the headings in Figs. 4 and 5.

**Ref #2:** 8-20: How were the uncertainties for fitting young waters of MTT1 calculated? Might also be good to call it MTT1 here one time since it is only used in Fig. 4 and leads to some confusion initially when looking at Fig. 4.

**Reply:** The "fitting uncertainties" of the apparent MTTs are the standard deviations of the GM fits to the mixed tritium concentrations. They are only shown for the two youngest MTT1s (3 years and 25 years) because the uncertainties are the biggest for these. However, we will show them for all of the MTT1 cirves.

**Ref #2:** 8-22: The fitting errors are important because of more complex LPM in a good or bad way?

**Reply:** A good way, because big fitting errors should tell the researcher that their simple LPM is misrepresenting the data.

**Ref #2:** 9-4: I guess it is 197 instead of 397.

**Reply:** Yes, it should be 197 years.

**Ref #2:** 9-14: Please define a "reasonable" choice of young water threshold.

**Reply:** We don't think we should define a reasonable choice of ty here, or at least this part of the text needs to be revised. I expect that a reasonable choice could be

inferred from Fig. 3a as being after the portion of the black curve which decreases nearly linearly from 2 TU. The black curve bends much more after 15-20 years.

**Ref #2:** 9-16f: I disagree that cutting-off of the long tail after ty and thus leaving only the short tail will ensure that the apparent Yf does not deviate from the true Yf. As the TTD sums to "1", the long tail influences the short parts of the TTD and vice versa. If one is changed, the other changes too. If we know that the long tail is wrong, we can't be certain that the short TTD part is correct if we basically just ignore the existence of the long tail by cutting it off. To use a metaphor, this is to me like healing a bleeding wound by just not looking at it. And if REALLY the part of the TTD model before ty is correct, how can it be that the part after it is NOT correct? The equation and the parameters to calculate the complete TTD do not suddenly change...we just cut off a certain section of it. If the part before the threshold is true, the part after it is true. If the part after it is wrong, the part before it is wrong. Maybe the question is: how wrong? Significantly? Probably not, looking at results from Kirchner 2016.

**Reply:** The fallacy behind this is that you are assuming that you know the correct TTD. If you have the correct TTD, then your argument is theoretically correct. Since you don't, you can't really say much about the long tail from the short tail and vice versa. Then the best you can do is make measurements, and Kirchner's (2016) and our measurements show that the simple LPM short tails must be near enough to the correct short tails to minimise aggregation error. What you can do with the short tails is subtract them from 1 in order to calculate the approximate amount of water older than the threshold age (this is still not saying anything about the long tails).

**Ref #2:** 9-22: No explanation follows why the reason for this relationship is found in the gamma distribution.

**Reply:** We will remove this part of the text because ty will now be taken as constant. in the revised manuscript.

**Ref #2:** 9-26: 6 to 16 years

**Reply:** Also this text will be removed.

**Ref #2:** 9-28 to 9-31: Discussion

**Reply:** We will consider moving this text to the Discussion section.

**Ref #2:** 10-25: MRT, which I assume to be Mean Residence Time, is introduced for the first time and replaces the MTT without explanation. Please rectify. Also, it should be Fig 10c.

**Reply:** We thank the Referee for spotting these typos and will change the text.

**Ref #2:** 12-21f: Following the reasoning about the different bias thresholds: Does this not mean that using tritium methods for streamflow, we would get bias-free estimates for transit times smaller than 6 years, which must include the seasonal cycle results if we would apply it to the stream, and ultimately agree with them?

**Reply:** We are not sure what is meant here. Using tritium, aggregation bias should be small for MTTs less than about 17.5 years (or present ty). Tritium concentrations show only a small seasonal variation (referred to as the "spring leak") and it has not so-far been found useful for age-dating. Using seasonal tracer cycles (stable isotopes or chloride), aggregation bias should be small for MTTs less than 2-3 months according to Kirchner (2016).

**Ref #2:** Title of 4.2: Consider changing to "How much has aggregation affected tritium MTTs in past studies?"

**Reply:** We thank the Referee for this suggested title and will change the title accordigly.

**Ref #2:** 12-30: Conclusion #1: I disagree with ONLY affected by bias if "older than 6 years" and "if determined by simple LPM", for reasons already explained above.

**Reply:** It may be wise to soften this statement a little, but we basically believe it to be a fair statement.

**Ref #2:** 13-1f: If we take the variance into account in the given examples of 10 plusminus 8 and 10 plusminus 5, there seem to be quite a few catchments that have less than 6 years MTT.

**Reply:** Yes there are some, but the very large variances are mostly caused by outliers on the old side not the young side. Most of the MTTs are close to the mean or just below it.

**Ref #2:** 13-8 Conclusion #2: As mentioned above, I see no evidence for this statement.

**Reply:** The four case studies in Section 3.4 are examples of this statement. Also the Blavoux et al (2013) study described from 13-12 to 13-18. And four more key examples are given from 13-19 to 13-27.

**Ref #2:** 14-23: I must have missed the part in the manuscript that shows that simple LPM still work in case of long series of tritium measurements? Where is that shown?

**Reply:** If the simple LPM accurately represents the hydrological system, then the MTT derived from it will not have much aggregation bias. This is equivalent to the cases in the virtual experiments where MTT1 and MTT2 are the same and the true and apparent MTTs of the mixture plot on the 1:1 line in Figs. 4 and 5. An indication that the simple LPM provides a good description of the hydrological system is given if the simple LPM provides a good fit to a long series of tritium measurements. We will add a short sentence about this in the revised manuscript.

**Ref #2:** Table 1: In the description change "The shape parameter of the best-fitting versions of the other models [: : :]", since it is not always the shape parameter for the other models, e.g. it is the dispersion parameter for DM.

**Reply:** The dispersion parameter is the shape parameter for the dispersion model since it controls the shape of its MTT.

**Ref #2:** Figure 2a: the scale parameter beta was fixed for each GM? Which value did it have?

**Reply:** $\beta$ is fixed for each case depending on the value used for $\tau_m$, since $\beta=\tau_m/\alpha$ (5-17). With $\tau_m=1$, $\beta$ ranges from 3.3 to 0.1 for $\alpha$ ranging from 0.3 to 10.

**Ref #2:** Figure 4: In the legend: the orange MTT1 actually says "MTT!"

**Reply:** We fixed the typo and thank the Referee for spotting it.

**Ref #2:** Figure 7 is not mentioned in the manuscript.

**Reply:** We added a reference to Fig. 7 in the revised manuscript.

**Ref #2:** Generally the figures should be unified more in layout, e.g., get rid of the outer border.

**Reply:** Ok. We will revise these.

**References**

Blavoux, B., Lachassagne, P., Henriot, A., Ladouche, B., Marc, V., Beley, J.-J., Nicoud, G., and Olive, P.: A fifty-year chronicle of tritium data for characterising the functioning of the Evian and Thonon (France) glacial aquifers, J. Hydrol., 494, 116–133, doi.org/10.1016/j.jhydrol.2013.04.029, 2013

Kirchner, J.W.: Aggregation in environmental systems – Part 1: Seasonal tracer cycles quantify young water fractions, but not mean transit times, in spatiallyheterogeneous catchments, Hydrol. Earth Syst. Sci., 20, 279-297, doi:10.5194/hess-20-279-2016, 2016.

---

## Author Comment (AC3) · 7 Feb 2017

We appreciate the many helpful comments of Anonymous Referee #3.

**Ref #3:** General comments
The results summarized in the first paragraph of the abstract and Figures 1-9 of the manuscript itself are, to the best of my knowledge, broadly correct. However, the results reported in the second paragraph (and the related discussion and analyses in section 3.4 and 4) need to be reconsidered.

**Reply:** We will reconsider the second paragraph of the abstract and related analyses and discussion as suggested by the Referee.

**Ref #3:** 1. The statement that "well-chosen compound lumped parameter models should be used as they will eliminate potential aggregation errors due to the application of simple lumped parameter models" directly implies that aggregation errors only arise with simple LPM's, but not with compound LPM's, or at least not with "well-chosen" ones (whatever that means).

**Reply:** We stand by this statement, except that we would now replace the word "eliminate" by "reduce or greatly reduce". In general, "well-chosen" means that the compound lumped parameter model (LPM) captures important aspects of the water flow in the catchment or groundwater system, as in a conceptual model. In particular, delineation between subsystems delivering young water and old water to the system outflow is very important in this context because the aggregation error is caused by the disproportionate effect of young water.

**Ref #3:** 1.a. This is not consistent with the analysis presented elsewhere in the paper. Figures 3-5 show clear aggregation errors from the use of simple lumped parameter models, but they would also show clear aggregation errors if more complex lumped parameter models were used. For example, a gamma model with alpha=0.3 closely approximates a compound LPM, but it is clearly vulnerable to aggregation errors, as shown in Figures 4 and 5.

**Reply:** As noted above, a compound model would have to have a transit time distribution (TTD) representing the relative contributions of young and old water subsystems to be able to reduce potential aggregation errors. This separation can be achieved approximately by a binary or more complicated LPM, but not by a simple LPM. If a simple LPM (such as a gamma model with alpha=0.3) accurately described the TTD of a system, it would not have much aggregation error. (This is shown by the increasing aggregation error as the contrast between the ages of the two water components increase in Fig. 4a).

**Ref #3:** 1.b. Since the analyses in Figures 3-5 have been used to demonstrate aggregation errors in simple LPM's, exactly the same analyses must be applied to compound LPM's to demonstrate that these aggregation errors disappear. Until this is done, the claims in the abstract have not been demonstrated, and must be removed.

**Reply:** We reject this demand, because the Referee has misunderstood the situation. It is obvious that compound (especially binary) models would work very well with the virtual experiments carried out here (i.e. the analyses in Figures 3-5), because the experiments are based on combining two subsystems with the same or different mean transit times. Optimised compound models would simply separate the components back out again and produce (near) zero aggregation error. Applying compound models to the results of the virtual experiments is of course not the same

thing as experimenting with real systems, which is why we considered some cases from the literature.

**Ref #3:** 2. In some of the examples that are presented, the compound LPM's clearly fit the data better, but of course they should, because they have more free parameters. Whether these parameters are fitted by formal calibration or by "expert judgment" and fitting by hand makes little practical difference; in either case they make the fitted curves more flexible and thus more conformable to the data. (This comes at the cost of greater parameter uncertainty; more about that below.)

**Reply:** What the Referee has not understood is that the improvement in fit to the data by an optimised compound LPM (due to there being more free parameters) means that the model more accurately represents the TTD of the system. This means that the compound LPM has less potential to produce aggregation error.

**Ref #3:** 2.a. In the case of the "DDM" in Figure 10 and Table 2, for example, there are FIVE adjustable parameters: b, tau_s, tau_d, P_Ds, and P_Dd (incorrectly labeled as a second "P_Ds" in the table). So Figure 10 shows a five-parameter fit to just six data points (which are themselves not fully independent of one another). Is it any surprise that the curve fits well? The other models have at least two parameters, for a data set that effectively has only two or three unique values; those near the peak and those in the 2000's. Again, it is not at all surprising that these can be calibrated to fit the data.

**Reply:** This criticism is also invalid because it takes no account of the wealth of evidence for this particular compound LPM (i.e. the DDM) presented in Stewart and Thomas (2008). This evidence is referred to in the sentence (P10-L22): "The DDM was used because δ18O and Cl measurements showed that there were two separate water systems contributing to the Main Spring (a shallow system and a deep system)". We also reject the description of the tritium dataset as "effectively … only two or three unique values". There are six fully independent measurements covering 40 years (including most of the rise and fall of the bomb tritium peak).

Evidence in Stewart & Thomas includes determination of recharge and discharge for the catchment and Main Spring based on extensive flow measurements and the 18O balance. The two flow systems are identified by the 18O values and most clearly by their chloride concentrations, since chloride is a powerful tracer here because the deep system contains a small proportion of sea water while the shallow one does not. The proportions of the flows (shallow fraction, b) for the Main Spring comes from the relationships between the flows, 18O and chloride and concentrations. The dating was based on tritium, CFC-11 and 18O evidence (CFC-12 gave anomalous results as it frequently has done in other groundwater studies).

**Ref #3:** 3. Figure 10c is presented as evidence that "the mean residence times were sharply constrained close to 8 years". This is at best unproven and at worst misleading.

**Reply:** See replies to sections 3 and 3a – 3d following 3e and 3f.

**Ref #3:** 3.a. Consider, for example, the red curve for the DDM. In the DDM, the mean residence time (MRT) is a function of three parameters (b, tau_s, and tau_d), and the tritium curve, and thus the fit to the data (SD) is determined by these three parameters, plus two others (P_Ds and P_Dd). It is mathematically impossible for the relationship between MRT (which depends on three independent parameters) and SD (which depends on five independent parameters) to be described by a single

curve. There will be multiple combinations of b, tau_s, and tau_d that give the same MRT but different values of SD, and the range of SD will be inflated further by variations in P_Ds and P_Dd.

3.b. The same problem arises, in simpler form, for the EPM and DM. The DM, for example, depends on a residence time and a dispersion parameter P_D; for any individual value of the dispersion parameter, one can draw a curve relating the residence time to the misfit parameter SD. But to describe the relationship between SD and the residence time, one needs a full family of curves, to represent the range of possible values of the dispersion parameter.

3.c. It is impossible to know for sure (since the methods are unacceptably vague on this point), but it seems likely that Figure 10c was generated by choosing fixed values for all-but-one parameter in each model, and then varying just one parameter and tracing out the resulting relationship between MRT and SD.

3.d. From a parameter estimation standpoint, this is a fundamentally flawed procedure, because (1) it ignores the extra degrees of freedom from the other parameters that are arbitrarily held constant, and (2) it therefore underestimates the uncertainty in the MRT, possibly by large factors. This is true even if the parameters were fixed by "expert judgment" rather than algorithms, as long as the experts were free to revise their "judgment" based on whether the tritium curves made sense.

3.e. Methods for multi-variable parameter estimation and uncertainty analysis are widely available. There is no valid excuse for not using them. The revised manuscript must eliminate all claims (explicit or implied) about MRT's estimated from tritium measurements using multi-parameter models, unless and until proper parameter estimation and uncertainty analysis are done.

**Reply:** This demand would have eliminated all tritium papers in the past up to that recently published by Gallart et al. (2016)! As far as we know, their paper is the only attempt to apply multi-variable parameter estimation methods to tritium measurements, and it was by no means a trivial exercise. Presumably the Referee would have cited any relevant earlier work if any had been available. Gallart et al. produced complex parameter diagrams with multiple solutions, from which likelihood-weighted cumulative density functions were determined. However, their work was concerned with a Northern Hemisphere location with a relatively short record (1996 to 2013), so multiple solutions were to be expected (Stewart and Morgenstern, 2016). The very different Southern Hemisphere tritium input function and the sample record from 1966 would give very different and very much more straight-forward parameter diagrams for the Waikoropupu Springs (Stewart and Morgenstern, 2016). It will be interesting to apply the methods to Southern Hemisphere locations and we plan to carry out such calculations, but we think it is beyond the scope of this paper.

**Ref #3:** 3.f. There is likewise no valid reason for ignoring the uncertainties in the tritium measurements themselves, and their consequences for parameter uncertainties. Looking at the error bars in Figure 10a, for example, one can estimate that the pooled standard deviation (due to the measurement uncertainties themselves) is about 1-2 TU. Therefore, Figure 10c implies that the MRT is only constrained within about plus or minus two years (for a standard deviation of 1 TU) or about plus or minus four years (for a standard deviation of 2 TU), which is quite a contrast to the paper's assertions that the MRT is "sharply constrained". And this estimate does not even begin to account for the additional uncertainty introduced by

the other four parameters. Again, there are standard methods for propagating these uncertainties in parameter estimation, and there is no valid excuse for not using them.

**Reply:** We have estimated the effects of the tritium measurement errors and considered the effects of adjusting the (previously unadjusted) model parameters below. Tritium data from the Waikoropupu Spring are given in Table 1 (data from Table 7 in Stewart and Thomas, 2008). The fractional tritium measurement error decreased between 1966/76 and 1998/2006 because of methodological improvements (Morgenstern and Taylor, 2009).

Table 1: Tritium concentrations in the Main Spring of the Waikoropupu Springs. Errors are one standard deviation.

| Date | Tritium (TU) | Error (TU) | Fractional error |
|------|------|------|------|
| 27-05-66 | 14.0 | 0.9 | 0.064 |
| 29-07-72 | 15.2 | 1.9 | 0.125 |
| 20-03-76 | 11.0 | 1.2 | 0.109 |
| 26-02-98 | 2.25 | 0.06 | 0.027 |
| 16-03-99 | 2.08 | 0.08 | 0.038 |
| 21-03-06 | 1.53 | 0.05 | 0.033 |

Table 2 gives the mean transit times determined taking account of the measurement errors. We have considered a worst case scenario by fitting LPMs to "Low", "Mid" and "High" cases, in which all of the errors are subtracted from the tritium measurements in the "Low" case, and all are added on in the "High" case. This will give a larger range of MTTs than a Monte Carlo sampling technique would have done, because the measurements have uncorrelated errors and there is very low likelihood that all of the measurements would have been low or high together. The fits were optimised using different numbers of the model parameters (as shown in column 2). The parameter b (fraction of the shallow system) was set at 0.26 ± 0.10 based on flow and 18O balance measurements (Table 4, Stewart and Thomas, 2008). (This produced a relatively minor change in MTT of ±0.2 years about the value with b=0.26.) The range of variation around the mid MTT (1.4 to 2.2 years, with one outlier) gives a good indication of the uncertainty of the mean transit times. We consider that this qualifies as "sharply constrained".

Table 2: Mean transit times ($\tau_m$) determined by fitting the LPMs to the tritium concentrations by adjusting the parameters shown. "Low", "Mid" and "High" columns show the $\tau_m$ obtained for tritium value minus error, tritium value and tritium value plus error. The plus/minus column shows the average variation around the mid age. Parameters not adjusted in a particular LPM (i.e. not listed in column 2) were set at their optimised mid values. Parameter b for the DDM was set at 0.26 based on other measurements.

| LPM | Parameters | Low $\tau_m$ (yr) | Mid $\tau_m$ (yr) | High $\tau_m$ (yr) | $\pm$ (yr) |
|------|-----------|---------|---------|---------|------|
| EPM | $\tau_m$ | 9.5 | 7.9 | 6.7 | 1.4 |
| EPM | $\tau_m$, f | 9.5 | 8.7 | 6.6 | 1.5 |
| DM | $\tau_m$ | 10.4 | 8.2 | 6.5 | 1.9 |
| DM | $\tau_m$, $P_D$ | 13.0 | 8.7 | 5.7 | 3.8 |
| DDM | $\tau_d$ | 10.0 | 7.9 | 6.3 | 1.9 |
| DDM | $\tau_s$, $\tau_d$, | 9.3 | 7.8 | 6.2 | 1.6 |
| DDM | $\tau_s$, $\tau_d$, $P_{Ds}$, $P_{Dd}$ | 10.2 | 7.7 | 5.8 | 2.2 |

**Ref #3:** 4. As was also pointed out by another reviewer, the claims that compound LPM's have less aggregation bias are not supported by clear lines of reasoning. For example:

**Reply:** We think that there are clear lines of reasoning for this claim, which perhaps we have not explained clearly enough. Simple LPMs assume homogeneous systems, compound ones are binary or more complicated systems. In terms of our virtual experiments, the simple LPMs yield the "apparent" MTTs while the binary LPMs yield the "true" MTTs; in these experiments the "true" MTTs actually are true because we have built the systems by adding two subsystems together (in the proportions of 1:1 for convenience) making binary systems. When the two subsystems have the same MTTs, the simple and compound LPMs yield the same MTTs (and therefore plot on the 1:1 line in Fig. 4). As the subsystem MTTs become more and more different, so the simple (i.e. apparent) and compound (i.e. true) MTTs become more different and the aggregation error increases. This is quite clear and so is the reason for it, i.e. that the young water component outweighs the old water component (or as shown in Fig. 3, the relationship between MTT and tritium concentration is non-linear).

When it comes to applying simple and compound LPMs to real systems (the four case studies), the simple LPMs yield apparent MTTs, but the compound LPMs yield true MTTs if and only if they capture the separation between young and old water subsystems in the overall system. If the compound model does not capture this separation, then it may not help in terms of aggregation error. In the case of the virtual experiments, the compound (i.e. binary) model would of course be perfect, because that is how we have set the experiments up.

To summarise, application of a compound LPM to real systems will greatly reduce the possibility of aggregation error, if it captures approximately the separation between young and old water subsystems. The question then becomes: how well do the compound models capture this separation in the four case studies described in Section 3.4? The examples were chosen to cover the tritium range and because there was evidence that simple LPMs had been inadequate in their cases (the evidence included both hydrological/chemical/geological and tritium evidence). We think that the compound models are well-chosen (in the sense defined above) in these four cases. But we agree that explanation of this background and further analysis of the compound models is required (as pointed out by Refs #2 and #3).

**Ref #3:** 4.a. In 3.4.1, the manuscript says that there is little aggregation bias because the simple and compound LPM's have similar mean residence times. But why does this imply an absence of aggregation bias, rather than a similar aggregation bias across all three LPM's? The manuscript also argues that we should expect little aggregation bias because the two model components have MRT's that are similar to, or shorter than, the half-life of tritium. This is only a valid argument if we have independent evidence about the ages of the system components. What evidence do we have that the deep aquifer really contributes 74% of the flow and has a MRT of 10.2 years, instead of (say) 35% of the flow with a MRT of 100 years? If such independent information exists, the reader should be made aware of it. Alternatively, the manuscript needs to demonstrate that the MRT's of the individual system components can be reliably constrained through parameter estimation (which will not be easy).

**Reply:** We have explained why there is little aggregation bias if the simple and well-chosen compound LPMs give the same MTTs (just above). All three of the optimised LPMs for the Waikoropupu Main Spring match the tritium concentrations very well, and they all give very similar TTDs.

There would have been no need to invoke a compound LPM, except that there is overwhelming evidence that the springs are fed by two flow subsystems (Stewart and Thomas, 2008). There is no possibility that (for example) the deep aquifer contributes 35% of the flow with an MTT of 100 years, as perusal of Stewart and Thomas (2008) would make clear. 1. The recharge/discharge model for the springs and overall system would need to be completely different. 2. The 18O concentrations would not balance, and its variations in time would need to be different. 3. The chloride concentrations would need to be different. 4. The relationships between the flows, chloride concentrations and 18O in the springs would need to be different. 5. The tritium concentrations in the Springs would need to be lower. 6. The CFC-11 concentrations in the springs would need to be lower. We will include this explanation in the revised paper as requested by the Referee.

**Ref #3:** 4.b. In 3.4.2 and 3.4.3, the claim seems to be that the simple LPM's are subject to aggregation bias because they disagree with each other or with the compound LPM, which fits the data better. But again, the compound LPM has at least twice as many parameters as the simple LPM's, so one would need to somehow show that the better fit does not simply arise from this rather obvious explanation. And of course the simple LPM's will disagree with each other; they have different shapes, so it is unsurprising that they may have different MRT's when fitted to data.

**Reply:** We applied one simple and one compound LPM in each of these two cases. Our claim is that there is aggregation bias in each case because there is disagreement between the simple and the compound LPMs. This of course requires that the compound LPMs give good representations of their catchments in regard to separation of young and old water subsystems, i.e. that the compound LPMs be "well-chosen" for their catchments. Compound LPMs have the possibility of correctly combining different parts of the catchment with different characteristics when they are optimised to the data, whereas simple LPMs do not.

We agree that we need to establish more conclusively that these compound LPMs are in fact well chosen for these catchments. For the Kuratau River (Sect. 3.4.2), geological evidence strongly supports two subsystems within the catchment. The

area within the catchment with the very impermeable Whakamaru Group ignimbrites and andesitic and basaltic lavas produces very young water, while the area with the highly permeable Taupo/Oruanui ignimbrites and tephras produces much older water (Morgenstern, 2007). The highly contrasting permeabilities of these rocks is corroborated by observations in adjacent catchments. Distributed groundwater models calibrated with groundwater levels, river discharges and tritium concentrations also substantiated these flows and their contrasting ages (Gusyev et al., 2013; 2014).

For Hangarua Spring and Hamurana Stream (Sect. 3.4.3), and for many other streams and springs drawing from the Mamaku Ignimbrite plateau, two different flow contributions are demonstrated by the tritium measurements (Morgenstern et al., 2015). These contributions are (relatively) young water from shallow aquifers seen in minor streams maintained by shallow aquifers, and old water from deep aquifers seen in aquifers with very deep groundwater tables in the area (Rosen et al., 1998).

**Ref #3:** 5. One needs to recognize that the abstract's claim that "The choice of a suitable lumped parameter model can be assisted by matching simulations to time series of tritium measurements (underlining the value of long series of tritium measurements)" is mostly a statement about the past, and is misleading as a generalization about the future.

**Reply:** We thank the Referee for this comment, and acknowledge that there is some truth in this. Note that we really meant "(underlining the value of long series of past tritium measurements)". There is no doubt that identifying LPMs from tritium data will become problematical in the future. However, Northern Hemisphere hydrological systems still contain some bomb tritium and although this can cause problems with ambiguous ages (see Gusyev et al. (2016)), it can also assist identification of LPMs (although identifying suitable *compound* LPMs may be a step too far). Gallart et al. (2016) used Monte Carlo sampling to account for measurement error in tritium and parameter estimation errors to demonstrate that tritium measurements taken now combined with future measurements will enable effective identification of MTTs provided high quality tritium measurements are used (their Fig. 13). They used the EPM model.

Southern Hemisphere systems contain much less bomb tritium and identification of LPMs (i.e. mixing models) is becoming more difficult unless past tritium data is available. High quality measurements are essential, especially in the Southern Hemisphere, because of the low levels of cosmogenic tritium. There are small seasonal variations in cosmogenic tritium (see data for the last 25 years for Kaitoke NZ in Fig. 1) that may become more useful in the future. This situation has already been reported for some areas in Japan (Gusyev et al. 2016) and will occur in other Northern Hemisphere areas.

**Ref #3:** 5.a. In the (few) springs and aquifers where tritium analyses were performed decades ago, during and after the bomb peak, those analyses have turned out to be quite useful for comparison with the more recent measurements. Indeed, as Figure 12 shows, it is these early samples that allow one to distinguish between the differently shaped LPM's, and the more recent samples have almost no power to discriminate between those same LPM's.

5.b. And that is precisely the problem: going forward into the future, long time series will be much less useful, for the simple reason that the bomb pulse tritium is largely

[Figure]

Figure 1. Tritium concentrations in monthly precipitation at Kaitoke, New Zealand. The sizes of the points show the measurement errors (±0.03 TU).

gone and we are approaching an equilibrium between tritium production and decay.

Thus, going forward, long time series will not help, because tritium concentrations are becoming less and less dynamic over time. As the bomb pulse tritium vanishes, we will just be measuring the same value over and over.

**Reply:** We are not at this stage yet in the Northern Hemisphere, but when we are in 10 to 20 years we will not be able to use tritium for identifying the type of mixing model. To determine MTTs will require assuming a mixing model based on other criteria.

Actually we are unlikely to be in the situation of measuring the same value over and over because systems (especially streams) are inherently unsteady even during baseflow and as the MTT changes so will the tritium concentration. (But again we will need time-series with high-quality measurements to exploit this.)

**Ref #3:** 5.c. I am sure the authors know this, and it is disingenuous not to make it clear to the reader, particularly because they celebrate the one clear benefit of the fading of the bomb pulse (the end of double solutions for many tritium models).

**Reply:** We have made the situation clear in other papers, e.g. we urged Northern Hemisphere researchers to start sampling now before all of the bomb tritium is gone (Stewart et al., 2012; Stewart and Morgenstern, 2016). However, we will re-emphasize the point in the revised paper.

**Ref #3:** 5.d. The fading of the bomb pulse will make the parameter estimation problem outlined above even more impossible than it is already. Consider the red curves in Figure 10 as an example. As mentioned above, these are five-parameter fits to six data points.

In the future, anywhere that we do not already have measurements of bomb pulse tritium, we will instead have a five-parameter fit to what is effectively just ONE data point (because in equilibrium, all future measurements are redundant).

**Reply:** There will certainly be complications in the future use of tritium, e.g. those connected with the fading of the bomb pulse, those with variations in flow, those due to aggregation bias (unfortunately aggregation bias still applies with cosmogenic tritium input). However, as pointed out in the paper, not much aggregation bias is expected in systems (such as many catchments) that have MTTs in the range of one to two decades. One cannot completely predict how things will turn out, but we can foresee that future measurements of tritium will not be redundant.

**Ref #3:** 5.e. There will still be value in sampling across a range of discharges in order to quantify how modeled tritium ages vary with different wetness conditions, as previous work from the New Zealand group has very nicely demonstrated.

**Reply**: Agreed.

**Ref #3:** Specific comments
1. As other reviewers have pointed out, the organization and clarity of the presentation must be improved. Many necessary details have also been left unmentioned.

**Reply:** We will reorganise the paper as noted in the replies to this and the other referees.

**Ref #3:** 2. Needless confusion is created by the alphabet soup of acronyms. Saying "dispersion model", "exponential model", "lumped parameter model", and so on is preferable to forcing your readers to learn a dozen acronyms just so they can get through your paper.

**Reply:** Different people have different preferences in terms of jargon, but we agree with Ref #3 and will reduce the jargon to improve the readability of the paper.

**Ref #3:** 3. Inconsistencies abound. The double exponential piston flow model is called both DEPM and (apparently) BMM. Using inconsistent terminology like this is bad enough, but what's even worse is that readers are never told, and are left to figure this out for themselves. Most of the text uses MTT but some of the figures and captions use MRT, and again readers are never told whether these are the same things or different things.

These are just a few examples of a general problem, and it should not be a reviewer's job to flag all these issues.

**Reply:** We thank the Referee for spotting these inconsistencies, which we will remove.

**Ref #3:** 4. The wiggles in the black curves in Figures 3b and 3c are obvious numerical artifacts, since the real theoretical curves should be smooth. It is troubling that such visually obvious numerical errors have not been noticed and corrected. One naturally wonders whether there are other technical issues that are less visually obvious, and also have not been caught.

**Reply:** We will correct these in the revised paper.

**Ref #3:** 5. at line 8 on page 3, Bethke and Johnson (2008) should be cited; otherwise it looks like the authors are taking credit for this observation.

**Reply:** We will cite this reference.

**References**

Gallart, F., Roig-Planasdemunt, M., Stewart, M.K., Llorens, P., Morgenstern, U., Stichler, W., Pfister, P., Latron, J. A GLUE-based uncertainty assessment framework for tritium-inferred transit time estimations under baseflow conditions. Hydrological Processes (Keith Beven Tribute), 30, 4741-4760, 2016. DOI: 10.1002/hyp.10991

Gusyev, M.A., Abrams, D., Toews, M.W., Morgenstern, U., and Stewart, M.K. A comparison of particle-tracking and solute transport methods for simulation of tritium concentrations and groundwater transit times in river water. Hydrol. Earth Syst. Sci. 18, 3109-3119, 2014. doi:10.5194/hess-18-3109-2014

Gusyev M.A., Toews M., Morgenstern U., Stewart M., White P., Daughney C. and Hadfield J. Calibration of a transient transport model to tritium data in streams and simulation of ground water ages in the western Lake Taupo catchment, New Zealand, Hydrol. Earth Syst. Sci. 17, 1217-1227, 2013, doi:10.5194/hess-17-1217-2013,

Gusyev, M.A., Morgenstern, U., Stewart, M.K., Yamazaki, Y., Kashiwaya, K., Nishihara, T., Kuribayashi, D., Sawano, H., and Iwami, Y. Application of tritium in precipitation and baseflow in Japan: a case study of groundwater transit times and storage in Hokkaido watersheds. Hydrol. Earth Syst. Sci. 20, 3043-3058, 2016. doi:10.5194/hess-20-3043-2016

Gusyev, M.A., Toews, M., Morgenstern, U., Stewart, M., White, P., Daughney, C., Hadfield, J. Calibration of a transient transport model to tritium data in streams and simulation of groundwater ages in the western Lake Taupo catchment, New Zealand. Hydrology and Earth System Sciences 17(3), 1217-1227, 2013.Morgenstern, U.: Lake Taupo streams –Water age distribution, fraction of landuse impacted water, and future nitrogen load. Environment Waikato Technical Report TR 2007/26, Document #1194728, p. 28, 2007.

Morgenstern, U., Daughney, C.J., Leonard, G., Gordon, D., Donath, F.M. and Reeves, R. Using groundwater age and hydrochemistry to understand sources and dynamics of nutrient contamination through the catchment into Lake Rotorua, New Zealand. Hydrol. Earth Syst. Sci., 19, 803–822, 2015. doi:10.5194/hess-19-803-2015

Morgenstern U, Taylor CB. Ultra low-level tritium measurement using electrolytic enrichment and LSC. Isotopes in Environmental and Health Studies 45: 96–117, 2009. DOI:10.1080/10256010902931194

Rosen, M. R., Milner, D.,Wood, C. P., Graham, D., and Reeves, R. Hydrogeologic investigation of groundwater flow in the Taniwha Springs area. Institute of Geological and Nuclear Sciences Client Report 72779C.10, GNS Science, Lower Hutt, New Zealand, 1998.

Stewart, M.K., Morgenstern, U. 2016: Importance of tritium-based transit times in hydrological systems. WIREs Water 3, 145–154. doi: 10.1002/wat2.1134

Stewart, M.K., Morgenstern, U., McDonnell, J.J., Pfister, L. The 'hidden streamflow' challenge in catchment hydrology: A call to action for streamwater transit time analysis. Hydrol. Process. 26(13), 2061-2066, 2012. doi: 10.1002/hyp.9262.

Stewart, M.K. and Thomas, J.T.: A conceptual model of flow to the Waikoropupu Springs, NW Nelson, New Zealand, based on hydrometric and tracer (18O, Cl, 3H and CFC) evidence, Hydrol. Earth Syst. Sci. 12, 1-19, 2008. http://www.hydrol-earth-syst-sci.net/12/1/2008/hess-12-1-2008.html

---

## Author Comment (AC4) · 7 Feb 2017

We thank Julien Farlin and Jim Kirchner for interesting Comments on our paper (Stewart et al.). These raise important questions, some of which are beyond the scope of our paper, but we give some observations below in response to their comments.

Comment SC1 by J. Farlin

We agree with Farlin's deduction from Fig. 3 in Stewart et al. that "The relationships between mean transit time and tritium activity shown on Figure 3 clearly display approximately linear segments over which the mixture of water coming in equal volume from two different subcatchments would lead to a negligible underestimation of the true MTT (for instance, on figure 3d for MTTs between 0 and 20 years)."

Farlin asks: "What degree of heterogeneity, and hence how large a difference in subcatchments' MTTs, can usually be expected in real-world catchments?". Stewart et al. looked for evidence of real-world aggregation error by examining some tritium dating case studies from the literature. These were chosen to cover the range of tritium dating (0-200 years) and were cases where we knew there were "few and distinct" heterogeneities in the catchments (in the words of Luther and Haitjema, 1998). These cases showed substantial aggregation error for MTTs greater than about 10-20 years. Other case studies in which there were apparently no "few and distinct" heterogeneities could have been chosen; we expect that these would have shown considerably less aggregation error.

Farlin questions conclusions drawn from the use of Kirchner's (2016) and Stewart et al's virtual experimental model (water from two subsystems with different MTTs mixing to give the system outflow) as being too conservative for real-world systems (i.e. as indicating too much aggregation error). The model produces aggregation errors that range from zero when the MTTs of the subsystems are the same to very large when the MTTs are very different. Luther and Haitjema (1998) is cited to assert that most real-world groundwater systems are effectively relatively homogeneous (describable by a simple exponential LPM at baseflow). Many catchments, on the other hand, have outflows composed of quickflow and baseflow, which could have very different MTTs leading to large aggregation errors. We note Farlin's support for our suggestion that using compound LPMs could reduce aggregation errors greatly by conceptually catering for young and old MTT waters within catchments or groundwater systems.

We also thank him for the comment about tritium input and will explain that tritium data for Trier before 1978 was calculated by regression from Vienna data.

Comment SC3 by J. Kirchner

Kirchner says that an important question is: "Given how little we know about the patterns of heterogeneity in catchments' characteristics and/or their transit time distributions, how sanguine should we be about the risk of aggregation errors?". He

answers: "We know that important catchment properties (hydraulic conductivity, depth to bedrock, soil characteristic curves, etc.) typically vary by large factors, in spatially correlated fashion, across all the scales at which they can be measured. Given this pervasive multiscale heterogeneity, the burden of proof should be on those who claim that it doesn't matter, or who want to use techniques that are prone to aggregation errors (such as estimating MTT from seasonal tracer cycles). Alternatively, we should develop – and use – methods that are much less vulnerable to aggregation errors (such as the young water fraction concept presented by Kirchner, 2016)."

Farlin had cited Luther and Haitjema (1998) to support effective homogeneity in groundwater aquifers. However, Kirchner suggests that the range of variation in hydraulic conductivity used by Luther and Haitjema was limited in the simulated cases with analytic element model GFLOW, and that the fine-scale heterogeneity assumed in the finite-difference MODFLOW model lacked spatial correlation. Hence, the finding of effective homogeneity, although reasonable for the tested conditions, would not be applicable to many real-world situations where there would be "significant and distinct" heterogeneity.

We believe that the use of compound LPMs could strongly reduce aggregation errors in hydrological systems with "significant and distinct" heterogeneity. For example, we consider a simple case of a catchment split into two parts by two very different rock types that produce waters with very different MTTs; i.e. the most extreme "significant and distinct" heterogeneity one can imagine. A binary LPM is ideally suited for this type of system, and when optimised with suitable data would very effectively separate the young MTT from the old MTT waters in the catchment outflow, and therefore minimise aggregation errors in MTT. If we now consider a catchment split into four parts with two areas of each rock type, the binary LPM when optimised is still very effective for separating the two types of water, while the potential for aggregation errors is smaller. In systems which are split into eight, sixteen, etc. parts the binary LPM retains its effectiveness and the potential for aggregation errors becomes very much smaller because the system starts to look homogeneous at larger scales. There is, of course, a wide range of types of hydrological systems, but the binary LPM is likely to remain effective in cases of "significant and distinct" heterogeneity, which are the ones of most concern for aggregation error.

References
Kirchner JW, Aggregation in environmental systems – Part 1: Seasonal tracer cycles quantify young water fractions, but not mean transit times, in spatially heterogeneous catchments, Hydrology and Earth System Sciences, 20, 279-297, doi:10.5194/hess-20-279-2016, 2016.
Luther KH and Haitjema HM, Numerical experiments on the residence time distributions of heterogeneous groundwatersheds, Journal of Hydrology 207, 1-17, 1998).

---

## Author Response (AR1)

**Editor Initial Decision: Reconsider after major revisions** (25 Oct 2016) by Markus
Hrachowitz: Comments to the Author

**Editor Comment** As you have seen, the three reviewers do, in principle, appreciate the topic you
address in your manuscript but they also raise a few, quite critical issues.

**Reply** We appreciate the work of the Editor and Reviewers on this manuscript.

**Editor Comment 1)** The first is the quality of the presentation of your work. The manuscript is, in
places, written in quite a confusing way with quite some inconsistencies in terminology and the
symbols used, making it difficult to follow. In addition, some of the methods of your analysis are
not explained in sufficient detail and clarity so as to allow the reader to repeat your experiments
and to (ideally) reproduce your results. I think this first issue can (and needs to) be addressed and
solved quite easily.

**Reply** Major changes have been made to the manuscript in response to the Reviewers' comments.
We have changed the structure (it is now structured as: 1. Introduction, 2. Methods, 3. Results, 4.
Case studies from the literature, 5. Discussion, and 6. Summary and Conclusions), but more
importantly we have expanded the description of the calculations in the Methods with two new
sections (Sects. 2.2 and 2.3), and expanded the description in the Results with section 3.1 now
divided into subsections 3.1.1 and 3.1.2. In response to the Reviewers' comments we have also
eliminated inconsistencies in terminology, and have reduced jargon by trying to spell out lumped
parameter model names (exponential model not EM, etc.). The tables have been revamped and
figures improved.

**Editor Comment 2)** The second point is, in my opinion, somewhat more problematic, as it
touches on the core assumptions of the experiment. The reviewers emphasized, and I fully agree
with that, that real systems and their heterogeneity are in general not known. Neither are their
*real* TTDs and MTTs. While it is plausible that compound TTDs will reduce the effect of natural
heterogeneity on the aggregation bias, it is, at this point and in the absence of the necessary
information, impossible to say what a "well-chosen" compound TTD is. If, for example, a *real*
system can be represented by a hyper-exponential TTD, it is still not known, how many
components , i.e. parallel, exponential processes active at different (unknown) time scales, are
necessary to correctly describe this system. Is two components enough? If not, how do we know
and how many do we need? What indeed has been shown in your manuscript is that more
flexibility (i.e. more parameters, and thus potentially more processes) can reduce the problem.
However, in real world systems it is far from obvious how relevant this reduction is, as the
absolute aggregation bias is not known. The bottom line is the question: "What is the real world
reference, which we can compare our models to?"

**Reply** This is one of those propositions that could go round and round without getting anywhere.

To make progress we think one needs to ask "What is the kind of spatial heterogeneity that leads particularly to aggregation bias when simple LPMs are used? We think this is captured in the phrase by Luther and Haitjema (1998), that is that "significant and distinct" heterogeneity has particularly marked effects on the bulk hydrogeological quantities (see also the comments by Farlin and Kirchner, and our response). The calculation procedure uses the most extreme form of this type of heterogeneity (two homogeneous subsystems) and this is fully described by a compound (binary) LPM (or hyper-exponential if the subsystems are exponential models).

When it comes to real-world systems, the optimum compound LPM could in theory be very complicated. But binary LPMs have the capacity of separating young water from old water, which is the crux of the aggregation problem. More complicated LPMs would not add much, and would have more parameters which would be difficult to quantify. This has been the basis for our approach, which is explained in various places (such as Sects. 2.1, 2.2, 5.1, 5.2).

**Editor Comment 3)** In addition, it is stated that, for example a gamma distribution as TTD will result in considerable aggregation error (fig.3). I would argue that such a generalizing statement is not necessarily justified, as the choice of the gamma distribution parameters very much depends on the performance metric and "calibration" strategy used. Fitting a distribution to another one is always a difficult task and many factors will considerable influence the results: how many points are used for fitting (e.g. only the points at the actual time steps used in the model or also some sub-steps?)? How long a tail is considered for fitting (i.e. the longer the tail, the more the performance metric will tend to get the tails right at the cost of a worse representation of the early phases)? Another aspect that is not fully clear, is if we should rather fit the PDFs or the CDFs of the distributions (depends on the type of sample we use)? All these aspects will strongly influence, which TTD is chosen as meaningful.

**Reply** We assume that this comment refers to the Fig. 3 caption "… The relationships between MTTs and tritium concentrations given by the simple models (black curves) are strongly non-linear causing marked differences between the true and apparent MTTs." This refers to mixing of two waters with MTTs of 3 and 197 years respectively and resultant tritium concentrations determined by convolution with constant tritium input. The black curve results from fitting a gamma model with the specified $\alpha$ to the tritium concentrations determined by mixing the two waters. The tritium concentrations of the mixture vary as the proportions of the two waters are varied (shown by the dashed red line). The difference between the true and apparent MTTs is displayed for a 50:50 mixture. This is an idealised system and we have only adjusted the $\beta$ ($=\tau_m/\alpha$) of the gamma model to fit it to the tritium concentrations of the mixture.

**Editor Comment 4)** For example, experimenting a bit with a gamma distribution over 500 time steps, with the squared errors computed from these 500 time steps, I found that a gamma distribution with alpha=0.4 and beta=250 (i.e. MTT=100) gives a sum of squared errors (compared to the compound exponential with MTTs of 3 and 197, as in the manuscript) much lower than for the gamma distribution used in the manuscript (alpha~0.3, beta~150, MTT~45),

while exhibiting essentially *NO* aggregation error (MTTs are almost equal at 100). And, as also pointed out by the reviewers, there are many more different parameter combinations that fit better than the combination given in the manuscript, while exhibiting partly considerably lower aggregation bias.

**Reply** Our calculations for Figs. 3b and 3c in the original ms. had errors which we have now corrected. We have also omitted the calculations for the gamma model with α = 0.3 from the revised paper (Figs. 3, 4, 5, etc.) because they are not applicable for tritium.

10 **Editor Comment 5)** These points are not yet sufficiently well explained and clarified in the author replies to the reviewers. Given their central importance for the analysis and its interpretation, these points need particular attention when revising your manuscript.

**Reply** OK

15 **Reference**

[revised manuscript text omitted]
 relative to the young water threshold (2-3 months for seasonal tracer cycles and 18 years for tritium), unless both components are especially young.

5.2 Does the compound MTT represent the "true" MTT?

The answer to this depends on the nature of the heterogeneity. The type of heterogeneity leading particularly to aggregation bias is heterogeneity that produces flows with very different MTTs. The extreme case is when the catchment or groundwater system is divided into two parts and the flows from each part have very different transit times. This is the case examined by the virtual experiments in Sections 3.1 and 3.2. As noted by Luther and Haitjema (1998), such cases of "few and distinct" heterogeneity have marked effects on the bulk hydrogeological quantities necessary for digital models. We looked for evidence of real-world aggregation error by examining some tritium dating case studies from the literature (Section 4). These were cases where we knew there were "few and distinct" heterogeneities in the catchments. It was not difficult to find such cases, but others with no such heterogeneities could also have been chosen. Many catchments have outflows composed of quickflow and baseflow, which could also have very different MTTs leading to large aggregation errors.

We believe that the use of compound LPMs could strongly reduce aggregation errors in hydrological systems with "significant and distinct" heterogeneity. For example, we consider a simple case of a catchment split into two parts by two

very different rock types that produce waters with very different MTTs; i.e. the most extreme "significant and distinct" heterogeneity one can imagine. A binary LPM is ideally suited for this type of system, and when optimised with suitable data would very effectively separate the young MTT from the old MTT waters in the catchment outflow, and therefore minimise aggregation errors in MTT. If we now consider a catchment split into four parts with two areas of each rock type, the binary LPM when optimised is still very effective for separating the two types of water, while the potential for aggregation errors is smaller. In systems which are split into eight, sixteen, etc. parts the binary LPM retains its effectiveness, but the potential for aggregation errors becomes very much smaller because the system starts to look homogeneous at larger scales. There is, of course, a wide range of types of hydrological systems, but the binary LPM is likely to remain effective in cases of "significant and distinct" heterogeneity, which are the ones of most concern for aggregation error.

**45.2 How much have aggregation effects affected tritium MTTs in past studies?**

Seasonal tracer cycles have been far more widely used to determine MTTs in streams than tritium concentrations. It is clear that many (if not most) studies using seasonal tracer cycles *interpreted with simple LPMs* will have been affected by aggregation bias. because the values of the MTTs determined were between 2-3 months and 4-5 years. But, we contend that tritium studies will have been affected far less, despite aggregation bias also applying to tritium-derived MTTs, because:

(1) Tritium-derived MTTs can only have been affected by aggregation bias if they are greater than 6-16 years (or 6-12 years for α in the range 1 to 10) and were determined using simple LPMs. Many stream studies have shown that the waters are young. Stewart et al. (2010) surveyed most of the 
[revised manuscript text omitted]

[Figure]

[Figure]

**Figure 1<s>1</s>2.** (a) Tritium measurements and model fits for <s>(a)</s> Hangarua Spring <s>and (b) Hamurana Stream</s>, Rotorua, New Zealand. (b) Transit time distributions of the best-fit simulations.

[Figure]

[Figure]

**Figure 12. (a) Tritium measurements and model fits for Hamurana Stream, Rotorua, New Zealand. (b) Transit time distributions of the best-fit simulations.**

[Figure]

**Figure 13. (a) Tritium measurements and simulations for groundwater well M35/3637 in Christchurch, New Zealand. (b) Variation of the goodness-of-fit criterion (sd) with fraction of old water component.**

---

## Referee Report (RR1)

This paper has already been through a number of thoughtful reviews. Here I will focus on a few contentious elements rather than provide a complete review.

This paper aims first to extend the results from Kirchner (2016) on the effects of 'aggregation errors' on transit times estimated from seasonal cycles of stable water isotopes (or other passive tracer) to water ages estimated from tritium observations. I understand the "aggregation error" (as Kirchner used the term) as error in data interpretation arising from a poor choice of probability distribution for the transit time distribution (or 'Lumped Parameter Model' – LPM – as the groundwater community prefers) in certain circumstances. Specifically, it is the case where the sampled water contains contributions from sources whose individual transit time distributions have very different means. If the chosen distribution for the combined sample is unable to represent the breadth (variance and skewness) of the combined transit time distribution it will provide biased estimates of the mean transit time.

If the aims of the paper were simply to point out that spatially variable transit time distributions have similar effects on Tritium observations as Kirchner found they did on stable water isotopes, the paper could be a useful contribution – particularly if the robustness of the <18 year old fraction were more convincingly established. However the paper aims to have further-reaching conclusions regarding the superiority of 'compound' LPMs, and I have some slight issues with these as they currently stand – but I believe these can be remedied with some revision.

The heart of the analysis is the set of virtual experiments described in section 2.2. I think there is a limitation with the virtual experiments described, and which has perhaps led to some of the contentious reviews in the previous round. Specifically, the experiments are conducted by combining two 'simple' LPMs to predict a stream tritium concentration. This concentration is then analyzed by assuming a single 'simple' LPM again. The results show that this leads to significant bias. The authors conclude that 'simple' LPMs are susceptible to aggregation errors, and that a 'compound' LPM is required.

The problem with this is that their choices for the assumed 'simple' LPM are unnecessary constrained. As far as I can tell, the analysis is restricted to cases where both the 'true' LPM used to construct the data, and the 'assumed' LPM used to analyze it, are identical in form: both piston (Dirac delta distributions) or both gamma distributions with identical shape parameters (and always shape parameters greater than 1). The authors do not consider the case where (for example) the 'true' LPMs are exponential, but the 'assumed' LPM is a gamma distribution with a shape parameter less than 1. There is no reason not to do so that I can think of, since in practice we will not know the underlying distributions of the contributing parts. If we do not know the 'true' LPM, we are at liberty to adopt any physically reasonable distribution for the 'assumed' LPM.

In this case, it is possible to choose a shape parameter for which there is zero aggregation error (at least in terms of mean age). The figure below is identical to Figure 3a (where alpha is 1) but with the curves for alpha=3 and 10 (like figures 3 b and 3c) included, along with the curve for alpha=0.24, which was not considered by the authors. For the latter case there is zero aggregation error. The predicted tritium concentration and MTT (100 years) of the 'simple' assumed LPM are both almost exactly that of the 'compound' true LPM.

[Figure]

This seems to contradict the conclusion of the paper that suggests that 'simple' LPM should not be used because they have higher aggregation errors. Here a simple LPM perfectly reproduces the MTT of the aggregate. It is able to do so because the small alpha gives it a large variance and skewness, enabling it to capture the influence of the young and old components. Note that, of course, it is not an accurate representation of the true TTD – however the data (a single tritium observation) is insufficient to determine that. Also, I was only able to choose the 'right' value of alpha because I know what I was aiming for.

As the authors point out a compound distributions is a sensible choice where distinct sources can be identified, and the partitioning of flow between them is known. It is useful to be able to incorporate such information into the model used to interpret the data. However in the absence of such auxillary information there is nothing fundamentally different about 'compound' LPMs that makes them immune to aggregation errors in some way that other type of distribution are not. There are many other distributions that are also reasonable choices even in the case of large

heterogeneity, so long as they have enough flexibility to accommodate larger amounts of skew than an exponential or other type of distribution tested by the authors can.

Of course, in the absence of such auxillary information the compound LPMs and other flexible distributions will have multiple free parameters that must be estimated. As the number of free parameters increase, the ability of the model to reproduce calibration data inevitably increases, without necessarily increasing the physical realism of the calibrated parameters, or any metrics derived from the model (like the mean). This brute fact must always be acknowledged and dealt with by those urging us to adopt more complex models, and is not dealt with here (as several reviewers of the previous version pointed out, to no avail).

In conclusion, this paper makes a useful contribution, and I believe it warrants publication if the authors are able to clarify the issues raised above, and pay a little more attention to the trade-offs associated with increasing the number of free parameters.

---

## Author Response (AR2)

**Response to Editor and Reviewers**

Authors' responses in red

**Editor Report**

Dear authors,

thank you for the revised manuscript. The reviewers and myself appreciate the efforts made. Besides appreciating the general objective and direction of the paper, we think that the revised version is strongly improved with respect to organization, structure and clarity.

Unfortunately, the most critical issue is not yet adequately resolved, as concisely pointed out by reviewer #3 (reflecting the other previous reviewers as well as reviewer #5):
"[...] IF the heterogeneity in the catchment actually coincides with the chosen compound model (that is, if the catchment actually consists of two compartments that each behave according to their chosen age distributions), and IF the parameter estimation procedures actually retrieve parameter values that match the true values for the two compartments, then that model will, indeed, subsume the heterogeneity in the real-world system and eliminate the aggregation bias. But this only says that IF the model is actually the correct model, then it will be correct. The problem is, how can we know whether the model (including the underlying perceptual model) and its parameters are correct? [...]"

As large parts of the interpretation of the manuscript critically hinge on the assumption of a "well chosen" model, it will be necessary to unambiguously substantiate this claim with data.
Another way forward could be, as also pointed out by reviewer #3, to "[...] restrict the claims that are made to ones that actually demonstrated [...]".

Response: We have omitted contentious material and restricted the claims to ones that are actually demonstrated.

[Comment: Of course nobody knows the hydrogeologic flow situation, and never will know it because once one digs into the aquifer to make observations, it is being changed through the process. Therefore, we make observations as to how a tracer moves through the system and deduce a flow model from those observations, also taking into account other hydrogeologic information. With increasing complexity of the tracer input(s), increasingly complex flow processes can be resolved. That is the whole nature of the work. Tritium input and output records that cover the full range of the bomb pulse (like those for the case studies previously given in this paper, and for example by Blavoux et al., 2012) are excellent in terms of their large ranges of variation, and superior to those used for rainfall/runoff and stable isotope modelling. Hence, kneejerk application of lessons from those latter types of modelling are probably not justified. This will not apply to tritium samples collected in places where there is not already a record of past tritium concentrations.]

In addition, the question concerning parameter uncertainty issue (or "how sure are you that the chosen parameters are indeed the most appropriate ones?") has not at all been addressed, raising questions about the robustness of the chosen parameters.

Response: We have removed the entire second part of the paper and plan to submit it somewhere when we have done a full GLUE analysis (a la Gallart et al., 2016) of the simple and compound models.

[Comment: We have applied the GLUE analysis for the EPM model to the case studies previously included in the paper and found that the best-fit and likelihood-weighted MTTs and their uncertainty ranges are in good correspondence.]

The above points need to be addressed in detail before the manuscript can be considered for publication. I would thus be glad if the authors invested some more effort to develop the manuscript to the point.

Best regards,

Markus Hrachowitz

**Report#1**

This paper has already been through a number of thoughtful reviews. Here I will focus on a few contentious elements rather than provide a complete review.

This paper aims first to extend the results from Kirchner (2016) on the effects of 'aggregation errors' on transit times estimated from seasonal cycles of stable water isotopes (or other passive tracer) to water ages estimated from tritium observations. I understand the "aggregation error" (as Kirchner used the term) as error in data interpretation arising from a poor choice of probability distribution for the transit time distribution (or 'Lumped Parameter Model' – LPM – as the groundwater community prefers) in certain circumstances. Specifically, it is the case where the sampled water contains contributions from sources whose individual transit time distributions have very different means. If the chosen distribution for the combined sample is unable to represent the breadth (variance and skewness) of the combined transit time distribution it will provide biased estimates of the mean transit time.

Response: Agreed

If the aims of the paper were simply to point out that spatially variable transit time distributions have similar effects on Tritium observations as Kirchner found they did on stable water isotopes, the paper could be a useful contribution – particularly

if the robustness of the <18 year old fraction were more convincingly established. However the paper aims to have further-reaching conclusions regarding the superiority of 'compound' LPMs, and I have some slight issues with these as they currently stand – but I believe these can be remedied with some revision.

Response: OK

The heart of the analysis is the set of virtual experiments described in section 2.2. I think there is a limitation with the virtual experiments described, and which has perhaps led to some of the contentious reviews in the previous round. Specifically, the experiments are conducted by combining two 'simple' LPMs to predict a stream tritium concentration. This concentration is then analyzed by assuming a single 'simple' LPM again. The results show that this leads to significant bias. The authors conclude that 'simple' LPMs are susceptible to aggregation errors, and that a 'compound' LPM is required.

The problem with this is that their choices for the assumed 'simple' LPM are unnecessary constrained. As far as I can tell, the analysis is restricted to cases where both the 'true' LPM used to construct the data, and the 'assumed' LPM used to analyze it, are identical in form: both piston (Dirac delta distributions) or both gamma distributions with identical shape parameters (and always shape parameters greater than 1). The authors do not consider the case where (for example) the 'true' LPMs are exponential, but the 'assumed' LPM is a gamma distribution with a shape parameter less than 1. There is no reason not to do so that I can think of, since in practice we will not know the underlying distributions of the contributing parts. If we do not know the 'true' LPM, we are at liberty to adopt any physically reasonable distribution for the 'assumed' LPM.

Response: The procedure described above was followed in order to produce Figs. 3a-d (Sect 3.1.1) which were for explanatory purposes. For determining the actual (virtual) aggregation effects in Figs. 5, 6 (Sect. 3.1.3) the shape factor ($\alpha$) was allowed to change along with the MTT to produce the best-fit of the simple LPM to the tritium data.

In this case, it is possible to choose a shape parameter for which there is zero aggregation error (at least in terms of mean age). The figure below is identical to Figure 3a (where alpha is 1) but with the curves for alpha=3 and 10 (like figures 3 b and 3c) included, along with the curve for alpha=0.24, which was not considered by

5  the authors. For the latter case there is zero aggregation error. The predicted tritium concentration and MTT (100 years) of the 'simple' assumed LPM are both almost exactly that of the 'compound' true LPM.

[Figure]

10  Response: This is an interesting observation. The dispersion model also has large skewness when the dispersion parameter has high values. This is referred to in the Discussion (Sect. 4.1).

This seems to contradict the conclusion of the paper that suggests that 'simple' LPM should not be used because they have higher aggregation errors. Here a simple LPM

15  perfectly reproduces the MTT of the aggregate. It is able to do so because the small alpha gives it a large variance and skewness, enabling it to capture the influence of

the young and old components. Note that, of course, it is not an accurate representation of the true TTD – however the data (a single tritium observation) is insufficient to determine that. Also, I was only able to choose the 'right' value of alpha because I know what I was aiming for.

5  As the authors point out a compound distributions is a sensible choice where distinct sources can be identified, and the partitioning of flow between them is known. It is useful to be able to incorporate such information into the model used to interpret the data. However in the absence of such auxillary information there is nothing fundamentally different about 'compound' LPMs that makes them immune

10  to aggregation errors in some way that other type of distribution are not. There are many other distributions that are also reasonable choices even in the case of large heterogeneity, so long as they have enough flexibility to accommodate larger amounts of skew than an exponential or other type of distribution tested by the authors can.

Response: We agree that both simple and compound LPMs can be free of aggregation error or conversely be affected by aggregation error. We have removed this part of the paper.

Of course, in the absence of such auxillary information the compound LPMs and

20  other flexible distributions will have multiple free parameters that must be estimated. As the number of free parameters increase, the ability of the model to reproduce calibration data inevitably increases, without necessarily increasing the physical realism of the calibrated parameters, or any metrics derived from the model (like the mean). This brute fact must always be acknowledged and dealt with

25  by those urging us to adopt more complex models, and is not dealt with here (as several reviewers of the previous version pointed out, to no avail).

Response: We agree and have stated this in the revised ms (Sect. 4.2).

30  In conclusion, this paper makes a useful contribution, and I believe it warrants publication if the authors are able to clarify the issues raised above, and pay a little more attention to the trade-offs associated with increasing the number of free

parameters.

Response: We thank the Referee for his constructive input and addressed his comments in the revised manuscript.

**Report#2**

The revised version of this manuscript is an improvement. It resolves several of the issues that were raised in the first round of reviews.

Unfortunately, there are also important areas where the manuscript has hardly changed in response to major issues identified by the reviewers as well as the editor. To the extent that aspects of the paper haven't changed, my assessments of those aspects also remain unchanged.

The paper's central claim is that "well-chosen compound lumped parameter models should be used as they will reduce potential aggregation errors due to the application of simple lumped parameter models". This claim, from the abstract, is stated much more strongly in the text, e.g., "... the binary models have very much less potential for aggregation bias than the simple models".

These claims must be demonstrated with evidence, and not simply asserted.

Response: This is not the central claim of the paper. The central claim is "that MTTs derived from tritium concentrations in streamflow are just as susceptible to aggregation bias as those from seasonal tracer cycles. Likewise, groundwater wells or springs fed by two or more water sources with different MTTs will also show aggregation bias." (Quoted from the abstract – first paragraph.) This is also what is given in the title. The revised paper deals only with this topic.

The reviewer's asserted "central claim" above was always subsidiary and has now been removed.

To the extent that evidence is offered, it is mostly true by definition, since: a) a "well-chosen" model is operationally defined in the paper as one that is consistent with the perceptual geological model, which is assumed to be correct b) the authors define the "true" MTT as the MTT of the compound model for the purposes of the results in Section 4, and c) the authors measure the aggregation error as the deviation of the simple model MTT from the compound model MTT (which is considered to be the "true" MTT).

Under this system of definitions, of course the compound models are better than the simple models... because the compound models are defined from the start as being correct and therefore having no aggregation error!

Response: This is not relevant now.

Of course IF the heterogeneity in the catchment actually coincides with the chosen compound model (that is, if the catchment actually consists of two compartments that each behave according to their chosen age distributions), and IF the parameter estimation procedures actually retrieve parameter values that match the true values for the two compartments, then that model will, indeed, subsume the heterogeneity in the real-world system and eliminate the aggregation bias. But this only says that IF the model is actually the correct model, then it will be correct.

Response: Agreed

The problem is, how can we know whether the model (including the underlying perceptual model) and its parameters are correct?

The authors' position seems to be that as long as the model fits the perceptual model and the data, it is correct. But long experience in hydrological modeling has shown that models with many parameters can often fit the available data, whether or not those models are structurally correct. In many cases, the models will also fit well for multiple, widely differing parameter sets – in other words, the parameter values will not be identifiable, even though the models fit the data nicely.

Response: Not relevant now.

To justify the statements made in the manuscript, one would need to present evidence that if the model fits the perceptual model and the data, then the model structure and the parameter values are guaranteed to be correct – that is, that it is not possible for the real-world structure, or its parameter values, to be significantly in error. No evidence is presented to meet this burden of proof.

There is another way forward, and that is to restrict the claims that are made to ones that actually demonstrated. For example, the authors have analyzed several cases where there is geological evidence for a multicomponent catchment system, and in those cases, compound LPM's that mirror these perceptual models fit the tritium data better than simple LPM's do. So that is a claim that could be made in the paper.

Response: Not relevant now.

But if the manuscript is to make broader claims (for example, about whether compound LPM's will reduce or eliminate aggregation errors), then those claims need to be substantiated with evidence. That evidence also needs to be nontrivial; it is not enough to posit a system of two exponential distributions, and then assert that a compound LPM with two exponential distributions would fit this system, and would have no aggregation bias. That is simply irrelevant to the problem we face in the real world, which is that we do not already know the "right answer", including the structure of the system we are trying to analyze.

The manuscript demonstrates an important point, which is that where we already have geological information about aquifer partitioning, then that information can be very useful in constraining transit time models. Where we have that information, we don't need to rely on tracers alone. Another way to say it is, if the transit time models agree with the geological information and the tracers, then they strengthen our confidence in both of them. That is a useful point to make.

Response: Not relevant now.

But the generalization that "well-chosen" models are always preferred is either true by definition (of course "well-chosen" is better than "poorly chosen"), or else it leaves open the critical issue: how do we know when our models are "well chosen"? Is two compartments "well chosen", or just one? Or six? How should we decide?

Response: Not relevant now.

More complex models, with more adjustable parameters, will almost always fit data sets better, but the parameters themselves may be highly uncertain. The critical question that the manuscript must come to grips with is: how well can the parameters be constrained? What are their uncertainties? Holding all the other parameter values constant and just varying one of them is not a valid way to estimate parameter uncertainty, unless those other values are actually known to have their assigned values. But as far as I can tell (the manuscript doesn't say, and that is itself a problem), either this, or nothing at all, is what has been done.

Response: Not relevant now.

My previous review said that there was no excuse for not properly analyzing parameter uncertainty, and the response was that this had not been done in tritium papers until very recently. The fact that it has rarely been done before is no excuse for not doing it now. The only alternative is to remove all claims about whether parameters (including the MTT's that are derived from them) are well constrained.

Response: Not relevant now.

Some particularly problematic passages are quoted below (this is an incomplete list):

"We believe that the use of compound LPMs could strongly reduce aggregation errors in hydrological systems with "significant and distinct" heterogeneity. For example, we consider a simple case of a catchment split into two parts by two very different rock types that produce waters with very different MTTs; i.e. the most extreme "significant and distinct" heterogeneity one can imagine. A binary LPM describes this type of system, and when optimised with suitable data would very effectively separate the young MTT from the old MTT waters in the catchment outflow, and therefore minimise aggregation errors in MTT. "

This assumes the existence of "suitable data", but gives no criteria by which to determine whether data are "suitable". No analysis is presented to support any of the statements in the passage. They may be true, or they may not, but unless they can be supported by evidence they should be removed.

Response: Not relevant now.

"If we now consider a catchment split into four parts with two areas of each rock type, the binary LPM when optimised is still very effective for separating the two types of water, while the potential for aggregation errors is smaller. In systems which are split into eight, sixteen, etc. parts the binary LPM retains its effectiveness, but the potential for aggregation errors becomes very much smaller because the system starts to look homogeneous at larger scales."

No evidence is presented to substantiate these statements. In Kirshner [2016], splitting the model system into more components does not reduce the aggregation bias, which suggests that the statements made here are not correct.

Response: Not relevant now.

"There is, of course, a wide range of different types of hydrological systems, but the binary LPM is likely to remain effective in cases of "significant and distinct" heterogeneity, which are the ones of concern for aggregation error."

What do "remain effective" and "significant and distinct" mean?

Response: Not relevant now.

It is clear that many (if not most) studies using seasonal tracer cycles interpreted with simple LPMs will have been affected by aggregation bias. But we contend that tritium studies will have been affected less, despite aggregation bias also applying to tritium-derived MTTs, because many of the tritium studies in the literature applied compound models calibrated by fitting to time series of tritium measurements rather than or as well as using simple LPMs. Provided the compound LPMs were well-chosen based on the characteristics of the catchments, they will produce more accurate TTDs than the simple LPMs and therefore will reduce aggregation

bias on MTTs.

These statements may or may not be true, but they are not supported by any clear evidence that I can find in the manuscript.

5  Response: Not relevant now.

"A good example is the study of Blavoux et al. (2013) describing the interpretation of an exceptionally long and very detailed record of tritium concentrations from the Evian-Cachat Spring in France. The tritium record was much too complicated to be fitted by a simple LPM. Instead, the detailed records of input and output allowed accurate specification of a combined model comprising of exponential ($\tau m = 8$ yr) and dispersion ($\tau m = 60$ yr)
10  models in series, with a small bypass flow in parallel with them, followed by a piston flow model ($\tau m = 2.5$ yr) in series giving an overall $\tau m$ of 70 yr. The combined model was closely related to the hydrogeology of the area and produced an accurate TTD for the average stationary state of the system, so there is little possibility of aggregation bias."

15  The Blavoux et al. study relied on samples taken over decades, including ones that nicely traced out the bomb pulse. It is a nice study, but is completely misleading as an indication of what will be possible with tritium data, except in the very few places where records go back that far.

Response: The quoted paragraph has now been omitted. We agree that the Blavoux et al. approach can only be applied to places where there is sufficient past tritium data (such as a number of sites in New Zealand). [But we
20  are puzzled, was the Blavoux et al., 2012, approach also "inexcusable" (in the reviewer's terminology) or is this word only applied to our approach although we do the same thing?]

A central problem with the paper is that it gives the reader the impression that fitting more complex LPM's will solve all kinds of problems. But now and into the future, with the bomb pulse gone, there will be very little tritium variation available to fit even simple models to, let alone more complex ones.

25  Response: It was not our intention to give such an impression. The paper states in several places that identifying parameters of LPMs will be difficult in the future.

"Compound LPMs are to be preferred, but often there can be considerable difficulty in uniquely quantifying the parameters especially if the output data is limited."

30  The statement that compound LPM's "are to be preferred" is unsubstantiated, except perhaps as a statement of the authors' preferences. And the second part of the sentence effectively cancels the first, since (although this is not stated), the "difficulty in uniquely quantifying the parameters" becomes geometrically more difficult as the number of parameters grows. The manuscript needs to show that the "considerable difficulty" is not a problem in the cases presented here, and also in the types of situations where tritium is likely to be used in the future, when
35  there will be no bomb pulse to work with.

Response: Not relevant now

"We find that MTT aggregation errors are small when the component waters have similar MTTs to each other. On the other hand, aggregation errors can be large when very young water components are mixed with older components."

Unlike in the case of seasonal tracer cycles, with tritium the absolute ages (such as "very young") are not relevant because tritium decays exponentially. Aggregation errors can arise in tritium whenever the component ages span

a large range, regardless of where that range is centered. The exponential decay curve implies that the aggregation error depends on the ratios of the ages, not their absolute values.

Response: Not relevant now. This argument is mostly correct, but ignores the effect of mixing models.

Even if we have no "very young" water at all, for example, adding some much older (tritium-depleted) water can potentially lead to a large aggregation error, depending on how much older that water is. If, for example, we added 50% zero-tritium water to our mixture, we would decrease the average 3H concentration by half and therefore increase the model age by 12.3 years... but we could increase the ACTUAL mean age by hundreds or thousands of years, depending on how old the zero-tritium water actually is.

The authors point out in section 4.4 that additions of tritium-free water will often have a very small effect on model fits to tritium time series, but (although they don't emphasize this), this could have a potentially huge effect on what the true mean transit time is (because the tritium-free water could be 200, or 2000, or 20,000 years old). Thus the true mean transit time can become decoupled from the mean transit time estimated from tritium. This is a clear aggregation error, and it should be identified as such. The same potential aggregation error exists (but is not mentioned) in the other case examples as well. Every TTD model in the paper, both simple and compound, is vulnerable to it.

Response: This is a well-known problem mentioned in Sect. 4.1 (and that was illustrated in the now-removed section 4.5).

"In general, well-chosen compound lumped parameter models should be used as they will reduce potential aggregation errors due to the application of simple lumped parameter models. An opportunity to determine a realistic compound lumped parameter model is given by matching simulations to time series of tritium measurements (underlining the value of long series of past tritium measurements), but such results should be validated by reference to the characteristics of the hydrological system to ensure that the parameters found by modelling correspond to reality."

Again, the value of comparing model assumptions to "the characteristics of the hydrological system" (presumably determined independently?) is clear, but claims that any model is "realistic" – and, more generally, that the approach outlined here is a reliable guide to model "realism" – need to be supported with evidence.

Response: Not relevant now.

Minor items:

The claim that tritium measurements are usable over a 200-year time range should be substantiated. 200 years is almost 17 half-lives, during which tritium concentrations will decay to 10^-5 of their original values. Even assuming we start with 5,000 TU (the peak of the bomb pulse), this will decay to about 0.05 TU after 200 years. Even if such a measurement is analytically feasible, it is hard to see how it is useful in practice, because contamination with even a tiny bit of younger water would obscure the tiny traces of the 200-year-old water. (And remember, this is starting from the strongest possible tritium signal, the peak of the bomb pulse.)

Response: This is very easy to substantiate. To adopt the rhetoric of the reviewer, there is no excuse for the calculation made by the reviewer here. His/her calculation is based on the piston flow mixing model (an assumption that has dogged the use of all radioactive isotopes and some chemicals (e.g. tritium, carbon-14, CFCs, etc.) for water dating over much of their history. The PFM does not apply to very many water flow situations at

all (if any) and particularly not when older waters are involved. Waters with MTTs of 200 years will often have measurable tritium concentrations when appropriate mixing models are used (see table). For example, using a mixing model which has been found to apply to many situations (EPM with parameter f = 0.75), 200-year-old MTT water would have tritium concentrations of 0.07 TU in the Southern Hemisphere (measurement error about ±0.02 TU) and 2.65 TU in the Northern Hemisphere (measurement error about ±0.1 TU). Note that such ages are mixing model-dependent (as are all ages calculated using tritium and other isotopes/chemicals), and that the tritium observed in the sample would have come almost entirely from the younger water in the mixture which would have contained bomb pulse tritium.

Table 1. Tritium concentrations in 2017 for water with mean transit times of 200 years calculated using an exponential piston flow mixing model. Pre-bomb tritium inputs of 2 TU in the Southern Hemisphere, and 8 TU in the Northern Hemisphere were assumed.

| Exponential piston flow model (EPM) | Alternative model name | Tritium concentration (TU) in 2017 of 200-year MTT water | |
|---|---|---|---|
| Parameter f | | Southern Hemisphere | Northern Hemisphere |
| 0 | PFM | 0.00002 | 0.0001 |
| 0.5 | | 0.0008 | 0.004 |
| 0.75 | | 0.07 | 2.65 |
| 1.0 | EM | 0.25 | 3.12 |

The statement that there is no aggregation bias when the MTT's of the two components are the same (see the abstract, and Section 5.1, for example) are rather trivial. If there is no heterogeneity, then of course no bias can result from it.

Response: Not relevant.

On page 5, line 19, the dimensions of beta are wrong.

Response: Agreed.

The conclusions are mostly a word-for-word repetition of the abstract. If the authors don't have anything further to say, then there is no need for the conclusions. There is need to print the same sentences in two different places.

Response: Both conclusions and abstract have been revised.

**Report#3**

General Comments:
I was not a reviewer in the initial round of reviews and therefore I cannot judge exactly how much improvement has been made in the revision process so far. Looking at the initial reviews and the revised manuscript I can however see that the authors did make some efforts to resolve problems that were brought forward.
I can also say that the manuscript is well-written in terms of style and structure.

The part I am struggling with is more a general one. The authors conclude that using a well-chosen compound model solves the issue of spatial aggregation errors. Still, they never really show how to 'choose well'. They demonstrate which of the models causes the smallest aggregation error, but what does that really mean? If a gamma model with shape parameter 10 represents the dynamics of the real-world hydrology best, this model should still be used regardless of the fact that it causes more aggregation errors than a gamma model with shape parameter 1 (or is that a wrong assumption?).

Response: We agree that any simple LPM should be used if it represents the dynamics of the real-world hydrology. There would be no aggregation error in that case. Simple LPMs should be tried in any case to see if they can fit the data.

Also, the f parameter bothers me. How can it be constrained and how does it influence the results? How does it affect the young water fractions and the young water thresholds? The authors often state that they strongly believe that the use of compound LPMs could strongly reduce aggregation errors. What about a little demonstration of whether the model actually works. If you mix two waters of different ages in a specified ratio (not only 50:50), can you define both MTTs correctly using the binary model or do you get lost in equifinality problems when using 5 fitting parameters? You could set up a model scenario with known inputs and outputs to demonstrate that your method actually works.

Response: We think the reviewer means b (the fraction of the younger water component in the mixture) not f.

This is an interesting point, although not now applicable to our revised paper. We tried fitting a DEPM model to the tritium values resulting from the combination of two EPM models (EPM1 and EPM2) using the Kaitoke (Southern Hemisphere) tritium input function. This procedure is equivalent to a situation where one had a time sequence of 75 tritium samples (one for each year from 1940 to 2015), i.e. a very good record of the tritium in the stream or groundwater. The fitting procedure using the Solver Add-In of the Excel spreadsheet gave mixed results, but generally was unable to recapture uniquely all five parameters of the DEPM (MTT1, f1, MTT2, f2 and b). Likewise with fitting four parameters, i.e. after fixing b based on measurements in the catchment or groundwater system. Varying three parameters (the MTTs and b), on the other hand, gave unique solutions which recovered the original parameter values. This required fixing the two f values at their original values. Work by Gallart et al. (2016) on the uncertainties of the parameters of the exponential piston flow model had shown that the f parameter had remarkably low identifiability (i.e. almost any value would do), whereas the MTT was usually much more clearly identifiable.

The procedure we have usually adopted when fitting a time sequence of samples is to initially insert parameter values deemed likely from the hydrogeology of the system (and our previous experience) and try to get close to a good fit by manually tweaking the parameter values. Then apply the Solver fitting procedure to locate the best-fit position or positions. However, there can be several parameter combinations giving fits of various qualities (i.e. equifinality), so further experimentation to explore the full parameter spaces is often necessary.

Another issue is the exclusion of anything non-stationary from the discussion. How would the omnipresent time-variance influence the results? Are all the assumptions (and conclusions) still valid if the fast component (or both components) change velocity over time? How would aggregation in time affect the apparent mTTs?

Response: Our work did not include the effect of non-stationarity on aggregation error.

Is the determination of the young water fraction threshold simply done by minimizing the difference between apparent and true young water fractions? You do not specify this method. So what exactly is the young water fraction? Is it a random number fulfilling some mathematical/statistical requirements – like for example the value where the young water transit time distribution has its peak? What is it actually good for, if its value ranges from

0.1 to 18 years? You state that certain thresholds are important (1 year, 60 years) but you do not show/test whether they can be predicted accurately (I guess they cannot because they are far from the value of 18 years). Maybe you can discuss some of these issues in more detail.

In essence, what I want to say is that this manuscript would profit considerably from going even further beyond
5 reproducing Kirchner's 2016 paper with tritium.

Response: The young water threshold of 18 years was determined by trial and error, with the requirement that the apparent and true young water fractions not deviate from each other by more than 10%. 18 years was about the highest value for which this requirement was met.

Specific Comments:
10 Page 13, Line 21: What do you mean by 'using the DM and DEPM together or the EPM and DDM'? This is confusing me. This sentence has been removed. (It was just a suggestion that the EPM and DM models should not be mixed for that particular case study. If one starts with the EPM then it should be compared with the DEPM not the DDM, when comparing results from single and binary LPMs.).

The young water fraction in Table 2 should not have dimensions of (yr). Agreed

[revised manuscript text omitted]